# Projection-Free Algorithms for Minimax Problems

**Khanh-Hung Giang-Tran** [1]   **Soroosh Shafiee** [1]   **Nam Ho-Nguyen** [2]

## Abstract

This paper addresses constrained smooth saddle-point problems in settings where projection onto the feasible sets is computationally expensive. We bridge the gap between projection-based and projection-free optimization by introducing a unified *dual dynamic smoothing* framework that enables the design of efficient *single-loop* algorithms. Within this framework, we establish convergence results for *nonconvex-concave* and *nonconvex-strongly concave* settings. Furthermore, we show that this framework is naturally applicable to *convex-concave* problems. We propose and analyze three algorithmic variants based on the application of a linear minimization oracle over the minimization variable, the maximization variable, or both. Notably, our analysis yields *anytime* convergence guarantees without requiring a pre-specified iteration horizon. These results significantly narrow the performance gap between projection-free and projection-based methods for minimax optimization.

## 1. Introduction

First-order methods for constrained smooth optimization problems traditionally fall into two classes based on their oracle complexity. The standard approach uses projection-based algorithms, such as projected gradient descent and its accelerated variants (Nesterov, 1983; 2013b), which are highly efficient when the projection onto the feasible set is computationally cheap. However, when this projection is expensive, projection-free algorithms, like the Frank-Wolfe algorithm (Frank et al., 1956), provide a compelling alternative, replacing the costly projection with a more tractable linear minimization oracle (Jaggi, 2013; Braun et al., 2025).

[1]Cornell University [2]The University of Sydney. Correspondence to: Khanh-Hung Giang-Tran <tg452@cornell.edu>, Soroosh Shafiee <Shafiee@cornell.edu>, Nam Ho-Nguyen <nam.ho-nguyen@sydney.edu.au>.

*Proceedings of the 43rd International Conference on Machine Learning*, Seoul, South Korea. PMLR 306, 2026. Copyright 2026 by the author(s).

This constrained setting belongs to the broader composite optimization framework where the objective is the sum of smooth and nonsmooth functions. Proximal methods are the standard choice for these problems. For example, proximal gradient descent (Daubechies et al., 2004; Parikh & Boyd, 2014) and its accelerated variants (Nesterov, 2013a; Beck & Teboulle, 2009) combine a gradient step on the smooth part with a proximal step on the nonsmooth term. When the proximal operator is computationally expensive, the generalized conditional gradient algorithms offer a projection-free alternative for the composite setting (Beck, 2017; Lan, 2020).

While this division between projection-based and projection-free methods is well-understood for standard minimization, its implications for saddle-point optimization are less developed. This paper bridges this gap by addressing the constrained smooth saddle-point problem

$$\min_{x \in \mathcal{X}} \max_{y \in \mathcal{Y}} \ \mathcal{L}(x, y), \tag{1}$$

where the feasible sets $\mathcal{X} \subseteq \mathbb{R}^{d_x}$ and $\mathcal{Y} \subseteq \mathbb{R}^{d_y}$ are compact and convex, and the payoff function $\mathcal{L} : \mathbb{R}^{d_x} \times \mathbb{R}^{d_y} \to \mathbb{R}$ is smooth in both variables, concave in $y$, and potentially nonconvex in $x$. The majority of existing algorithms are projection-based or, more generally, of the proximal type. Despite the success of projection-free methods in standard minimization problems, the development of their counterparts for the minimax problems has lagged significantly. This work aims to introduce and analyze *single-loop* algorithms that rely on linear minimization oracle (LMO) over $\mathcal{X}$, $\mathcal{Y}$ or both. When LMO is not used over a set, we will use projection oracle (PO) over it. Our analysis covers convex-concave (C-C), nonconvex-concave (NC-C), convex-strongly concave (C-SC) and nonconvex-strongly concave (NC-SC) problems. We do not consider the strongly convex-strongly concave (SC-SC) setting, as it is well understood.

The remainder of Section 1 surveys related literature and highlights the primary contributions of this work. Section 2 introduces our single-loop framework for the LMO-LMO setting. In Sections 3 and 4, we extend this framework and analysis to the LMO-PO and PO-LMO settings, respectively. Convergence results in the main text are reported using standard big-$O$ and $\widetilde{O}$ notations, where the latter suppresses logarithmic factors for clarity. Detailed proofs, including explicit convergence constants, are provided in the Appendix.

## 1.1. Related Works

### 1.1.1. LMO-BASED METHODS

**Convex-Concave Payoff Function.** Projection-free methods for minimax problems date back to Hammond (1984), who showed that a single-loop FW algorithm with an open-loop stepsize converges *asymptotically* for variational inequality (VI) problems over strongly convex sets. While several works have since established non-asymptotic complexity bounds using *multi-loop* or *nested structures* to approximate proximal and extragradient steps (He & Harchaoui, 2015; Chen et al., 2020; Baghbadorani et al., 2025), the development of single-loop algorithms has focused on achieving competitive rates by exploiting specific geometries or regularity conditions. Gidel et al. (2017) pioneered this direction by establishing the first linear convergence rates for SC-SC problems with interior solutions, as well as for C-C settings where the constraint sets are strongly convex and gradients are lower-bounded. Furthermore, Chen & Mazumdar (2024) addressed last-iterate convergence, proving an $O(\epsilon^{-2})$ rate for a single-loop *generalized* FW method using strongly convex and coercive regularization in the monotone VI setting. In this work, we bridge the gap between these specialized settings and general minimax problems by introducing and analyzing both one-sided and two-sided projection-free, single-loop algorithms.

**Nonconvex-Concave Payoff Function.** In this setting, complexity results are more recent and typically involve more complex loop structures. Nouiehed et al. (2019) provided a *multi-loop* one-sided projection-free method that achieves an $\epsilon$-game stationary solution in $O(\epsilon^{-3.5})$ iterations. This was followed by Boroun et al. (2023), who established the first *single-loop* fully projection-free analysis for NC-C problems. Their algorithm achieves an $\epsilon$-gap stationary solution within $O(\epsilon^{-6})$ iterations, which improves to $O(\epsilon^{-4})$ for the NC-SC setting, provided the maximization constraint set is strongly convex. Additionally, Boroun et al. (2023) proposed a one-sided projection-free variant that achieves a faster rate $O(\epsilon^{-4})$ for NC-C setting, and $O(\epsilon^{-2})$ for NC-SC. In this work, we provide a unified analysis for single-loop algorithms that not only matches these rates but also ensures *anytime* convergence without requiring a fixed iteration horizon. Furthermore, by exploiting the geometry of the constraint sets, we establish improved complexity bounds, significantly narrowing the gap between projection-free and projection-based methods in the nonconvex regime.

### 1.1.2. PO-BASED METHODS

**Convex-Concave Payoff Function.** Nedić & Ozdaglar (2009) analyzed the projected subgradient descent-ascent algorithm, establishing a convergence rate of $O(\epsilon^{-2})$. In the smooth setting, Nemirovski (2004) introduced the mirror-prox algorithm, which achieves a faster $O(\epsilon^{-1})$ gradient

complexity, a rate proven optimal by Ouyang & Xu (2021). The two most prominent projection-based algorithms for the smooth setting are the extragradient (EG) method (Korpelevich, 1976) and optimistic gradient descent ascent (OGDA) (Popov, 1980). Both EG and OGDA can be interpreted as approximate variants of the proximal point method (Mokhtari et al., 2020), allowing them to recover the optimal $O(\epsilon^{-1})$ rate. In this work, we establish convergence guarantees for algorithms based on various combinations of LMO and PO.

**Nonconvex-Concave Payoff Function** Early mirror-descent–type algorithms achieved complexity bounds of $\widetilde{O}(\epsilon^{-2})$ for NC-SC and $\widetilde{O}(\epsilon^{-6})$ for NC-C when targeting an $\epsilon$-stationary point of the primal function (Rafique et al., 2022; Lin et al., 2020a). These rates were later refined through proximal-point and accelerated frameworks. Namely, Thekumprampil et al. (2019) achieved $\widetilde{O}(\epsilon^{-3})$ using a proximal dual implicit method, while Lin et al. (2020b) further improved this to $\widetilde{O}(\epsilon^{-2.5})$ via an inexact proximal point approach. Beyond primal stationarity, alternative measures such as $\epsilon$-game stationarity have been investigated. For instance, Nouiehed et al. (2019) proposed a multi-step gradient descent ascent method achieving $O(\epsilon^{-3.5})$ complexity. More recently, Lu et al. (2020) introduced a hybrid block successive approximation method that harmonizes these perspectives, yielding improved rates of $O(\epsilon^{-2})$ and $O(\epsilon^{-4})$ for the NC-SC and NC-C settings, respectively. In this work, we analyze both one-sided and two-sided LMO-based algorithms within this setting.

### 1.1.3. OTHER SETTINGS

**Bilinear Payoff Functions.** Bilinear problems are typically addressed via projection-based methods achieving linear convergence (Liang & Stokes, 2019; Azizian et al., 2020) or specialized projection-free methods reaching $O(\epsilon^{-1})$ (Kolmogorov & Pock, 2021). While applicable, our general C-C analysis does not explicitly exploit bilinear structures and thus does not recover these specialized optimal rates.

**Lagrangian-Based Payoff Functions.** Lagrangian of affinely constrained convex problems are often solved via projection-based methods (Chambolle & Pock, 2011; Lan & Monteiro, 2016) or specialized projection-free augmented Lagrangian frameworks (Yurtsever et al., 2019; Lan et al., 2021; Asgari & Neely, 2022). Our general C-C approach provides a unified analysis for these cases but does not exploit their specific structure to reach specialized rates.

**Variational Inequalities.** Recent methods for functional-constrained VIs include ADMM-interior-point hybrids (Yang et al., 2023), stochastic Korpelevich variants using random feasibility updates (Chakraborty & Nedić, 2025), and primal approaches relying on local linear approximations of the feasible set (Zhang et al., 2025). While these methods are restricted to the C-C setting, we extend the analysis to the more general NC-C case.

*Table 1.* Convergence Guarantees in Nonconvex-Concave Setting.

| TYPES OF ORACLES | | SETTINGS | | REQS. |
|---|---|---|---|---|
| | | NC–C | NC–SC | |
| LMO – LMO | R-PDCG | $O(\epsilon^{-6})$ | $O(\epsilon^{-4})$ | SC $\mathcal{Y}$ |
| | OURS | $O(\epsilon^{-6})$ | $O(\epsilon^{-4})$ | |
| | | $O(\epsilon^{-5})$ | $O(\epsilon^{-3})$ | SC $\mathcal{Y}$ |
| LMO – PO | CG-RPGA | $O(\epsilon^{-4})$ | $O(\epsilon^{-2})$ | |
| | OURS | $O(\epsilon^{-4})$ | $O(\epsilon^{-2})$ | |
| PO – LMO | OURS | $O(\epsilon^{-6})$ | $O(\epsilon^{-4})$ | |
| | | $O(\epsilon^{-5})$ | $O(\epsilon^{-3})$ | SC $\mathcal{Y}$ |

## 1.2. Contributions

The primary challenge in solving the minimax problem (1) is the potential nonsmoothness of the primal objective function

$$f(x) := \max_{y \in \mathcal{Y}} \mathcal{L}(x, y). \qquad (2)$$

Even when the payoff function $\mathcal{L}$ is smooth, the maximization process can yield a nonsmooth $f$, complicating the direct application of standard first-order algorithms.

To overcome this, we introduce a unified *dual dynamic smoothing* framework. Drawing inspiration from classical smoothing techniques (Nesterov, 2005; Beck & Teboulle, 2012) and recent advances in minimax optimization (Xu et al., 2023; Boroun et al., 2023), we approximate the original problem using the regularized payoff function

$$\mathcal{L}_\beta(x, y) := \mathcal{L}(x, y) - \frac{\beta}{2}\|y - y_0\|^2,$$

where $y_0 \in \mathcal{Y}$ is a reference point and $\beta \geq 0$ is a dynamic smoothing parameter. This regularization yields a smooth primal objective, enabling efficient single-loop optimization.

Our main contributions are as follows.

◇ **Unified Single-Loop Framework.** We introduce a unified dual dynamic smoothing framework that serves as a general template for designing and analyzing single-loop minimax algorithms. A key strength of this framework is its versatility, as it enables us to analyze both NC-C and C-C settings within a rigorous structure. Using this, we develop and analyze three algorithmic variants (LMO-LMO, LMO-PO and PO-LMO) that adapt to the specific oracle types available for the primal and dual variables.

◇ **Improved Rates for Nonconvex Settings.** We establish state-of-the-art convergence rates for NC-C and NC-SC problems, as summarized in Table 1. We highlight the relationships between our proposed methods and existing literature. The LMO-LMO-based Algorithm 1 is structurally identical to R-PDCG (Boroun et al., 2023). However, our convergence analysis adopts different parameter

settings that enable us to obtain *anytime* convergence rates *without* requiring or knowing the strong convexity of $\mathcal{Y}$. Similarly, the LMO-PO-based Algorithm 2 shares the structure of CG-RPGA (Boroun et al., 2023). However, our analysis provides parameter choices that yield anytime guarantees. To the best of our knowledge, The PO-LMO-based Algorithm 3 represents a novel structure in the literature. Notably, our framework provides a unified analysis for all three algorithms through per-iteration bounds, leading to improved complexity results under various combinations of problem assumptions.

◇ **General Convex-Concave Guarantees.** We extend our analysis to the classical C-C and C-SC problems, as summarized in Table 2. Notably, we do not require $\mathcal{L}(\cdot, y)$ to be strongly convex, unlike several existing results marked with *. To the best of our knowledge, we provide the first single-loop projection-free results for the general C-C case that do not require highly restrictive assumptions. Existing LMO-LMO methods, such as SP-FW (Gidel et al., 2017) and STORC (Chen et al., 2020), either fail to provide guarantees for general C-C settings or require the payoff function $\mathcal{L}(\cdot, y)$ to be strongly convex in the primal variable, a condition we do not assume. Furthermore, while one-sided projection-free methods like O-FW (Kolmogorov & Pock, 2021) achieve faster rates, they are limited to bilinear coupling and strongly convex primal sets. In contrast, our framework handles general C-C structures and yields significantly improved rates whenever the feasible set $\mathcal{Y}$ is strongly convex.

◇ **Anytime Convergence Guarantees.** We establish anytime convergence guarantees for all proposed algorithms. This is of significant practical importance as it eliminates the need to pre-specify a target accuracy or a total iteration budget to tune parameters. Our algorithms maintain convergence properties throughout their execution.

*Table 2.* Convergence Guarantees in Convex-Concave Setting.

| TYPES OF ORACLES | | SETTINGS | | REQS. |
|---|---|---|---|---|
| | | C–C | C–SC | |
| LMO – LMO | SP-FW | N/A | $O(\log(\epsilon^{-1}))$ | INT. SOL.* |
| | | N/A | $O(\log(\epsilon^{-1}))$ | $\mathcal{X}, \mathcal{Y}$ POLY.* |
| | | $O(\log(\epsilon^{-1}))$ | N/A | SC $\mathcal{X}, \mathcal{Y}$, GRAD. LB |
| | STORC | N/A | $\widetilde{O}(\epsilon^{-2})$ | |
| | OURS | $O(\epsilon^{-5})$ | $O(\epsilon^{-3})$ | |
| | | $O(\epsilon^{-4})$ | $O(\epsilon^{-2})$ | SC $\mathcal{Y}$ |
| LMO – PO | O-FW | $O(\epsilon^{-1})$ | N/A | SC $\mathcal{X}$, BILINEAR |
| | OURS | $O(\epsilon^{-3})$ | $O(\epsilon^{-1})$ | |
| PO – LMO | OURS | $O(\epsilon^{-3})$ | $\widetilde{O}(\epsilon^{-2})$ | |
| | | $O(\epsilon^{-3})$ | $\widetilde{O}(\epsilon^{-\frac{3}{2}})$ | SC $\mathcal{Y}$ |

### 1.2.1. ASSUMPTIONS AND CONVERGENCE CRITERIA

Throughout the paper, we operate under the following assumptions.

**Assumption 1.1.** The following hold.

(i) The feasible sets $\mathcal{X} \subset \mathbb{R}^{n_x}$ and $\mathcal{Y} \subset \mathbb{R}^{n_y}$ are compact, convex sets with diameters $D_{\mathcal{X}}$ and $D_{\mathcal{Y}}$, respectively.

(ii) For any $x \in \mathcal{X}$, the payoff function $\mathcal{L}(x, \cdot)$ is a $\mu$-strongly-concave function over $\mathcal{Y}$ for some $\mu \geq 0$.

(iii) The payoff function $\mathcal{L}$ is continuously differentiable over an open set containing $\mathcal{X} \times \mathcal{Y}$.

(iv) There exist $L_{xx}, L_{yy} \geq 0, L_{yx} > 0$ such that for any $(x, y), (x', y') \in \mathcal{X} \times \mathcal{Y}$, it holds that

$$\|\nabla_x \mathcal{L}(x, y) - \nabla_x \mathcal{L}(x', y')\| \leq L_{xx} \|x - x'\| + L_{yx} \|y - y'\|,$$
$$\|\nabla_y \mathcal{L}(x, y) - \nabla_y \mathcal{L}(x', y')\| \leq L_{yx} \|x - x'\| + L_{yy} \|y - y'\|.$$

In settings where we employ an LMO update over $y$, we may further assume that the following geometric property.

**Assumption 1.2.** The feasible set $\mathcal{Y}$ is an $\alpha$-strongly convex set for some $\alpha > 0$, that is, for any $y, y' \in \mathcal{Y}, z \in \mathbb{R}^{n_y}$ with $\|z\| = 1$, and $\xi \in [0, 1]$, it holds that

$$\xi y + (1 - \xi) y' + \xi (1 - \xi) \frac{\alpha}{2} \|y - y'\|^2 z \in \mathcal{Y}.$$

To quantify the quality of a candidate solution $(x, y) \in \mathcal{X} \times \mathcal{Y}$ and its proximity to a saddle point, we define several measures of stationarity. We begin by considering the general nonconvex regime, where optimality for the primal and dual variables is characterized by the following metrics. We first introduce a dual gap function to assess stationarity relative to the maximization variable $y$

$$G_{\mathcal{Y}}(x, y) := \max_{u \in \mathcal{Y}} \mathcal{L}(x, u) - \mathcal{L}(x, y). \tag{3}$$

Since $\mathcal{L}(x, \cdot)$ is concave by Assumption 1.1, this measure provides a full characterization of stationarity for the dual variable, satisfying the equivalence

$$\forall x \in \mathcal{X}, \quad y \in \arg\max_{u \in \mathcal{Y}} \mathcal{L}(x, u) \iff G_{\mathcal{Y}}(x, y) = 0.$$

Next, we define measures to evaluate stationarity relative to the minimization variable $x$

$$G_{\mathcal{X}}^{\text{LMO}}(x, y) := \max_{v \in \mathcal{X}} \nabla_x \mathcal{L}(x, y)^\top (x - v), \tag{4a}$$

$$G_{\mathcal{X}}^{\text{PO}}(x, y) := \left\| \frac{1}{\sigma} \left( \text{PO}_{\mathcal{X}} \left( x - \sigma \nabla_x \mathcal{L}(x, y) \right) - x \right) \right\|, \tag{4b}$$

for any fixed parameter $\sigma > 0$. In our framework, we switch between these measures depending on the oracle complexity of the primal variable. Specifically, we use $G_{\mathcal{X}}^{\text{LMO}}$ when $x$

is updated via an LMO, and $G_{\mathcal{X}}^{\text{PO}}$ when a PO is employed. These measures are well-motivated as under Assumption 1.1 the following implication holds

$$\forall y \in \mathcal{Y}, \quad x \in \arg\min_{v \in \mathcal{X}} \mathcal{L}(v, y)$$
$$\implies G_{\mathcal{X}}^{\text{LMO}}(x, y) = G_{\mathcal{X}}^{\text{PO}}(x, y) = 0. \tag{5}$$

The relationship between these two stationarity measures is formally established in the following lemma.

**Lemma 1.3.** *If Assumption 1.1 holds then for any $(x, y) \in \mathcal{X} \times \mathcal{Y}$ and $\sigma > 0$, it holds that*

$$G_{\mathcal{X}}^{\text{LMO}}(x, y) \leq (\sigma \|\nabla_x \mathcal{L}(x, y)\| + D_{\mathcal{X}}) G_{\mathcal{X}}^{\text{PO}}(x, y).$$

In the general nonconvex case, the implication (5) is one-way. However, these implications become equivalences (if and only if) under additional convexity assumption.

**Assumption 1.4.** $\mathcal{L}$ is convex-concave over $\mathcal{X} \times \mathcal{Y}$.

Under Assumption 1.4, a standard metric for assessing convergence is the *saddle-point duality gap*, defined as

$$G_{\mathcal{X},\mathcal{Y}}^{\text{Dual}}(x, y) := \max_{u \in \mathcal{Y}} \mathcal{L}(x, u) - \min_{v \in \mathcal{X}} \mathcal{L}(v, y).$$

By establishing rates for $G_{\mathcal{Y}}$ and either $G_{\mathcal{X}}^{\text{LMO}}$ or $G_{\mathcal{X}}^{\text{PO}}$, we can directly bound the duality gap for the ergodic averages of the iterates.

**Lemma 1.5.** *Suppose $\{x_t, y_t\}_{t=1}^T$ is any sequence in $\mathcal{X} \times \mathcal{Y}$. If Assumptions 1.1 and 1.4 hold, then we have*

$$G_{\mathcal{X},\mathcal{Y}}^{\text{Dual}}(\bar{x}_T, \bar{y}_T) \leq \frac{1}{T} \sum_{t=1}^T \left( G_{\mathcal{X}}^{\text{LMO}}(x_t, y_t) + G_{\mathcal{Y}}(x_t, y_t) \right),$$

*where $\bar{x}_T := \frac{1}{T} \sum_{t=1}^T x_t$ and $\bar{y}_T := \frac{1}{T} \sum_{t=1}^T y_t$ denote the average iterates.*

Thanks to Lemma 1.3, we can directly substitute $G_{\mathcal{X}}^{\text{LMO}}$ with $G_{\mathcal{X}}^{\text{PO}}$ in the above bound. This ensures that equivalent convergence guarantees for $G_{\mathcal{X},\mathcal{Y}}^{\text{Dual}}$ hold for projection-based sequences as well, differing only by a constant factor.

While the duality gap provides a comprehensive characterization of optimality, our dual dynamic smoothing framework is designed with an asymmetric structure to specifically accommodate potential nonconvexity in the primal variable. Therefore, in the convex setting we primarily focus our analysis on the *primal optimality gap*

$$G_{\mathcal{X}}^{\text{Opt}}(x) := f(x) - \min_{v \in \mathcal{X}} f(v), \tag{6}$$

where $f$ is the primal function defined in (2). By prioritizing this measure over the full duality gap, we are able to establish faster convergence rates. Furthermore, we observe that if $(x, y)$ is a solution to (1), then the primal optimality gap naturally vanishes, that is, $G_{\mathcal{X}}^{\text{Opt}}(x) = 0$. Finally, we note that all measures of stationarity and optimality introduced in this section are non-negative for any $(x, y) \in \mathcal{X} \times \mathcal{Y}$.

## 2. A Fully LMO Algorithm

In this section, we introduce `LMO-LMO` Algorithm 1, which solves the minimax problem (1) by relying exclusively on LMO for both variables. For any compact convex set $\mathcal{U}$, the LMO at a given direction $d$ is defined as

$$\text{LMO}_{\mathcal{U}}(d) \in \arg\min_{v \in \mathcal{U}} d^\top v.$$

By using this oracle, our algorithm avoids the need for expensive projection steps, making it particularly suitable for problems over complex constraint sets. While the structural design of the algorithm is intuitive, a key technical distinction lies in the dual variable update, which is performed using the gradient of the regularized objective rather than the original payoff function. Although Algorithm 1 shares structural similarities with the `R-PDCG` algorithm (Boroun et al., 2023), it uniquely admits an *anytime* implementation and does not require knowledge of the strong convexity of $\mathcal{Y}$ for stepsize tuning.

We first present the convergence rates in nonconvex settings.

**Theorem 2.1.** *Let $\{x_t, y_t\}_{t \geq 0}$ be the sequence generated by Algorithm 1 with $\tau_t = \Theta(t^{-a})$ and $\beta_t = \Theta(t^{-b})$. The following hold for any $T \geq 0$.*

(i) *NC-C: If Assumption 1.1 holds and $a = \frac{5}{6}, b = \frac{1}{6}$, then*

$$\frac{1}{T+1} \sum_{t=0}^{T} G_{\mathcal{X}}^{\text{LMO}}(x_t, y_t) \leq O(T^{-1/6}),$$
$$G_{\mathcal{Y}}(x_T, y_T) \leq O(T^{-1/6}).$$

(ii) *NC-SC: If Assumption 1.1 holds with $\mu > 0$ and $a = \frac{3}{4}$, $\beta_t = 0$, then*

$$\frac{1}{T+1} \sum_{t=0}^{T} G_{\mathcal{X}}^{\text{LMO}}(x_t, y_t) \leq O(T^{-1/4}),$$
$$G_{\mathcal{Y}}(x_T, y_T) \leq O(T^{-1/4}).$$

(iii) *NC-C + SC $\mathcal{Y}$: If Assumptions 1.1 and 1.2 hold and $a = \frac{4}{5}, b = \frac{1}{5}$, then*

$$\frac{1}{T+1} \sum_{t=0}^{T} G_{\mathcal{X}}^{\text{LMO}}(x_t, y_t) \leq O(T^{-1/5})$$
$$G_{\mathcal{Y}}(x_T, y_T) \leq O(T^{-1/5}).$$

(iv) *NC-SC + SC $\mathcal{Y}$: If Assumptions 1.1 and 1.2 hold with $\mu > 0$ and $a = \frac{2}{3}, \beta_t = 0$, then*

$$\frac{1}{T+1} \sum_{t=0}^{T} G_{\mathcal{X}}^{\text{LMO}}(x_t, y_t) \leq O(T^{-1/3})$$
$$G_{\mathcal{Y}}(x_T, y_T) \leq O(T^{-1/3}).$$

---

**Algorithm 1** `LMO-LMO` Algorithm

---

**Initialize:** $x_0 \in \mathcal{X}, y_0 \in \mathcal{Y}$
**for** $t = 0, 1, \ldots, T - 1$ **do**
    Compute $v_t \leftarrow \text{LMO}_{\mathcal{X}}(\nabla_x \mathcal{L}(x_t, y_t))$
    Update $x_{t+1} \leftarrow x_t + \tau_t(v_t - x_t)$
    Compute $u_t \leftarrow \text{LMO}_{\mathcal{Y}}(-\nabla_y \mathcal{L}_{\beta_t}(x_t, y_t))$
    Compute $\gamma_t \leftarrow \min\left\{\frac{\nabla_y \mathcal{L}_{\beta_t}(x_t, y_t)^\top (u_t - y_t)}{(L_{yy} + \beta_t)\|u_t - y_t\|_2^2}, 1\right\}$
    Update $y_{t+1} \leftarrow y_t + \gamma_t(u_t - y_t)$
**end for**

---

The complexity results established in Theorem 2.1 represent a significant improvement over the current state-of-the-art bounds presented in (Boroun et al., 2023); see Table 1 for a detailed comparison. We note that Boroun et al. (2023) rely on the LMO gap for both $\mathcal{X}$ and $\mathcal{Y}$ and report convergence in terms of the sum of these two quantities. In contrast, our framework uses the LMO gap for $\mathcal{X}$ and introduces a finer dual optimality gap $G_{\mathcal{Y}}$ for the dual variable, which satisfies $G_{\mathcal{Y}} \leq G_{\mathcal{Y}}^{\text{LMO}}$. Notably, establishing rates for this dual measure enables a direct translation to the standard duality gap $G_{\mathcal{X}, \mathcal{Y}}^{\text{Dual}}$ through Lemma 1.5. However, given that our dual dynamic smoothing framework is asymmetric to accommodate potential nonconvexity in the primal variable, we focus our primary convex analysis on the primal optimality gap. As shown below, this perspective allows us to establish faster convergence rates in the convex setting.

**Theorem 2.2.** *Under Assumption 1.4, Let $\{x_t, y_t\}_{t \geq 0}$ be the sequence generated by Algorithm 1 with $\tau_t = \Theta(t^{-a})$ and $\beta_t = \Theta(t^{-b})$.*

(i) *C-C: If Assumption 1.1 also holds and $a = 1, b = \frac{1}{5}$, then for any $T \geq 1$,*

$$G_{\mathcal{X}}^{\text{Opt}}(x_T) \leq O(T^{-1/5}), \ G_{\mathcal{Y}}(x_T, y_T) \leq O(T^{-1/5}).$$

(ii) *C-SC: If Assumption 1.1 with $\mu > 0$ also holds and $a = 1, \beta_t = 0$, then for any $T \geq 1$,*

$$G_{\mathcal{X}}^{\text{Opt}}(x_T) \leq \widetilde{O}(T^{-1/3}), \ G_{\mathcal{Y}}(x_T, y_T) \leq O(T^{-2/3}).$$

(iii) *C-C + SC $\mathcal{Y}$: If Assumptions 1.1 and 1.2 also hold and $a = 1, b = \frac{1}{4}$, then for any $T \geq 1$,*

$$G_{\mathcal{X}}^{\text{Opt}}(x_T) \leq O(T^{-1/4}), \ G_{\mathcal{Y}}(x_T, y_T) \leq O(T^{-1/4}).$$

(iv) *C-SC + SC $\mathcal{Y}$: If Assumptions 1.1 and 1.2 with $\mu > 0$ also hold and $a = 1, \beta_t = 0$, then for any $T \geq 1$,*

$$G_{\mathcal{X}}^{\text{Opt}}(x_T) \leq \widetilde{O}(T^{-1/2}), \ G_{\mathcal{Y}}(x_T, y_T) \leq O(T^{-1}).$$

The results established in Theorem 2.2 demonstrate that our framework effectively eliminates the need for the specialized geometric assumptions or restrictive regularity conditions used by Gidel et al. (2017) for single-loop projection-free algorithms. A comprehensive comparison of our results against these existing benchmarks is provided in Table 2.

## 3. A Hybrid LMO-PO Algorithm

In settings where the dual domain $\mathcal{Y}$ admits an efficient projection oracle, we propose `LMO-PO` Algorithm 2. For any compact convex set $\mathcal{U}$, the PO at a point $u$ is defined as

$$\text{PO}_{\mathcal{U}}(u) := \underset{v \in \mathcal{U}}{\arg\min} \|u - v\|^2.$$

Algorithm 2 pairs LMO updates with dual projections, achieving faster rate than the fully LMO setting. While structurally similar to `CG-RPGA` (Boroun et al., 2023, Algorithm 2), our analysis employs *anytime* parameter schedules.

We first present the convergence rates in nonconvex settings.

**Theorem 3.1.** *Let $\{x_t, y_t\}_{t \geq 0}$ be the sequence generated by Algorithm 2 with $\tau_t = \Theta(t^{-a}), \beta_t = \Theta(t^{-b})$. The following hold for any $T \geq 1$.*

*(i) **NC-C:** If Assumption 1.1 holds and $a = \frac{3}{4}, b = \frac{1}{4}$, then*

$$\frac{1}{T} \sum_{t=1}^{T} G_{\mathcal{X}}^{\text{LMO}}(x_t, y_t) \leq O(T^{-1/4}),$$
$$\frac{1}{T} \sum_{t=1}^{T} G_{\mathcal{Y}}(x_t, y_t) \leq O(T^{-1/4}).$$

*(ii) **NC-SC:** If Assumption 1.1 holds with $\mu > 0$ and $a = \frac{1}{2}$, $\beta_t = 0$, then for any $T \geq 0$,*

$$\frac{1}{T} \sum_{t=1}^{T} G_{\mathcal{X}}^{\text{LMO}}(x_t, y_t) \leq O(T^{-1/2}),$$
$$\frac{1}{T} \sum_{t=1}^{T} G_{\mathcal{Y}}(x_t, y_t) \leq \widetilde{O}(T^{-1}).$$

By Lemma 1.5, Theorem 3.1 rates automatically extend to the duality gap $G_{\mathcal{X},\mathcal{Y}}^{\text{Dual}}$. In convex settings, we again use the primal optimality gap to exploit framework asymmetry.

**Theorem 3.2.** *Under Assumption 1.4, let $\{x_t, y_t\}_{t \geq 0}$ be the sequence generated by Algorithm 2 under $\tau_t = \Theta(t^{-a})$ and $\beta_t = \Theta(t^{-b})$. The following hold for any $T \geq 1$.*

*(i) **C-C:** If Assumption 1.1 also holds and $a = 1, b = \frac{1}{3}$, then*

$$G_{\mathcal{X}}^{\text{Opt}}(x_T) \leq \widetilde{O}(T^{-1/3}), \ \frac{1}{T} \sum_{t=1}^{T} G_{\mathcal{Y}}(x_t, y_t) \leq O(T^{-1/3}).$$

*(ii) **C-SC:** If Assumption 1.1 also holds with $\mu > 0$ hold and $a = 1, \beta_t = 0$, then*

$$G_{\mathcal{X}}^{\text{Opt}}(x_T) \leq \widetilde{O}(T^{-1}), \ \frac{1}{T} \sum_{t=1}^{T} G_{\mathcal{Y}}(x_t, y_t) \leq O(T^{-1}).$$

---

**Algorithm 2** `LMO-PO` Algorithm

**Initialize:** $x_0 \in \mathcal{X}, y_0 \in \mathcal{Y}$
**for** $t = 0, 1, \ldots, T - 1$ **do**
    Compute $v_t \in \text{LMO}_{\mathcal{X}}(\nabla_x \mathcal{L}(x_t, y_t))$
    Update $x_{t+1} \leftarrow x_t + \tau_t(v_t - x_t)$
    Compute $\gamma_t \leftarrow \frac{1}{L_{yy} + \beta_t}$
    Update $y_{t+1} \leftarrow \text{PO}_{\mathcal{Y}}(y_t + \gamma_t \nabla_y \mathcal{L}_{\beta_t}(x_t, y_t))$
**end for**

---

**Algorithm 3** `PO-LMO` Algorithm

**Initialize:** $x_0 \in \mathcal{X}, y_0 \in \mathcal{Y}$
**for** $t = 0, 1, \ldots, T - 1$ **do**
    Update $x_{t+1} \leftarrow \text{PO}_{\mathcal{X}}(x_t - \tau_t \nabla_x \mathcal{L}(x_t, y_t))$
    Compute $u_t \in \text{LMO}_{\mathcal{Y}}(-\nabla_y \mathcal{L}_{\beta_t}(x_t, y_t))$
    Compute $\gamma_t \leftarrow \min\left\{\frac{\nabla_y \mathcal{L}_{\beta_t}(x_t, y_t)^\top (u_t - y_t)}{(L_{yy} + \beta_t)\|u_t - y_t\|_2^2}, 1\right\}$
    Update $y_{t+1} \leftarrow y_t + \gamma_t(u_t - y_t)$
**end for**

---

## 4. A Hybrid PO-LMO Algorithm

We propose `PO-LMO` Algorithm 3 for scenarios where the primal domain $\mathcal{X}$ admits an efficient projection oracle while the dual domain $\mathcal{Y}$ is best handled via linear minimization. To the best of our knowledge, this specific hybrid configuration has not been explored in the existing literature. As with our previous methods, we employ the dual dynamic smoothing framework with anytime parameter schedules, allowing the algorithm to converge without needing to set a fixed horizon.

We first present the convergence rates in nonconvex settings.

**Theorem 4.1.** *Let $\{x_t, y_t\}_{t \geq 0}$ be the sequence generated by Algorithm 3 with $\tau_t = \Theta(t^{-a}), \beta_t = \Theta(t^{-b})$ and $\sigma = \tau_0$. The following hold for any $T \geq 0$.*

*(i) **NC-C:** If Assumption 1.1 holds and $a = \frac{2}{3}, b = \frac{1}{6}$, then*

$$\frac{1}{T+1} \sum_{t=0}^{T} G_{\mathcal{X}}^{\text{PO}}(x_t, y_t) \leq \widetilde{O}(T^{-1/6}),$$
$$\frac{1}{T+1} \sum_{t=0}^{T} G_{\mathcal{Y}}(x_t, y_t) \leq \widetilde{O}(T^{-1/6}).$$

*(ii) **NC-SC:** If Assumption 1.1 holds with $\mu > 0$ and $a = \frac{1}{2}$, $\beta_t = 0$, then*

$$\frac{1}{T+1} \sum_{t=0}^{T} G_{\mathcal{X}}^{\text{PO}}(x_t, y_t) \leq O(T^{-1/4}),$$
$$\frac{1}{T+1} \sum_{t=0}^{T} G_{\mathcal{Y}}(x_t, y_t) \leq \widetilde{O}(T^{-1/2}).$$

*(iii)* **NC-C + SC $\mathcal{Y}$:** *If Assumptions 1.1 and 1.2 hold and $a = \frac{3}{5}, b = \frac{1}{5}$, then*

$$\frac{1}{T+1}\sum_{t=0}^{T} G_{\mathcal{X}}^{\mathrm{PO}}(x_t, y_t) \le O(T^{-1/5}),$$

$$\frac{1}{T+1}\sum_{t=0}^{T} G_{\mathcal{Y}}(x_t, y_t) \le O(T^{-1/5}).$$

*(iv)* **NC-SC + SC $\mathcal{Y}$:** *If Assumptions 1.1 and 1.2 hold with $\mu > 0$ and $a = \frac{1}{3}, \beta_t = 0$, then*

$$\frac{1}{T+1}\sum_{t=0}^{T} G_{\mathcal{X}}^{\mathrm{PO}}(x_t, y_t) \le O(T^{-1/3}),$$

$$\frac{1}{T+1}\sum_{t=0}^{T} G_{\mathcal{Y}}(x_t, y_t) \le \widetilde{O}(T^{-2/3}).$$

Similar to the previous hybrid setting, the rates established in Theorem 4.1 can be directly translated to the saddle-point duality gap $G_{\mathcal{X},\mathcal{Y}}^{\mathrm{Dual}}$ by applying Lemmas 1.3 and 1.5. We now extend this analysis to the convex setting, focusing on the primal optimality gap.

**Theorem 4.2.** *Under Assumption 1.4, let $\{x_t, y_t\}_{t \ge 0}$ be the sequence generated by Algorithm 3 with $\tau_t = \Theta(t^{-a})$, $\beta_t = \Theta(t^{-b})$ and $\sigma = \tau_0$. The following hold for any $T \ge 0$.*

*(i)* **C-C:** *If Assumption 1.1 holds and $a = \frac{2}{3}, b = \frac{1}{3}$, then*

$$\frac{1}{T+1}\sum_{t=0}^{T} G_{\mathcal{X}}^{\mathrm{Opt}}(x_t) \le O(T^{-1/3}),$$

$$\frac{1}{T+1}\sum_{t=0}^{T} G_{\mathcal{Y}}(x_t, y_t) \le O(T^{-1/3}).$$

*(ii)* **C-SC:** *If Assumption 1.1 with $\mu > 0$ holds and $a = \frac{1}{2}, \beta_t = 0$, then for any $T \ge 0$,*

$$\frac{1}{T+1}\sum_{t=0}^{T} G_{\mathcal{X}}^{\mathrm{Opt}}(x_t) \le \widetilde{O}(T^{-1/2}),$$

$$\frac{1}{T+1}\sum_{t=0}^{T} G_{\mathcal{Y}}(x_t, y_t) \le \widetilde{O}(T^{-1/2}).$$

*(iii)* **C-SC + SC $\mathcal{Y}$:** *If Assumptions 1.1 and 1.2 hold with $\mu > 0$ and $a = \frac{1}{3}, \beta_t = 0$, then*

$$\frac{1}{T+1}\sum_{t=0}^{T} G_{\mathcal{X}}^{\mathrm{Opt}}(x_t) \le \widetilde{O}(T^{-2/3}),$$

$$\frac{1}{T+1}\sum_{t=0}^{T} G_{\mathcal{Y}}(x_t, y_t) \le \widetilde{O}(T^{-2/3}).$$

We remark that in the general convex-concave setting, the additional Assumption 1.1(ii) of strong convexity on $\mathcal{Y}$ does not yield a convergence rate faster than $\mathcal{O}(T^{-1/3})$ established in Theorem 4.2 (i). Therefore, we have omitted this result from the theorem statement.

## 5. Numerical Experiments

We demonstrate the performance of our proposed algorithms on dictionary learning and robust multiclass classification problems. The former is an instance of a nonconvex–concave setting, while the latter is an instance of a convex–concave setting. For performance comparison, we implement four algorithms: `SP-FW` (Gidel et al., 2017), `AGP` (Xu et al., 2023), `R-PDCG` (Boroun et al., 2023, Algorithm 1) and `CG-RPGA` (Boroun et al., 2023, Algorithm 2). In the following, we use $\|\cdot\|$, $\|\cdot\|_{\mathrm{F}}$ and $\|\cdot\|_*$ to denote the Euclidean norm, the Frobenius norm, and the nuclear norm, respectively. All implementation details follow those in the corresponding papers, described in Appendix G.

**Dictionary Learning** The goal is to identify a parsimonious basis or a dictionary $\mathbf{D} = [d_1, \ldots, d_p] \in \mathbb{R}^{m \times p}$, that can effectively reconstruct a high-dimensional dataset $\mathbf{A} \in \mathbb{R}^{m \times n}$ through linear combinations. Formally, this is often expressed as the following nonconvex problem

$$\min_{\mathbf{C},\mathbf{D}} \left\{ \frac{1}{2n}\|\mathbf{A} - \mathbf{DC}\|_{\mathrm{F}}^2 : \begin{array}{l} \|\mathbf{C}\|_* \le r, \\ \|d_j\|_2 \le 1 \ \forall j \in [p] \end{array} \right\}, \quad (7)$$

where $\mathbf{C} \in \mathbb{R}^{p \times n}$ denotes the coefficient matrix.

In the context of continual or lifelong learning, a model must incorporate new information without discarding previously acquired knowledge. Suppose we have already obtained an optimal dictionary $\mathbf{D}$ and coefficients $\mathbf{C}$ for an initial dataset $\mathbf{A}$. When presented with a new dataset $\mathbf{A}' \in \mathbb{R}^{m \times n'}$, we seek a refined dictionary $\mathbf{D}' \in \mathbb{R}^{m \times q}$ and new coefficients $\mathbf{C}' \in \mathbb{R}^{q \times n'}$ that accurately model the new data while maintaining a user-defined reconstruction fidelity $\delta$ on the original data $\mathbf{A}$. This task leads to the nonconvex problem

$$\min_{\mathbf{C}',\mathbf{D}'} \left\{ \frac{1}{2n'}\|\mathbf{A}' - \mathbf{D}'\mathbf{C}'\|_{\mathrm{F}}^2 : \begin{array}{l} \|\mathbf{A}' - \mathbf{D}'\tilde{\mathbf{C}}\|_{\mathrm{F}}^2 \le 2n\delta, \\ \|\mathbf{C}'\|_* \le r, \\ \|d'_j\|_2 \le 1 \ \forall j \in [q] \end{array} \right\} \quad (8)$$

where $\tilde{\mathbf{C}}$ is the previous coefficient matrix $\mathbf{C}$ zero-padded to match the new dictionary size $q$. By employing Lagrangian duality, we can transform this constrained task into a smooth nonconvex-concave minimax problem with payoff function

$$\mathcal{L}((\mathbf{D}', \mathbf{C}'), y) = \frac{1}{2n'}\|\mathbf{A}' - \mathbf{D}'\mathbf{C}'\|_{\mathrm{F}}^2$$
$$+ y\left(\frac{1}{2n}\|\mathbf{A}' - \mathbf{D}'\tilde{\mathbf{C}}\|_{\mathrm{F}}^2 - \delta\right)$$

and the feasible sets $\mathcal{X} := \{(\mathbf{D}', \mathbf{C}') \in \mathbb{R}^{m \times q} \times \mathbb{R}^{q \times n'} \mid \|\mathbf{C}\|_* \le r, \ \|d_j\|_2 \le 1, \ \forall j \in [p]\}$ and $\mathcal{Y} := [0, \infty)$. Given that the problem satisfies the Slater condition, one can establish an upper bound $B > 0$ such that the optimal dual variable $y^\star$ lies within $[0, B]$ (Palomar & Eldar, 2010, Lemma 1.1). Restricting $\mathcal{Y} = [0, B]$ ensures that both feasible sets are compact, satisfying the core assumptions of our NC-C regime of our theoretical analysis.

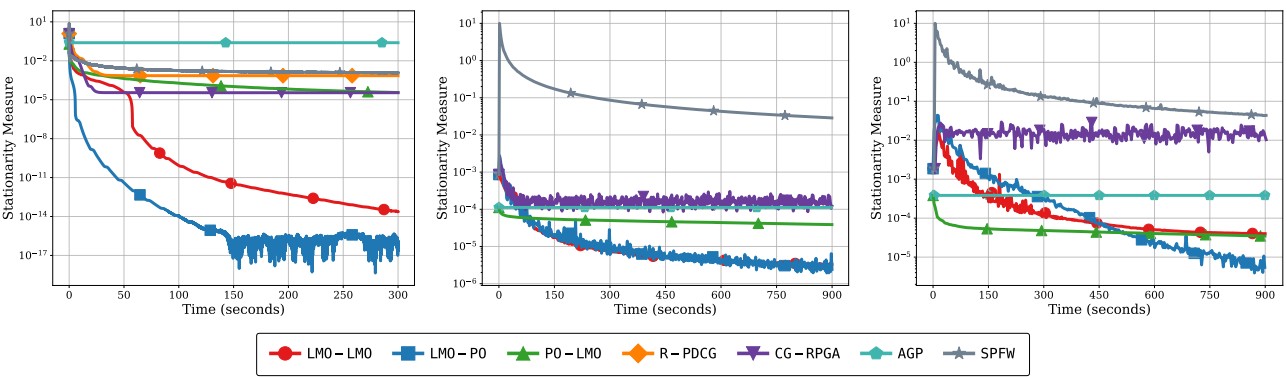

*Figure 1.* Stationarity measure over time across datasets: `DL`, `RCV1` and `NEWS20` (left to right).

**Robust Multiclass Classification** In a typical multiclass setting, we are given a training dataset $\mathcal{D}^{\text{tr}} = \{(a_i, b_i)\}_{i=1}^n$, where $a_i \in \mathbb{R}^d$ represents the feature vector and $b_i \in \{1, \ldots, k\}$ denotes the class label. The objective is to learn a predictor matrix $\Theta = [\theta_1, \ldots, \theta_k]^\top \in \mathbb{R}^{k \times d}$ such that for a new instance $\hat{a}$, the predicted label is given by $\hat{b} = \arg\max_{j \in [k]} \theta_j^\top \hat{a}$. A standard approach is minimizing the multinomial logistic loss, defined for each sample $(a_i, b_i)$ as $\ell_i(\Theta) = \log(1 + \sum_{j \neq b_i} \exp(\theta_j^\top a_i - \theta_{b_i}^\top a_i))$. To promote a low-rank structure in the predictor, it is common to enforce a nuclear norm constraint, such that $\mathcal{X} = \{\Theta \in \mathbb{R}^{k \times d} \mid \|\Theta\|_* \leq r\}$.

To ensure reliability under data uncertainty or potential distribution shifts, we employ the framework of distributionally robust optimization (DRO) (Kuhn et al., 2025). Specifically, rather than minimizing standard empirical risk, we adopt a *penalized* DRO formulation, following (Gotoh et al., 2018), which leads to a smooth convex-concave minimax problem of the form

$$\min_{\Theta \in \mathcal{X}} \max_{y \in \mathcal{Y}} \quad \frac{1}{n} \sum_{i=1}^n y_i \ell_i(\Theta) - \lambda D_\phi(y, \tfrac{1}{n} 1_n),$$

where $y$ is a probability weight vector, $\mathcal{Y} \subseteq \mathbb{R}^n$ is the $n$-dimensional unit simplex, and $D_\phi$ denotes a $\phi$-divergence measuring the discrepancy between distributions. Throughout this section, we utilize the Pearson $\chi^2$-divergence, defined as $D_\phi(y, \tfrac{1}{n} 1_n) \coloneqq \|ny - 1_n\|_2^2$. This choice is motivated by its close connection to variance regularization (Gotoh et al., 2018) and the fact that the resulting inner maximization problem is strongly concave, effectively mapping the problem to the C-SC regime of our theoretical analysis.

**Datasets and Summary of Results** For the dictionary learning problem, we generate a synthetic dataset, denoted by `DL`, according to the procedure described in (Boroun et al., 2023, Appendix F). In the case of robust multiclass classification, we evaluate our algorithms on the `RCV1` and `NEWS20` datasets obtained from the LIBSVM repository

(Chang & Lin, 2011). Details of our implementation and parameter choices are provided in Appendix G.

In Figure 1, we plot the stationarity gap, defined as $G_\mathcal{X}^{\text{LMO}} + G_\mathcal{Y}$ or $G_\mathcal{X}^{\text{PO}} + G_\mathcal{Y}$ depending on the specific primal oracle utilized, across the tested datasets. Our proposed algorithms demonstrate a faster and sustained convergence behavior compared to existing benchmarks. Notably, `LMO-PO` emerges as the most effective configuration two out three experiments, suggesting that when dual projections are computationally feasible, they provide significant numerical speedup and stability. Furthermore, although the fully projection-free `LMO-LMO` variant is initially less aggressive than the hybrid `LMO-PO`, it significantly outperforms existing approaches such as `R-PDCG` and `CG-RPGA`, both of which tend to plateau prematurely. We note that `R-PDCG` is not applicable since the dual feasible set is not strongly convex, which is a required assumption in (Boroun et al., 2023). Moreover, we observe that using the PO oracle to update the primal variable leads to consistently slow convergence and only marginal improvement over time in both the `PO-LMO` and `AGP` algorithms. We also report the number of iterations completed by each algorithm in Table 3. We observe that using the LMO oracle consistently allows more iterations to be completed than the PO oracle. In particular, in the dictionary learning experiment, the number of iterations is almost doubled when the LMO oracle is used to update the primal variable.

*Table 3.* Number of iterations completed within the time limit.

| Method | DL | RCV1 | NEWS20 |
|---|---|---|---|
| SP-FW | 48466 | 467 | 315 |
| AGP | 20997 | 387 | 302 |
| R-PDCG | 46688 | N/A | N/A |
| CG-RPGA | 47036 | 464 | 317 |
| LMO-LMO | 42016 | 453 | 313 |
| LMO-PO | 46678 | 467 | 319 |
| PO-LMO | 21991 | 386 | 305 |

## Acknowledgements

This work was supported by the NSF CAREER ECCS-2541066. We thank reviewers for their valuable feedback.

## Impact Statement

This paper presents work whose goal is to advance the field of Machine Learning. There are many potential societal consequences of our work, none which we feel must be specifically highlighted here.

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

## Appendix Roadmap and Summary of Results

This appendix provides the complete technical proofs, detailed parameter derivations, and auxiliary technical lemmas for the algorithms introduced in the main text. Throughout the appendix, $\| \cdot \|$ denotes the standard Euclidean norm. The content is structured as follows.

- ◇ Appendix A introduces the formal properties of the quadratic regularization used in our framework and establishes the fundamental one-step recursion bounds for a discrepancy term $H_t$ that are used throughout our analysis.

- ◇ Appendix B contains the proofs for the relationships between stationarity measures, presented in Section 1.

- ◇ Appendix C devotes the complete convergence proofs for the fully LMO-based Algorithm 1, presented in Section 2.

- ◇ Appendix D presents the technical analysis for the hybrid LMO-PO Algorithm 2, presented in Section 3.

- ◇ Appendix E details the convergence guarantees for the hybrid PO-LMO Algorithm 3, presented in Section 4.

- ◇ Appendix F includes standard technical lemmas, such as various AM-GM applications and summation bounds, required to finalize the convergence rates.

Table 4 summarizes the convergence guarantees for Algorithms 1–3 across all settings discussed in this work. While the main body provides high-level theorems in big-$O$ notation, this appendix presents the full technical proofs with explicit convergence constants and parameter settings. Due to the high level of detail required for these derivations, we have partitioned the primary theorems from the main text into several sub-theorems.

*Table 4.* Summary of convergence guarantees for Algorithms 1 to 3 given in Sections 2 to 4. Assumption 1.1 holds for all results.

|  | Main Text | Primal-Dual Rate | Asm 1.4 | Asm 1.2 | $\mu > 0$ | Appendix |
|---|---|---|---|---|---|---|
| **Algorithm 1** | Thm 2.1 (i) | $O(T^{-1/6})$ | - | - | - | Thm C.5 |
|  | Thm 2.1 (ii) | $O(T^{-1/4})$ | - | - | ✓ |  |
|  | Thm 2.1 (iii) | $O(T^{-1/5})$ | - | ✓ | - | Thm C.6 |
|  | Thm 2.1 (iv) | $O(T^{-1/3})$ | - | ✓ | ✓ |  |
|  | Thm 2.2 (i) | $O(T^{-1/5})$ | ✓ | - | - | Thm C.8 |
|  | Thm 2.2 (iv) | $\widetilde{O}(T^{-1/3})$ | ✓ | - | ✓ |  |
|  | Thm 2.2 (iii) | $O(T^{-1/4})$ | ✓ | ✓ | - | Thm C.9 |
|  | Thm 2.2 (iv) | $\widetilde{O}(T^{-1/2})$ | ✓ | ✓ | ✓ |  |
| **Algorithm 2** | Thm 3.1 (i) | $O(T^{-1/4})$ | - | - | - | Thm D.5 |
|  | Thm 3.1 (ii) | $O(T^{-1/2})$ | - | - | ✓ |  |
|  | Thm 3.2 (i) | $\widetilde{O}(T^{-1/3})$ | ✓ | - | - | Thm D.7 |
|  | Thm 3.2 (ii) | $O(T^{-1})$ | ✓ | - | ✓ | Thm D.9 |
| **Algorithm 3** | Thm 4.1 (i) | $\widetilde{O}(T^{-1/6})$ | - | - | - | Thm E.4 |
|  | Thm 4.1 (ii) | $O(T^{-1/4})$ | - | - | ✓ |  |
|  | Thm 4.1 (iii) | $O(T^{-1/5})$ | - | ✓ | - | Thm E.9 |
|  | Thm 4.1 (iv) | $O(T^{-1/3})$ | - | ✓ | ✓ |  |
|  | Thm 4.2 (i) | $O(T^{-1/3})$ | ✓ | - | - | Thm E.5 |
|  | Thm 4.2 (ii) | $\widetilde{O}(T^{-1/2})$ | ✓ | - | ✓ |  |
|  | Thm 4.2 (iii) | $\widetilde{O}(T^{-2/3})$ | ✓ | ✓ | ✓ | Thm E.10 |

# A. Foundations of Dual Dynamic Smoothing and One-Step Progress

A fundamental difficulty in minimax optimization is the potential nonsmoothness of the primal objective function

$$f(x) := \max_{y \in \mathcal{Y}} \mathcal{L}(x, y),$$

which can occur even when the payoff $\mathcal{L}$ is smooth. To address this challenge, we employ a quadratic regularization approach to approximate $f$ with a smooth surrogate. This smoothing technique was pioneered in standard optimization by Nesterov (2005) and Beck & Teboulle (2012), and was recently extended to projection-free minimax settings by Boroun et al. (2023). In this section, we first introduce the formal *dual-smoothing framework* and establish its key regularity properties. We then analyze the *one-step progress* of the iterates under various technical conditions, considering update rules based on both LMO and PO for the primal and dual variables. This analysis provides the necessary recursions to establish the convergence rates of our single-loop algorithms.

## A.1. The Dual Smoothing Framework

Given a payoff function $\mathcal{L} : \mathcal{X} \times \mathcal{Y} \to \mathbb{R}$, a reference point $y_0 \in \mathcal{Y}$, and a regularization parameter $\beta \geq 0$, we define the regularized payoff function $\mathcal{L}_\beta$ and the resulting regularized primal function $f_\beta$ as follows

$$\mathcal{L}_\beta(x, y) := \mathcal{L}(x, y) - \frac{\beta}{2} \|y - y_0\|^2 \tag{9a}$$

$$f_\beta(x) := \max_{y \in \mathcal{Y}} \mathcal{L}_\beta(x, y). \tag{9b}$$

For any $\beta > 0$, the function $\mathcal{L}_\beta(x, \cdot)$ is strongly concave in $y$, ensuring that for any $x \in \mathcal{X}$, there exists a unique maximizer

$$y_\beta^\star(x) := \arg\max_{y \in \mathcal{Y}} \mathcal{L}_\beta(x, y). \tag{9c}$$

The following lemma establishes the key regularity properties of this construction, providing the analytical foundation for our smoothing framework.

**Lemma A.1.** *Suppose Assumption 1.1 holds. For any $x, x' \in \mathcal{X}$, $y \in \mathcal{Y}$ and $\beta \geq \beta' \geq 0$ with $\max\{\beta, \mu\} > 0$, we have*

(i) $\frac{\beta+\mu}{2} \|y_\beta^\star(x) - y\|^2 \leq \mathcal{L}_\beta(x, y_\beta^\star(x)) - \mathcal{L}_\beta(x, y).$

(ii) $\mathcal{L}_\beta(x, y_\beta^\star(x)) - \mathcal{L}_\beta(x, y) \leq \frac{2}{\beta+\mu} \|\nabla_y \mathcal{L}_\beta(x, y)\|^2.$

(iii) $\|y_\beta^\star(x) - y_\beta^\star(x')\| \leq \frac{L_{yx}}{\beta+\mu} \|x - x'\|.$

(iv) $f_\beta$ *is differentiable with* $\nabla f_\beta(x) = \nabla_x \mathcal{L}_\beta(x, y_\beta^\star(x)) = \nabla_x \mathcal{L}(x, y_\beta^\star(x))$ *on an open set containing* $\mathcal{X}$.

(v) $\nabla f_\beta$ *is Lipschitz continuous over $X$ with the corresponding constant* $L_{f_\beta} := L_{xx} + L_{yx}^2/(\beta + \mu)$.

(vi) $0 \leq f_{\beta'}(x) - f_\beta(x) \leq \frac{D_\mathcal{Y}^2(\beta - \beta')}{2}.$

*Proof.* Assertion (i) is a direct consequence of the $(\beta + \mu)$-strong concavity of $\mathcal{L}_\beta(x, \cdot)$ with respect to $y$, which follows immediately from its definition. To prove assertion (ii), we use the concavity of $\mathcal{L}_\beta(x, \cdot)$. For any $y \in \mathcal{Y}$,

$$\begin{aligned}
\mathcal{L}_\beta(x, y_\beta^\star(x)) - \mathcal{L}_\beta(x, y) &\leq \nabla_y \mathcal{L}_\beta(x, y)^\top (y_\beta^\star(x) - y) \\
&\leq \|y_\beta^\star(x) - y\| \cdot \|\nabla_y \mathcal{L}_\beta(x, y)\| \\
&\leq \sqrt{\frac{2}{\beta + \mu} \left( \mathcal{L}_\beta(x, y_\beta^\star(x)) - \mathcal{L}_\beta(x, y) \right)} \|\nabla_y \mathcal{L}_\beta(x, y)\|,
\end{aligned}$$

where the first inequality follows from the concavity of $\mathcal{L}_\beta(x, \cdot)$, the second inequality follows from Cauchy-Schwartz inequality, and the last inequality follows from Lemma A.1 (i). Squaring both sides of the resulting inequality yields the desired claim. Assertion (iii) follows directly from the results established in Lin et al. (2020b, Lemma 4.3). Assertion (iv) follows from Danskin's theorem (Bertsekas, 1999, Proposition B.25(a)), which is applicable because the condition $\max\{\beta, \mu\} > 0$

guarantees a unique solution $y_\beta^\star(x)$ to the maximization problem in (9b). Furthermore, Danskin's theorem further implies that $\nabla_x \mathcal{L}_\beta(x, y_\beta^\star(x)) = \nabla_x \mathcal{L}(x, y_\beta^\star(x))$. Assertion (v) follows directly from the analysis provided in Lin et al. (2020b, Lemma 4.3). Finally, to prove assertion (vi), the condition $\beta \geq \beta'$ directly implies that for any $x \in \mathcal{X}$ and $y \in \mathcal{Y}$, we have

$$\mathcal{L}(x, y) - \frac{\beta'}{2} \|y - y_0\|^2 \geq \mathcal{L}(x, y) - \frac{\beta}{2} \|y - y_0\|^2.$$

By maximizing both sides over $y \in \mathcal{Y}$ and using the definition of the regularized primal function, we obtain $f_\beta(x) \leq f_{\beta'}(x)$, which establishes the lower bound. For the upper bound, note that

$$\mathcal{L}(x, y) - \frac{\beta'}{2} \|y - y_0\|^2 = \mathcal{L}(x, y) - \frac{\beta}{2} \|y - y_0\|^2 + \frac{\beta - \beta'}{2} \|y - y_0\|^2 \leq \mathcal{L}(x, y) - \frac{\beta}{2} \|y - y_0\|^2 + \frac{D_\mathcal{Y}^2(\beta - \beta')}{2}.$$

Maximizing both sides over $y \in \mathcal{Y}$ and using the definition of the regularized primal function yield the desired upper bound. This completes the proof. $\square$

Since we dynamically update the smoothing parameter $\beta$ at each iteration $t$, we define the following shorthands to simplify our subsequent analysis

$$\mathcal{L}_t(x, y) := \mathcal{L}_{\beta_t}(x, y), \quad f_t(x) := f_{\beta_t}(x), \quad y_t^\star(x) := y_{\beta_t}^\star(x). \tag{10}$$

### A.2. One-Step Progress for Single-Loop Algorithms

Standard gradient methods applied to $f_\beta$ require the *exact* maximizer $y_\beta^\star(x)$ to compute $\nabla f_\beta(x)$. However, this requires solving an inner optimization problem at every iteration, resulting in a computationally expensive double-loop structure. Furthermore, the smoothing parameter $\beta$ presents additional challenges. A fixed choice of $\beta$ introduces a persistent approximation error, whereas dynamically decreasing $\beta$ toward zero causes the smoothness constant of $f_\beta$ to explode. To achieve a more efficient single-loop architecture, our framework circumvents the inner maximization entirely. We instead perform a single oracle query, using either a linear minimization oracle or a projection oracle, directly on the regularized payoff function $\mathcal{L}_\beta$. To characterize the progress of these updates, we define a discrepancy term $H_t$ that approximately quantifies the distance of the current dual iterate from the regularized optimum. Using the shorthand (10), we define the discrepancy at iteration $t$ as follows

$$H_t := \mathcal{L}_t(x_t, y_t^\star(x_t)) - \mathcal{L}_t(x_t, y_t) \geq 0, \quad \forall t \geq 0. \tag{11}$$

Thanks to Lemma A.1 (i), we have $\frac{\beta_t + \mu}{2} \|y_t^\star(x_t) - y_t\|^2 \leq H_t$. This relationship shoes that $H_t$ provides a conservative estimate of the squared distance between the dual iterate and the unique maximizer of the regularized payoff.

The following lemma establishes a recursive bound between linking $H_t$ with the regularized payoff and primal functions.

**Lemma A.2.** *Suppose Assumption 1.1 holds. Let $\{x_t, y_t\}_{t \geq 0}$ be any sequence in $\mathcal{X} \times \mathcal{Y}$, and let $\{\beta_t\}_{t \geq 0}$ be a non-increasing sequence of smoothing parameters such that $\max\{\beta_t, \mu\} > 0$. Then, for any $t \geq 0$,*

$$\mathcal{L}_t(x_t, y_t^\star(x_t)) - \mathcal{L}_t(x_t, y_{t+1}) \geq H_{t+1} - \frac{\sqrt{2}\left(L_{yx}\|x_{t+1} - x_t\| + D_\mathcal{Y}(\beta_t - \beta_{t+1})\right)}{\sqrt{\beta_{t+1} + \mu}} \sqrt{H_{t+1}} - 2L_{xx}\|x_{t+1} - x_t\|^2.$$

*Furthermore, we have*

$$
\begin{aligned}
& f_{t+1}(x_{t+1}) + \frac{D_\mathcal{Y}^2 \beta_{t+1}}{2} \\
& \leq f_t(x_t) + \frac{D_\mathcal{Y}^2 \beta_t}{2} + L_{yx}\|y_t^\star(x_t) - y_t\|\|x_{t+1} - x_t\| + \nabla_x \mathcal{L}(x_t, y_t)^\top (x_{t+1} - x_t) + \frac{L_{f_t}}{2}\|x_{t+1} - x_t\|^2 \\
& \leq f_t(x_t) + \frac{D_\mathcal{Y}^2 \beta_t}{2} + L_{yx}\|x_{t+1} - x_t\|\sqrt{\frac{2H_t}{\beta_t + \mu}} + \nabla_x \mathcal{L}(x_t, y_t)^\top (x_{t+1} - x_t) + \frac{L_{f_t}}{2}\|x_{t+1} - x_t\|^2,
\end{aligned}
$$

*where the smoothness constant is defined as $L_{f_t} := L_{xx} + L_{yx}^2/(\beta_t + \mu)$.*

*Proof.* We first prove the first inequality. Using the definition of $y_t^\star$ and the Lipschitz continuity of $\nabla_x \mathcal{L}$, we expand the payoff function at iteration $t$ around $x_{t+1}$. For the regularized optimal response, we have

$$
\mathcal{L}_t(x_t, y_t^\star(x_t)) \geq \mathcal{L}_t(x_t, y_{t+1}^\star(x_{t+1}))
$$

$$
\geq \mathcal{L}_t(x_{t+1}, y_{t+1}^\star(x_{t+1})) + \nabla_x \mathcal{L}_t(x_t, y_{t+1}^\star(x_{t+1}))^\top (x_t - x_{t+1}) - \frac{L_{xx}}{2} \|x_t - x_{t+1}\|^2
$$

$$
= \mathcal{L}_t(x_{t+1}, y_{t+1}^\star(x_{t+1})) + \nabla_x \mathcal{L}(x_t, y_{t+1}^\star(x_{t+1}))^\top (x_t - x_{t+1}) - \frac{L_{xx}}{2} \|x_t - x_{t+1}\|^2,
$$

Similarly, for the dual iterate at $t + 1$, we have

$$
-\mathcal{L}_t(x_t, y_{t+1}) \geq -\mathcal{L}_t(x_{t+1}, y_{t+1}) - \nabla_x \mathcal{L}_t(x_{t+1}, y_{t+1})^\top (x_t - x_{t+1}) - \frac{L_{xx}}{2} \|x_t - x_{t+1}\|^2
$$

$$
= -\mathcal{L}_t(x_{t+1}, y_{t+1}) - \nabla_x \mathcal{L}(x_{t+1}, y_{t+1})^\top (x_t - x_{t+1}) - \frac{L_{xx}}{2} \|x_t - x_{t+1}\|^2.
$$

By summing these two inequalities, we obtain a lower bound for the difference

$$
\mathcal{L}_t(x_t, y_t^\star(x_t)) - \mathcal{L}_t(x_t, y_{t+1})
$$
$$
\geq \mathcal{L}_t(x_{t+1}, y_{t+1}^\star(x_{t+1})) - \mathcal{L}_t(x_{t+1}, y_{t+1})
$$
$$
+ (\nabla_x \mathcal{L}(x_t, y_\mu^\star(x_{t+1})) - \nabla_x \mathcal{L}(x_{t+1}, y_{t+1}))^\top (x_t - x_{t+1}) - L_{xx} \|x_t - x_{t+1}\|^2
$$
$$
\geq \mathcal{L}_t(x_{t+1}, y_{t+1}^\star(x_{t+1})) - \mathcal{L}_t(x_{t+1}, y_{t+1})
$$
$$
- \|\nabla_x \mathcal{L}(x_t, y_{t+1}^\star(x_{t+1})) - \nabla_x \mathcal{L}(x_{t+1}, y_{t+1})\| \|x_t - x_{t+1}\| - L_{xx} \|x_t - x_{t+1}\|^2
$$
$$
\geq \mathcal{L}_t(x_{t+1}, y_{t+1}^\star(x_{t+1})) - \mathcal{L}_t(x_{t+1}, y_{t+1})
$$
$$
- (L_{xx} \|x_t - x_{t+1}\| + L_{yx} \|y_{t+1}^\star(x_{t+1}) - y_{t+1}\|) \|x_t - x_{t+1}\| - L_{xx} \|x_t - x_{t+1}\|^2
$$
$$
\geq \mathcal{L}_t(x_{t+1}, y_{t+1}^\star(x_{t+1})) - \mathcal{L}_t(x_{t+1}, y_{t+1}) - L_{yx} \|x_{t+1} - x_t\| \cdot \|y_{t+1}^\star(x_{t+1}) - y_{t+1}\| - 2L_{xx} \|x_{t+1} - x_t\|^2,
$$

where the third inequality follows from Lipschitz continuity of $\nabla_x \mathcal{L}$. Using the definition of $\mathcal{L}_t$, the difference term $\mathcal{L}_t(x_{t+1}, y_{t+1}^\star(x_{t+1})) - \mathcal{L}_t(x_{t+1}, y_{t+1})$ in the last inequality can be written in terms of $H_{t+1}$ as follows

$$
\mathcal{L}_t(x_{t+1}, y_{t+1}^\star(x_{t+1})) - \mathcal{L}_t(x_{t+1}, y_{t+1})
$$
$$
= \mathcal{L}_{t+1}(x_{t+1}, y_{t+1}^\star(x_{t+1})) - \mathcal{L}_{t+1}(x_{t+1}, y_{t+1}) + \frac{\beta_{t+1} - \beta_t}{2} \left( \|y_{t+1}^\star(x_{t+1}) - y_0\|^2 - \|y_{t+1} - y_0\|^2 \right)
$$
$$
= H_{t+1} + \frac{\beta_{t+1} - \beta_t}{2} \left( \|y_{t+1}^\star(x_{t+1}) - y_0\|^2 - \|y_{t+1} - y_0\|^2 \right).
$$

By the property $\|a\|^2 - \|b\|^2 = (a - b)^\top (a + b)$ and the boundedness of $\mathcal{Y}$, it follows that

$$
\|y_{t+1}^\star(x_{t+1}) - y_0\|^2 - \|y_{t+1} - y_0\|^2 = (y_{t+1}^\star(x_{t+1}) - y_{t+1})^\top (y_{t+1}^\star(x_{t+1}) - y_0 + y_{t+1} - y_0)
$$
$$
\leq \|y_{t+1}^\star(x_{t+1}) - y_{t+1}\| \left( \|y_{t+1}^\star(x_{t+1}) - y_0\| + \|y_{t+1} - y_0\| \right)
$$
$$
\leq 2 D_\mathcal{Y} \|y_{t+1}^\star(x_{t+1}) - y_{t+1}\|,
$$

where the second inequality follows from the Cauchy-Schwartz and triangle inequality. Combining these results, we have

$$
\mathcal{L}_t(x_t, y_t^\star(x_t)) - \mathcal{L}_t(x_t, y_{t+1})
$$
$$
\geq H_{t+1} - (L_{yx} \|x_{t+1} - x_t\| + D_\mathcal{Y}(\beta_t - \beta_{t+1})) \|y_{t+1}^\star(x_{t+1}) - y_{t+1}\| - 2L_{xx} \|x_{t+1} - x_t\|^2.
$$

Applying Lemma A.1 (i) on the term $\|y_{t+1}^\star(x_{t+1}) - y_{t+1}\|$ concludes the proof of the first part.

For the second part, the smoothness of $f_t$ established in Lemma A.1 (v) implies

$$
f_t(x_{t+1})
$$
$$
\leq f_t(x_t) + \nabla f_t(x_t)^\top (x_{t+1} - x_t) + \frac{L_{f_t}}{2} \|x_{t+1} - x_t\|^2
$$

$$= f_t(x_t) + (\nabla f_t(x_t) - \nabla_x \mathcal{L}(x_t, y_t))^\top (x_{t+1} - x_t) + \nabla_x \mathcal{L}(x_t, y_t)^\top (x_{t+1} - x_t) + \frac{L_{f_t}}{2} \|x_{t+1} - x_t\|^2$$

$$= f_t(x_t) + (\nabla_x \mathcal{L}(x_t, y_t^\star(x_t)) - \nabla_x \mathcal{L}(x_t, y_t))^\top (x_{t+1} - x_t) + \nabla_x \mathcal{L}(x_t, y_t)^\top (x_{t+1} - x_t) + \frac{L_{f_t}}{2} \|x_{t+1} - x_t\|^2$$

$$\leq f_t(x_t) + \|\nabla_x \mathcal{L}(x_t, y_t^\star(x_t)) - \nabla_x \mathcal{L}(x_t, y_t)\| \cdot \|x_{t+1} - x_t\| + \nabla_x \mathcal{L}(x_t, y_t)^\top (x_{t+1} - x_t) + \frac{L_{f_t}}{2} \|x_{t+1} - x_t\|^2$$

$$\leq f_t(x_t) + L_{yx} \|y_t^\star(x_t) - y_t\| \cdot \|x_{t+1} - x_t\| + \nabla_x \mathcal{L}(x_t, y_t)^\top (x_{t+1} - x_t) + \frac{L_{f_t}}{2} \|x_{t+1} - x_t\|^2,$$

where the second inequality follows from Cauchy-Schwartz inequality and the third inequality follows from the Lipschitz continuity of $\nabla_x \mathcal{L}$. By Lemma A.1 (vi), we have

$$f_{t+1}(x_{t+1}) - f_t(x_{t+1}) \leq \frac{D_{\mathcal{Y}}^2}{2} (\beta_t - \beta_{t+1}) \iff f_{t+1}(x_{t+1}) + \frac{D_{\mathcal{Y}}^2}{2} \beta_{t+1} \leq f_t(x_{t+1}) + \frac{D_{\mathcal{Y}}^2}{2} \beta_t,$$

which together with applying Lemma A.1 (i) on $\|y_t^\star(x_t) - y_t\|$ implies the second part. This completes the proof. $\qquad \square$

To implement single-loop strategies, we formally define two classes of oracle updates used in our algorithms.

**Definition A.3** (LMO Sequences). Suppose $\{x_t, y_t\}_{t \geq 0}$ is a sequence in $\mathcal{X} \times \mathcal{Y}$.

$\diamond$ A sequence $\{x_t\}_{t \geq 0} \subset \mathcal{X}$ is a *primal LMO sequence* with stepsizes $\{\tau_t\}_{t \geq 0} \subset [0, 1]$ if for any $t \geq 0$

$$x_{t+1} = x_t + \tau_t (v_t - x_t), \text{ where } v_t = \mathrm{LMO}_{\mathcal{X}}(\nabla_x \mathcal{L}(x_t, y_t)). \tag{12a}$$

$\diamond$ A sequence $\{y_t\}_{t \geq 0} \subset \mathcal{Y}$ is a *dual LMO sequence* if for any $t \geq 0$ we have $\max\{\beta_t, \mu\} > 0$ and

$$y_{t+1} = y_t + \gamma_t (u_t - y_t), \text{ where } u_t = \mathrm{LMO}_{\mathcal{Y}}(-\nabla_y \mathcal{L}_{\beta_t}(x_t, y_t)) \text{ and } \gamma_t = \min \left\{ \frac{\nabla_y \mathcal{L}_{\beta_t}(x_t, y_t)^\top (u_t - y_t)}{(L_{yy} + \beta_t)\|u_t - y_t\|^2}, 1 \right\}. \tag{12b}$$

The dual updates utilize the property that $\mathcal{L}_t(x, \cdot)$ is $(L_{yy} + \beta_t)$-smooth and concave under Assumption 1.1. The stepsize $\gamma_t$ in the dual LMO sequence represents a *closed-loop strategy* that maximizes the quadratic lower bound of the objective. Similarly, the dual PO update below uses a stepsize that maximizes the corresponding quadratic lower bound.

**Definition A.4** (PO Sequences). Suppose $\{x_t, y_t\}_{t \geq 0}$ is a sequence in $\mathcal{X} \times \mathcal{Y}$.

$\diamond$ A sequence $\{x_t\}_{t \geq 0} \subset \mathcal{X}$ is a *primal PO sequence* with stepsizes $\{\tau_t\}_{t \geq 0} \subset \mathbb{R}_{++}$ if for any $t \geq 0$

$$x_{t+1} = \mathrm{PO}_{\mathcal{X}} \left( x_t - \tau_t \nabla_x \mathcal{L}(x_t, y_t) \right). \tag{12c}$$

$\diamond$ A sequence $\{y_t\}_{t \geq 0} \subset \mathcal{Y}$ is a *dual PO sequence* if for any $t \geq 0$ we have $\max\{\beta_t, \mu\} > 0$ and

$$y_{t+1} = \mathrm{PO}_{\mathcal{Y}} \left( y_t + \gamma_t \nabla_y \mathcal{L}_t(x_t, y_t) \right), \text{ where } \gamma_t = 1/(L_{yy} + \beta_t). \tag{12d}$$

The subsequent sections provide convergence results for every combination of these primal and dual updates to establish a unified analysis for our single-loop framework.

### A.3. Primal LMO sequence

We characterize the iterate progress when the primal sequence is updated via an LMO according to Definition A.3. The following lemma instantiates Lemma A.2 for the primal LMO update and establishes a recursive bound linking $H_t$ with the regularized payoff and primal functions.

**Lemma A.5.** *Suppose Assumption 1.1 holds. Let $\{x_t\}_{t \geq 0}$ be a primal LMO sequence with stepsizes $\{\tau_t\}_{t \geq 0}$ generated by the update rule* (12a), *let $\{y_t\}_{t \geq 0} \subset \mathcal{Y}$ be any dual sequence, and let $\{\beta_t\}_{t \geq 0}$ be a non-increasing sequence of smoothing parameters such that $\max\{\beta_t, \mu\} > 0$. Then, for any $t \geq 0$,*

$$\mathcal{L}_t(x_t, y_t^\star(x_t)) - \mathcal{L}_t(x_t, y_{t+1}) \geq H_{t+1} - \frac{\sqrt{2} \left( L_{yx} D_{\mathcal{X}} \tau_t + D_{\mathcal{Y}} (\beta_t - \beta_{t+1}) \right)}{\sqrt{\beta_{t+1} + \mu}} \sqrt{H_{t+1}} - 2L_{xx} D_{\mathcal{X}}^2 \tau_t^2,$$

*and if additionally $\beta_t \geq \beta_{t+1}$ then*

$$f_{t+1}(x_{t+1}) + \frac{D_{\mathcal{Y}}^2}{2}\beta_{t+1}$$

$$\leq f_t(x_t) + \frac{D_{\mathcal{Y}}^2}{2}\beta_t + L_{yx}D_{\mathcal{X}}\tau_t\|y_t^\star(x_t) - y_t\| + \tau_t\nabla_x\mathcal{L}(x_t, y_t)^\top(v_t - x_t) + \left(L_{xx}D_{\mathcal{X}}^2 + \frac{L_{yx}^2 D_{\mathcal{X}}^2}{\beta_t + \mu}\right)\frac{\tau_t^2}{2}$$

$$\leq f_t(x_t) + \frac{D_{\mathcal{Y}}^2}{2}\beta_t + L_{yx}D_{\mathcal{X}}\tau_t\sqrt{\frac{2H_t}{\beta_t + \mu}} + \tau_t\nabla_x\mathcal{L}(x_t, y_t)^\top(v_t - x_t) + \left(L_{xx}D_{\mathcal{X}}^2 + \frac{L_{yx}^2 D_{\mathcal{X}}^2}{\beta_t + \mu}\right)\frac{\tau_t^2}{2}.$$

*Proof.* The proof directly follows from Lemma A.2 by substituting the primal LMO update $x_{t+1} - x_t = \tau_t(v_t - x_t)$ and applying the bound $\|v_t - x_t\| \leq D_{\mathcal{X}}$. The details are omitted for brevity. $\qquad\square$

The next lemma provides a bound on the LMO and dual gaps in terms of the $H_t$.

**Lemma A.6.** *Under the assumptions of Lemma A.5, for any $t \geq 0$, the dual stationarity measure satisfies*

$$G_{\mathcal{Y}}(x_t, y_t) \leq H_t + \frac{D_{\mathcal{Y}}^2\beta_t}{2}.$$

*Furthermore, if the primal stepsizes satisfy $\tau_t \geq \tau_{t+1}$, the accumulated primal stationarity gap satisfies*

$$\sum_{t=0}^{T} G_{\mathcal{X}}^{\mathrm{LMO}}(x_t, y_t) \leq \frac{2\max_{x \in \mathcal{X}}|f(x)| + \beta_0 D_{\mathcal{Y}}^2}{\tau_T} + L_{yx}D_{\mathcal{X}}\sum_{t=0}^{T}\|y_t^\star(x_t) - y_t\| + \sum_{t=0}^{T}\left(L_{xx}D_{\mathcal{X}}^2 + \frac{L_{yx}^2 D_{\mathcal{X}}^2}{\beta_t + \mu}\right)\frac{\tau_t}{2}$$

$$\leq \frac{2\max_{x \in \mathcal{X}}|f(x)| + \beta_0 D_{\mathcal{Y}}^2}{\tau_T} + L_{yx}D_{\mathcal{X}}\sum_{t=0}^{T}\sqrt{\frac{2H_t}{\beta_t + \mu}} + \sum_{t=0}^{T}\left(L_{xx}D_{\mathcal{X}}^2 + \frac{L_{yx}^2 D_{\mathcal{X}}^2}{\beta_t + \mu}\right)\frac{\tau_t}{2}.$$

*Proof.* For any $y \in \mathcal{Y}$, we have

$$\mathcal{L}_t(x_t, y) - \mathcal{L}_t(x_t, y_t) = \mathcal{L}(x_t, y) - \mathcal{L}(x_t, y_t) - \frac{\beta_t}{2}\|y - y_0\|^2 + \frac{\beta_t}{2}\|y_t - y_0\|^2$$

$$\geq \mathcal{L}(x_t, y) - \mathcal{L}(x_t, y_t) - \frac{D_{\mathcal{Y}}^2\beta_t}{2}.$$

By maximizing both sides over $y \in \mathcal{Y}$ and using the definition of $H_t$, we obtain

$$G_{\mathcal{Y}}(x_t, y_t) = \max_{y \in \mathcal{Y}}\mathcal{L}(x_t, y) - \mathcal{L}(x_t, y_t) \leq H_t + \frac{D_{\mathcal{Y}}^2\beta_t}{2}.$$

This concludes the proof of the first part.

Using Lemma A.5, we have

$$f_{t+1}(x_{t+1}) + \frac{D_{\mathcal{Y}}^2}{2}\beta_{t+1} - f_t(x_t) - \frac{D_{\mathcal{Y}}^2}{2}\beta_t$$

$$\leq L_{yx}D_{\mathcal{X}}\tau_t\|y_t^\star(x_t) - y_t\| + \tau_t\nabla_x\mathcal{L}(x_t, y_t)^\top(v_t - x_t) + \left(L_{xx}D_{\mathcal{X}}^2 + \frac{L_{yx}^2 D_{\mathcal{X}}^2}{\beta_t + \mu}\right)\frac{\tau_t^2}{2},$$

which rearranges to

$$G_{\mathcal{X}}^{\mathrm{LMO}}(x_t, y_t) = \nabla_x\mathcal{L}(x_t, y_t)^\top(x_t - v_t)$$

$$\leq \frac{f_t(x_t) + \frac{D_{\mathcal{Y}}^2}{2}\beta_t}{\tau_t} - \frac{f_{t+1}(x_{t+1}) + \frac{D_{\mathcal{Y}}^2}{2}\beta_{t+1}}{\tau_t} + L_{yx}D_{\mathcal{X}}\|y_t^\star(x_t) - y_t\| + \left(L_{xx}D_{\mathcal{X}}^2 + \frac{L_{yx}^2 D_{\mathcal{X}}^2}{\beta_t + \mu}\right)\frac{\tau_t}{2}.$$

Summing the above inequality for $t = 0, \ldots, T$, we obtain

$$
\sum_{t=0}^{T} G_{\mathcal{X}}^{\text{LMO}}(x_t, y_t) \leq \sum_{t=0}^{T} \left( \frac{f_t(x_t) + \frac{D_{\mathcal{Y}}^2}{2}\beta_t}{\tau_t} - \frac{f_{t+1}(x_{t+1}) + \frac{D_{\mathcal{Y}}^2}{2}\beta_{t+1}}{\tau_t} \right)
$$

$$
+ L_{yx} D_{\mathcal{X}} \sum_{t=0}^{T} \|y_t^\star(x_t) - y_t\| + \sum_{t=0}^{T} \left( L_{xx} D_{\mathcal{X}}^2 + \frac{L_{yx}^2 D_{\mathcal{X}}^2}{\beta_t + \mu} \right) \frac{\tau_t}{2}
$$

$$
= \frac{f_0(x_0) + \frac{D_{\mathcal{Y}}^2}{2}\beta_0}{\tau_0} + \sum_{t=1}^{T} \left( \frac{1}{\tau_t} - \frac{1}{\tau_{t-1}} \right) \left( f_t(x_t) + \frac{D_{\mathcal{Y}}^2}{2}\beta_t \right) - \frac{f_{T+1}(x_{T+1}) + \frac{D_{\mathcal{Y}}^2}{2}\beta_{T+1}}{\tau_T}
$$

$$
+ L_{yx} D_{\mathcal{X}} \sum_{t=0}^{T} \|y_t^\star(x_t) - y_t\| + \sum_{t=0}^{T} \left( L_{xx} D_{\mathcal{X}}^2 + \frac{L_{yx}^2 D_{\mathcal{X}}^2}{\beta_t + \mu} \right) \frac{\tau_t}{2}.
$$

For any $t \geq 0$, it holds from Lemma A.1 (vi) that

$$
f(x_t) \leq f_t(x_t) + \frac{D_{\mathcal{Y}}^2}{2}\beta_t \leq f(x_t) + \frac{D_{\mathcal{Y}}^2}{2}\beta_0,
$$

which implies

$$
\left| f_t(x_t) + \frac{D_{\mathcal{Y}}^2}{2}\beta_t \right| \leq \max_{x \in \mathcal{X}} |f(x)| + \frac{D_{\mathcal{Y}}^2}{2}\beta_0.
$$

Since $\tau_t \leq \tau_{t-1}$, we have

$$
\sum_{t=0}^{T} G_{\mathcal{X}}^{\text{LMO}}(x_t, y_t)
$$

$$
\leq \left( \max_{x \in \mathcal{X}} |f(x)| + \frac{D_{\mathcal{Y}}^2}{2}\beta_0 \right) \left( \frac{1}{\tau_0} + \sum_{t=1}^{T} \left( \frac{1}{\tau_t} - \frac{1}{\tau_{t-1}} \right) + \frac{1}{\tau_T} \right) + L_{yx} D_{\mathcal{X}} \sum_{t=0}^{T} \|y_t^\star(x_t) - y_t\|
$$

$$
+ \sum_{t=0}^{T} \left( L_{xx} D_{\mathcal{X}}^2 + \frac{L_{yx}^2 D_{\mathcal{X}}^2}{\beta_t + \mu} \right) \frac{\tau_t}{2}
$$

$$
= \frac{2 \max_{x \in \mathcal{X}} |f(x)| + \beta_0 D_{\mathcal{Y}}^2}{\tau_T} + L_{yx} D_{\mathcal{X}} \sum_{t=0}^{T} \|y_t^\star(x_t) - y_t\| + \sum_{t=0}^{T} \left( L_{xx} D_{\mathcal{X}}^2 + \frac{L_{yx}^2 D_{\mathcal{X}}^2}{\beta_t + \mu} \right) \frac{\tau_t}{2}.
$$

Applying the strong concavity property from Lemma A.1 (i) completes the proof. $\square$

The guarantees established in Lemma A.5 for the primal objective function can be further refined under the additional assumption that the payoff function is convex-concave.

**Lemma A.7.** *Under the assumptions of Lemma A.5 and the additional Assumption 1.4, for any $t \geq 0$,*

$$
f_{t+1}(x_{t+1}) - f_{t+1}(x^\star) + \frac{D_{\mathcal{Y}}^2 \beta_{t+1}}{2}
$$

$$
\leq (1 - \tau_t) \left( f_t(x_t) - f_t(x^\star) + \frac{D_{\mathcal{Y}}^2 \beta_t}{2} \right) + \frac{D_{\mathcal{Y}}^2 \tau_t \beta_t}{2} + 2L_{yx} D_{\mathcal{X}} \tau_t \|y_t - y_t^\star(x_t)\| + \left( L_{xx} D_{\mathcal{X}}^2 + \frac{L_{yx}^2 D_{\mathcal{X}}^2}{\beta_t + \mu} \right) \frac{\tau_t^2}{2}
$$

$$
\leq (1 - \tau_t) \left( f_t(x_t) - f_t(x^\star) + \frac{D_{\mathcal{Y}}^2 \beta_t}{2} \right) + \frac{D_{\mathcal{Y}}^2 \tau_t \beta_t}{2} + 2L_{yx} D_{\mathcal{X}} \tau_t \sqrt{\frac{2H_t}{\beta_t + \mu}} + \left( L_{xx} D_{\mathcal{X}}^2 + \frac{L_{yx}^2 D_{\mathcal{X}}^2}{\beta_t + \mu} \right) \frac{\tau_t^2}{2}.
$$

*Proof.* From the definition of $v_t$ as the LMO solution, it holds that

$$
\nabla_x \mathcal{L}(x_t, y_t)^\top (v_t - x_t) \leq \nabla_x \mathcal{L}(x_t, y_t)^\top (x^\star - x_t)
$$

$$= \nabla_x \mathcal{L}(x_t, y_t^\star(x_t))^\top (x^\star - x_t) + (\nabla_x \mathcal{L}(x_t, y_t) - \nabla_x \mathcal{L}(x_t, y_t^\star(x_t)))^\top (x^\star - x_t).$$

Under Assumption 1.4, $L(\cdot, y)$ is convex, which implies that the regularized primal function $f_t$ is also convex on $\mathcal{X}$. By the gradient property of smoothed functions in Lemma A.1 (v), we have

$$\nabla_x \mathcal{L}(x_t, y_t^\star(x_t))^\top (x^\star - x_t) = \nabla_x \mathcal{L}_t(x_t, y_t^\star(x_t))^\top (x^\star - x_t) = \nabla f_t(x_t)^\top (x^\star - x_t) \leq f_t(x^\star) - f_t(x_t).$$

Applying the Cauchy-Schwarz inequality and the Lipschitz continuity of $\nabla_x \mathcal{L}$ yields

$$(\nabla_x \mathcal{L}(x_t, y_t) - \nabla_x \mathcal{L}(x_t, y_t^\star(x_t)))^\top (x^\star - x_t) \leq \|\nabla_x \mathcal{L}(x_t, y_t) - \nabla_x \mathcal{L}(x_t, y_t^\star(x_t))\| \|x^\star - x_t\|$$
$$\leq L_{yx} D_{\mathcal{X}} \|y_t - y_t^\star(x_t)\|.$$

where we used the fact that $\|x^\star - x_t\| \leq D_{\mathcal{X}}$. We thus obtain

$$\nabla_x \mathcal{L}(x_t, y_t)^\top (v_t - x_t) \leq L_{yx} D_{\mathcal{X}} \|y_t - y_t^\star(x_t)\| + f_t(x^\star) - f_t(x_t).$$

Substituting this bound into the primal progress result from Lemma A.5 leads to

$$f_t(x_{t+1}) - f_t(x^\star) \leq (1 - \tau_t)(f_t(x_t) - f_t(x^\star)) + 2L_{yx} D_{\mathcal{X}} \tau_t \|y_t - y_t^\star(x_t)\| + \left( \frac{L_{xx} D_{\mathcal{X}}^2}{2} + \frac{L_{yx}^2 D_{\mathcal{X}}^2}{2\beta_t} \right) \tau_t^2. \tag{13}$$

From the properties of changing the smoothing parameter in Lemma A.1 (vi), we have

$$f_{t+1}(x_{t+1}) - f_{t+1}(x^\star) + \frac{D_{\mathcal{Y}}^2}{2} \beta_{t+1} \leq \left( f_t(x_{t+1}) + \frac{D_{\mathcal{Y}}^2}{2} \beta_t - \frac{D_{\mathcal{Y}}^2}{2} \beta_{t+1} \right) - f_t(x^\star) + \frac{D_{\mathcal{Y}}^2}{2} \beta_{t+1}$$

$$= f_t(x_{t+1}) - f_t(x^\star) + \frac{D_{\mathcal{Y}}^2}{2} \beta_t,$$

Combining this with (13) and the strong concavity relation in Lemma A.1 (i) completes the proof. $\qquad \square$

### A.4. Dual LMO sequence

We characterize the iterate progress when the dual sequence is updated via an LMO according to Definition A.3, for which the stepsize is adaptive and follows a closed-loop strategy. The following lemma establishes a recursive bound linking $H_t$ with the regularized payoff functions.

**Lemma A.8.** *Suppose Assumption 1.1 holds. Let $\{x_t\}_{t \geq 0} \subset \mathcal{X}$ be any primal sequence and let $\{y_t\}_{t \geq 0}$ be a dual LMO sequence generated by the update rule* (12b). *Then, for any $t \geq 0$,*

$$\mathcal{L}_t(x_t, y_t^\star(x_t)) - \mathcal{L}_t(x_t, y_{t+1}) \leq \max \left\{ \frac{1}{2}, 1 - \frac{H_t}{2(L_{yy} + \beta_t) D_{\mathcal{Y}}^2} \right\} H_t. \tag{14}$$

*Proof.* From the Lipschitz continuity of $\nabla_y \mathcal{L}_t$ and the dual update rule, we have that

$$-\mathcal{L}_t(x_t, y_{t+1}) \leq -\mathcal{L}_t(x_t, y_t) - \gamma_t \nabla_y \mathcal{L}_t(x_t, y_t)^\top (u_t - y_t) + \frac{L_{yy} + \beta_t}{2} \|u_t - y_t\|^2 \gamma_t^2. \tag{15}$$

If

$$\frac{\nabla_y \mathcal{L}_t(x_t, y_t)^\top (u_t - y_t)}{(L_{yy} + \beta_t) \|u_t - y_t\|^2} \geq 1 \iff \nabla_y \mathcal{L}_t(x_t, y_t)^\top (u_t - y_t) \geq (L_{yy} + \beta_t) \|u_t - y_t\|^2,$$

then the dual stepsize $\gamma_t = 1$ and

$$-\mathcal{L}_t(x_t, y_{t+1}) \leq -\mathcal{L}_t(x_t, y_t) - \nabla_y \mathcal{L}_t(x_t, y_t)^\top (u_t - y_t) + \frac{L_{yy} + \beta_t}{2} \|u_t - y_t\|^2$$

$$\leq -\mathcal{L}_t(x_t, y_t) - \frac{1}{2} \nabla_y \mathcal{L}_t(x_t, y_t)^\top (u_t - y_t).$$

Otherwise, $\gamma_t = \frac{\nabla_y \mathcal{L}_t(x_t,y_t)^\top (u_t - y_t)}{(L_{yy} + \beta_t)\|u_t - y_t\|^2}$, and we obtain

$$-\mathcal{L}_t(x_t, y_{t+1}) \leq -\mathcal{L}_t(x_t, y_t) - \frac{(\nabla_y \mathcal{L}_t(x_t,y_t)^\top (u_t - y_t))^2}{(L_{yy} + \beta_t)\|u_t - y_t\|^2} + \frac{1}{2} \frac{(\nabla_y \mathcal{L}_t(x_t,y_t)^\top (u_t - y_t))^2}{(L_{yy} + \beta_t)\|u_t - y_t\|^2}$$

$$\leq -\mathcal{L}_t(x_t, y_t) - \frac{1}{2} \frac{(\nabla_y \mathcal{L}_t(x_t,y_t)^\top (u_t - y_t))^2}{(L_{yy} + \beta_t)\|u_t - y_t\|^2}.$$

Combining these two cases yields

$$-\mathcal{L}_t(x_t, y_{t+1}) \leq -\mathcal{L}_t(x_t, y_t) - \frac{1}{2} \min\left\{1, \frac{\nabla_y \mathcal{L}_t(x_t,y_t)^\top (u_t - y_t)}{(L_{yy} + \beta_t)\|u_t - y_t\|^2}\right\} \nabla_y \mathcal{L}_t(x_t,y_t)^\top (u_t - y_t)$$

By the definition of $u_t$ and the concavity of $\mathcal{L}_t(x_t, \cdot)$ it holds that

$$\nabla_y \mathcal{L}_t(x_t,y_t)^\top (u_t - y_t) \geq \nabla_y \mathcal{L}_t(x_t,y_t)^\top (y_t^\star(x_t) - y_t) \geq \mathcal{L}_t(x_t, y_t^\star(x_t)) - \mathcal{L}_t(x_t, y_t) = H_t \geq 0.$$

Applying this relation and the diameter bound $\|u_t - y_t\| \leq D_{\mathcal{Y}}$ we deduce

$$\mathcal{L}_t(x_t, y_t^\star(x_t)) - \mathcal{L}_t(x_t, y_{t+1}) \leq H_t - \frac{1}{2} \min\left\{1, \frac{H_t}{(L_{yy} + \beta_t)D_{\mathcal{Y}}^2}\right\} H_t,$$

which concludes the proof. $\qquad\square$

The guarantees established in Lemma A.8 for the regularized payoff function can be further refined under the additional assumption that the feasible set $\mathcal{Y}$ is strongly convex.

**Lemma A.9.** *Under the assumptions of Lemma A.8 and the additional Assumption 1.2, for any $t \geq 0$,*

$$\mathcal{L}_t(x_t, y_t^\star(x_t)) - \mathcal{L}_t(x_t, y_{t+1}) \leq \max\left\{\frac{1}{2}, 1 - \frac{\alpha\sqrt{(\beta_t + \mu)H_t}}{8\sqrt{2}\,(L_{yy} + \beta_t)}\right\} H_t. \tag{16}$$

*Proof.* Using inequality (15) and definition of $H_t$, we have

$$\mathcal{L}_t(x_t, y_t^\star(x_t)) - \mathcal{L}_t(x_t, y_{t+1}) \leq H_t - \gamma_t \nabla_y \mathcal{L}_t(x_t,y_t)^\top (u_t - y_t) + \frac{L_{yy} + \beta_t}{2}\|u_t - y_t\|^2 \gamma_t^2.$$

Since $\gamma_t$ is chosen to minimize this quadratic upper bound, the following inequality remains valid for any $\gamma \in [0, 1]$

$$\mathcal{L}_t(x_t, y_t^\star(x_t)) - \mathcal{L}_t(x_t, y_{t+1}) \leq H_t - \gamma \nabla_y \mathcal{L}_t(x_t,y_t)^\top (u_t - y_t) + \frac{L_{yy} + \beta_t}{2}\|u_t - y_t\|^2 \gamma^2. \tag{17}$$

To exploit the geometry of the constraint set, we define the point

$$p_t := \frac{1}{2}(y_t + u_t) + \frac{\alpha}{8}\|y_t - u_t\|^2 w_t,$$

where $w_t \in \mathbb{R}^{n_y}$ satisfies $w_t^\top \nabla_y \mathcal{L}_t(x_t,y_t) = \|\nabla_y \mathcal{L}_t(x_t,y_t)\|$. By construction and Assumption 1.2, we have $p_t \in \mathcal{Y}$. Besides, by the optimality of $u_t$ for the dual LMO update, it holds that

$$\nabla_y \mathcal{L}_t(x_t,y_t)^\top (u_t - y_t) \geq \nabla_y \mathcal{L}_t(x_t,y_t)^\top (p_t - y_t)$$

$$= \frac{1}{2}\nabla_y \mathcal{L}_t(x_t,y_t)^\top (u_t - y_t) + \frac{\alpha}{8}\|y_t - u_t\|^2 \|\nabla_y \mathcal{L}_t(x_t,y_t)\|$$

$$\geq \frac{1}{2}\nabla_y \mathcal{L}_t(x_t,y_t)^\top (y_t^\star(x_t) - y_t) + \frac{\alpha}{8}\|y_t - u_t\|^2 \|\nabla_y \mathcal{L}_t(x_t,y_t)\|$$

$$\geq \frac{1}{2}H_t + \frac{\alpha}{8}\|y_t - u_t\|^2 \|\nabla_y \mathcal{L}_t(x_t,y_t)\|,$$

where the last inequality follows from concavity of $\mathcal{L}(x_t, \cdot)$ and the definition of $H_t$. Using the above lower bound and plugging it in (17), we may conclude that

$$\mathcal{L}_t(x_t, y_t^\star(x_t)) - \mathcal{L}_t(x_t, y_{t+1}) \leq \left(1 - \frac{\gamma}{2}\right) H_t + \frac{\|u_t - y_t\|^2}{2} \left((L_{yy} + \beta_t)\gamma^2 - \frac{\alpha}{4}\|\nabla_y \mathcal{L}_t(x_t, y_t)\|\gamma\right).$$

If $(L_{yy} + \beta_t) < \frac{\alpha}{4}\|\nabla_y \mathcal{L}_t(x_t, y_t)\|$, then we choose $\gamma = 1$ and hence,

$$\mathcal{L}_t(x_t, y_t^\star(x_t)) - \mathcal{L}_t(x_t, y_{t+1}) \leq \frac{1}{2} H_t.$$

Otherwise, we choose $\gamma = \frac{\alpha}{4(L_{yy} + \beta_t)}\|\nabla_y \mathcal{L}_t(x_t, y_t)\|$ and hence,

$$\mathcal{L}_t(x_t, y_t^\star(x_t)) - \mathcal{L}_t(x_t, y_{t+1}) \leq \left(1 - \frac{\alpha\|\nabla_y \mathcal{L}_t(x_t, y_t)\|}{8(L_{yy} + \beta_t)}\right) H_t.$$

Applying the strong concavity property from Lemma A.1 (i), which implies $\|\nabla_y \mathcal{L}_t(x_t, y_t)\| \geq \sqrt{2(\beta_t + \mu)H_t}$, we obtain the desired bound. $\qquad\square$

## A.5. Primal PO sequence

Under the primal PO sequence, the stationarity measure is characterized by $G_{\mathcal{X}}^{\mathrm{PO}}$ as defined in (4b) with $\sigma = \tau_0$:

$$G_{\mathcal{X}}^{\mathrm{PO}}(x_t, y_t) = \left\|\frac{1}{\tau_0}\left(\mathrm{PO}(x_t - \tau_0 \nabla_x \mathcal{L}(x_t, y_t)) - x_t\right)\right\| \geq \left\|\frac{1}{\tau_t}\left(\mathrm{PO}(x_t - \tau_t \nabla_x \mathcal{L}(x_t, y_t)) - x_t\right)\right\| = \left\|\frac{x_{t+1} - x_t}{\tau_t}\right\|,$$

where the inequality follows from Beck (2017, Theorem 10.9) when $\tau_0 \geq \tau_t$. The following lemma establishes a fundamental inequality for this sequence.

**Lemma A.10.** *Under Assumption 1.1, let $\{x_t\}_{t \geq 0}$ be a primal PO sequence with stepsizes $\{\tau_t\}_{t \geq 0}$ generated by the update rule (12c), let $\{y_t\}_{t \geq 0} \subset \mathcal{Y}$ be any dual sequence, and let $\{\beta_t\}_{t \geq 0}$ be a non-increasing sequence of smoothing parameters such that $\max\{\beta_t, \mu\} > 0$. Then, for any $t \geq 0$,*

$$\left(\frac{3}{4\tau_t} - \frac{L_{xx}}{2} - \frac{5L_{yx}^2}{2(\beta_t + \mu)}\right)\|x_{t+1} - x_t\|^2$$

$$\leq \left(f_t(x_t) + \frac{D_{\mathcal{Y}}^2 \beta_t}{2}\right) - \left(f_{t+1}(x_{t+1}) + \frac{D_{\mathcal{Y}}^2 \beta_{t+1}}{2}\right) + \frac{1}{4}\min\left\{H_t, \frac{H_t^2}{(L_{yy} + \beta_t)D_{\mathcal{Y}}^2}\right\} + \frac{4L_{yx}^4(L_{yy} + \beta_t)D_{\mathcal{Y}}^2 \tau_t^2}{(\beta_t + \mu)^2}.$$

*Proof.* For any $x \in \mathcal{X}$, the first-order optimality condition for the projection step implies

$$(x_{t+1} - x_t + \tau_t \nabla_x \mathcal{L}(x_t, y_t))^\top (x - x_{t+1}) \geq 0.$$

Setting $x = x_t$ yields the following bound on the linearized gradient

$$\nabla_x \mathcal{L}(x_t, y_t)^\top (x_{t+1} - x_t) \leq -\frac{1}{\tau_t}\|x_{t+1} - x_t\|^2.$$

Applying Lemma A.2 and the strong concavity relation $\|y_t^\star(x_t) - y_t\| \leq \sqrt{2H_t/(\beta_t + \mu)}$, we have

$$f_{t+1}(x_{t+1}) + \frac{D_{\mathcal{Y}}^2 \beta_{t+1}}{2}$$

$$\leq f_t(x_t) + \frac{D_{\mathcal{Y}}^2 \beta_t}{2} + L_{yx}\|y_t^\star(x_t) - y_t\|\|x_{t+1} - x_t\| - \left(\frac{1}{\tau_t} - \frac{L_{f_t}}{2}\right)\|x_{t+1} - x_t\|^2$$

$$\leq f_t(x_t) + \frac{D_{\mathcal{Y}}^2 \beta_t}{2} + L_{yx}\|x_{t+1} - x_t\|\sqrt{\frac{2H_t}{\beta_t + \mu}} - \left(\frac{1}{\tau_t} - \frac{L_{f_t}}{2}\right)\|x_{t+1} - x_t\|^2. \tag{18}$$

To handle the interaction term, we apply the AM-GM inequality in two ways. First, using the inequality for four numbers, we have

$$L_{yx}\|x_{t+1} - x_t\|\sqrt{\frac{2H_t}{\beta_t + \mu}} = \left(\left(\frac{\sqrt{2}L_{yx}(L_{yy} + \beta_t)^{1/4}D_{\mathcal{Y}}^{1/2}\|x_{t+1} - x_t\|}{\sqrt{\beta_t + \mu}}\right)^{1/3}\right)^3 \sqrt{\frac{H_t}{(L_{yy} + \beta_t)^{1/2}D_{\mathcal{Y}}}}$$

$$\leq \frac{H_t^2}{4(L_{yy} + \beta_t)D_{\mathcal{Y}}^2} + \frac{3}{4}\left(\frac{\sqrt{2}L_{yx}(L_{yy} + \beta_t)^{1/4}D_{\mathcal{Y}}^{1/2}\|x_{t+1} - x_t\|}{\sqrt{\beta_t + \mu}}\right)^{4/3}.$$

Second, by the standard AM-GM inequality for two numbers, we obtain

$$\frac{2L_{yx}\|x_{t+1} - x_t\|}{\sqrt{\beta_t + \mu}}\sqrt{\frac{H_t}{2}} \leq \frac{H_t}{4} + \frac{2L_{yx}^2\|x_{t+1} - x_t\|^2}{\beta_t + \mu}.$$

Combining these two bounds, the interaction term is bounded as follows

$$L_{yx}\|x_t - x_{t+1}\|\sqrt{\frac{2H_t}{\beta_t + \mu}} \leq \frac{1}{4}\min\left\{H_t, \frac{H_t^2}{(L_{yy} + \beta_t)D_{\mathcal{Y}}^2}\right\} + \frac{2L_{yx}^2\|x_{t+1} - x_t\|^2}{\beta_t + \mu}$$

$$+ \frac{3}{4}\left(\frac{\sqrt{2}L_{yx}(L_{yy} + \beta_t)^{1/4}D_{\mathcal{Y}}^{1/2}\|x_{t+1} - x_t\|}{\sqrt{\beta_t + \mu}}\right)^{4/3}.$$

Further, we apply the AM-GM inequality for three numbers to the last term in the above inequality to get

$$\left(\frac{\sqrt{2}L_{yx}(L_{yy} + \beta_t)^{1/4}D_{\mathcal{Y}}^{1/2}\|x_{t+1} - x_t\|}{\sqrt{\beta_t + \mu}}\right)^{4/3} = \left(\left(\frac{\|x_{t+1} - x_t\|}{\sqrt{2\tau_t}}\right)\right)^{4/3}\left(\frac{2L_{yx}(L_{yy} + \beta_t)^{1/4}\sqrt{D_{\mathcal{Y}}\tau_t}}{\sqrt{\beta_t + \mu}}\right)^{4/3}$$

$$= \left(\left(\frac{\|x_{t+1} - x_t\|}{\sqrt{2\tau_t}}\right)^{2/3}\right)^2\left(\frac{2L_{yx}(L_{yy} + \beta_t)^{1/4}\sqrt{D_{\mathcal{Y}}\tau_t}}{\sqrt{\beta_t + \mu}}\right)^{4/3}$$

$$\leq \frac{2}{3}\frac{\|x_{t+1} - x_t\|^2}{2\tau_t} + \frac{1}{3}\frac{16L_{yx}^4(L_{yy} + \beta_t)D_{\mathcal{Y}}^2\tau_t^2}{(\beta_t + \mu)^2},$$

which implies

$$L_{yx}\|x_{t+1} - x_t\|\sqrt{\frac{2H_t}{\beta_t + \mu}} \leq \frac{1}{4}\min\left\{H_t, \frac{H_t^2}{(L_{yy} + \beta_t)D_{\mathcal{Y}}^2}\right\} + \left(\frac{1}{4\tau_t} + \frac{2L_{yx}^2}{\beta_t + \mu}\right)\|x_{t+1} - x_t\|^2$$

$$+ \frac{4L_{yx}^4(L_{yy} + \beta_t)D_{\mathcal{Y}}^2\tau_t^2}{(\beta_t + \mu)^2}.$$

Substituting this bound into (18) and using the definition of the primal smoothness modulus $L_{f_t} = L_{xx} + L_{yx}^2/(\beta_t + \mu)$ completes the proof. $\square$

Under the additional assumption of convexity from Assumption 1.4, we establish the convergence of the primal suboptimality gap in the following lemma.

**Lemma A.11.** *Under the assumptions of Lemma A.10 and the additional Assumption 1.4, let $\{\tau_t\}_{t\geq 0}$ be a stepsize sequence satisfying $0 < \tau_t < 1/L_{f_t}$. Then, for any $T \geq 0$,*

$$\frac{1}{T+1}\sum_{t=0}^T f(x_{t+1}) - f(x^\star) \leq \left(1 + \sup_{t\geq 0}\frac{L_{yx}^2\tau_t}{(1 - L_{f_t}\tau_t)(\beta_t + \mu)}\right)\frac{1}{T+1}\sum_{t=0}^T H_t + \frac{D_{\mathcal{X}}^2}{2(T+1)\tau_T} + \frac{D_{\mathcal{Y}}^2}{2(T+1)}\sum_{t=0}^T \beta_t.$$

*Proof.* Recalling that $H_t = f_t(x_t) - \mathcal{L}_t(x_t, y_t)$ and using the $L_{f_t}$-smoothness of $f_t$, we have

$$f_t(x_{t+1}) \leq f_t(x_t) + \nabla f_t(x_t)^\top(x_{t+1} - x_t) + \frac{L_{f_t}}{2}\|x_{t+1} - x_t\|^2$$

$$= H_t + \mathcal{L}_t(x_t, y_t) + \nabla f_t(x_t)^\top (x_{t+1} - x_t) + \frac{L_{f_t}}{2} \|x_{t+1} - x_t\|^2.$$

By the convexity of $\mathcal{L}_t(\cdot, y_t)$, we can link the current iterate to the optimizer $x^\star$ as follows

$$f_t(x_{t+1})$$

$$\leq H_t + \mathcal{L}_t(x^\star, y_t) + \nabla_x \mathcal{L}(x_t, y_t)^\top (x_t - x^\star) + \nabla f_t(x_t)^\top (x_{t+1} - x_t) + \frac{L_{f_t}}{2} \|x_{t+1} - x_t\|^2$$

$$= H_t + \mathcal{L}_t(x^\star, y_t) + \nabla_x \mathcal{L}(x_t, y_t)^\top (x_t - x_{t+1} + x_{t+1} - x^\star) + \nabla_x \mathcal{L}(x_t, y_t^\star(x_t))^\top (x_{t+1} - x_t) + \frac{L_{f_t}}{2} \|x_{t+1} - x_t\|^2$$

$$\leq H_t + f_t(x^\star) + \|\nabla_x \mathcal{L}(x_t, y_t^\star(x_t)) - \nabla_x \mathcal{L}(x_t, y_t)\| \cdot \|x_{t+1} - x_t\| + \nabla_x \mathcal{L}(x_t, y_t)^\top (x_{t+1} - x^\star) + \frac{L_{f_t}}{2} \|x_{t+1} - x_t\|^2,$$

where the last inequality follows from the definition of $f_t$ and Cauchy-Schwartz inequality. By Lemma A.1 (vi), we have $f_t(x) \leq f(x) \leq f_t(x) + \mathcal{D}_{\mathcal{Y}}^2 \beta_t / 2$, implying $f(x_{t+1}) - f(x^\star) - D_{\mathcal{Y}}^2 \beta_t / 2 \leq f_t(x_{t+1}) - f_t(x^\star)$. This yields

$$f(x_{t+1}) - f(x^\star) - \frac{D_{\mathcal{Y}}^2 \beta_t}{2}$$

$$\leq H_t + \|\nabla_x \mathcal{L}(x_t, y_t^\star(x_t)) - \nabla_x \mathcal{L}(x_t, y_t)\| \cdot \|x_{t+1} - x_t\| + \nabla_x \mathcal{L}(x_t, y_t)^\top (x_{t+1} - x^\star) + \frac{L_{f_t}}{2} \|x_{t+1} - x_t\|^2.$$

The first-order optimality condition for the projection step at $x_{t+1}$ implies

$$(x_{t+1} - x_t + \tau_t \nabla_x \mathcal{L}(x_t, y_t))^\top (x - x_{t+1}) \geq 0, \quad \forall x \in \mathcal{X},$$

which yields the standard three-point relation

$$\frac{1}{2}(\|x_t - x^\star\|^2 - \|x_t - x_{t+1}\|^2 - \|x_{t+1} - x^\star\|^2) = (x_{t+1} - x_t)^\top (x^\star - x_{t+1})$$

$$\geq \tau_t \nabla_x \mathcal{L}(x_t, y_t)^\top (x_{t+1} - x^\star).$$

Using this bound, we arrive at

$$f(x_{t+1}) - f(x^\star) - \frac{D_{\mathcal{Y}}^2 \beta_t}{2} \leq H_t + \|\nabla_x \mathcal{L}(x_t, y_t^\star(x_t)) - \nabla_x \mathcal{L}(x_t, y_t)\| \cdot \|x_{t+1} - x_t\|$$

$$+ \frac{1}{2\tau_t}(\|x_t - x^\star\|^2 - \|x_t - x_{t+1}\|^2 - \|x_{t+1} - x^\star\|^2) + \frac{L_{f_t}}{2} \|x_{t+1} - x_t\|^2.$$

Using this Lipschitz smooth of $\nabla_x \mathcal{L}$ and the strong concavity relation $\|y_t^\star(x_t) - y_t\| \leq \sqrt{2H_t/(\beta_t + \mu)}$, we have

$$f(x_{t+1}) - f(x^\star) - \frac{D_{\mathcal{Y}}^2 \beta_t}{2}$$

$$\leq H_t + L_{yx} \sqrt{\frac{2H_t}{\beta_t + \mu}} \|x_{t+1} - x_t\| - \left( \frac{1}{2\tau_t} - \frac{L_{f_t}}{2} \right) \|x_{t+1} - x_t\|^2 + \frac{\|x_t - x^\star\|^2 - \|x_{t+1} - x^\star\|^2}{2\tau_t}$$

$$\leq H_t + \frac{L_{yx}^2 \tau_t H_t}{(1 - L_{f_t} \tau_t)(\beta_t + \mu)} + \frac{\|x_t - x^\star\|^2 - \|x_{t+1} - x^\star\|^2}{2\tau_t},$$

where the last inequality follows from the fact that $ax - \frac{bx^2}{2} \leq \frac{a^2}{2b}$ for any $x, a, b > 0$. As a result,

$$\sum_{t=0}^{T} \left( f(x_{t+1}) - f(x^\star) - \frac{D_{\mathcal{Y}}^2 \beta_t}{2} \right) \leq \left( 1 + \sup_{t \geq 0} \frac{L_{yx}^2 \tau_t}{(1 - L_{f_t} \tau_t)(\beta_t + \mu)} \right) \sum_{t=0}^{T} H_t + \sum_{t=0}^{T} \frac{\|x_t - x^\star\|^2 - \|x_{t+1} - x^\star\|^2}{2\tau_t}.$$

Observing that

$$\sum_{t=0}^{T} \frac{\|x_t - x^\star\|^2 - \|x_{t+1} - x^\star\|^2}{2\tau_t} = \frac{\|x_0 - x^\star\|^2}{2\tau_0} + \sum_{t \in [T]} \left( \frac{1}{2\tau_t} - \frac{1}{2\tau_{t-1}} \right) \|x_t - x^\star\|^2 - \frac{\|x_{T+1} - x^\star\|^2}{2\tau_T}$$

$$\leq \frac{D_{\mathcal{X}}^2}{2\tau_0} + \sum_{t=1}^{T} \left( \frac{1}{2\tau_t} - \frac{1}{2\tau_{t-1}} \right) D_{\mathcal{X}}^2$$

$$= \frac{D_{\mathcal{X}}^2}{2\tau_T}$$

together with a simple rearrangement conclude the proof. $\qquad \square$

### A.6. Dual PO sequence

We conclude this section by analyzing the dual PO update rule, for which the stepsize is adaptive and follows a closed-loop strategy. The following lemma provides a standard result established by viewing the update for $y_{t+1}$ as a step of the proximal gradient method on the $(\beta_t + \mu)$-strongly convex function $-\mathcal{L}_t(x_t, \cdot)$, which has a Lipschitz gradient modulus $L_{yy} + \beta_t$ and a unique minimizer $y_t^\star(x_t)$.

**Lemma A.12.** *Under Assumption 1.1, let $\{x_t\}_{t \geq 0} \subseteq \mathcal{X}$ be any primal sequence, let $\{y_t\}_{t \geq 0}$ be dual PO sequence generated by the update rule* (12d), *and let $\{\beta_t\}_{t \geq 0}$ be a non-increasing sequence of smoothing parameters such that $\max\{\beta_t, \mu\} > 0$. Then, for any $t \geq 0$,*

$$\mathcal{L}_t(x_t, y_t^\star(x_t)) - \mathcal{L}_t(x_t, y_{t+1}) \leq \frac{L_{yy} - \mu}{2} \|y_t - y_t^\star(x_t)\|^2 - \frac{L_{yy} + \beta_t}{2} \|y_{t+1} - y_t^\star(x_t)\|^2.$$

*Proof.* By the $(L_{yy} + \beta_t)$-Lipschitz continuity of $\nabla_y \mathcal{L}_t(x_t, \cdot)$, we have

$$-\mathcal{L}_t(x_t, y_{t+1}) \leq -\mathcal{L}_t(x_t, y_t) - \nabla_y \mathcal{L}_t(x_t, y_t)^\top (y_{t+1} - y_t) + \frac{L_{yy} + \beta_t}{2} \|y_{t+1} - y_t\|^2.$$

By the $(\beta_t + \mu)$-strong concavity of $\mathcal{L}_t(x_t, \cdot)$, it holds that

$$\mathcal{L}_t(x_t, y_t^\star(x_t)) \leq \mathcal{L}_t(x_t, y_t) + \nabla_y \mathcal{L}_t(x_t, y_t)^\top (y_t^\star(x_t) - y_t) - \frac{\beta_t + \mu}{2} \|y_t^\star(x_t) - y_t\|^2.$$

Summing these two inequalities yields

$$
\begin{aligned}
&\mathcal{L}_t(x_t, y_t^\star(x_t)) - \mathcal{L}_t(x_t, y_{t+1}) \\
&\leq \nabla_y \mathcal{L}_t(x_t, y_t)^\top (y_t^\star(x_t) - y_{t+1}) + \frac{L_{yy} + \beta_t}{2} \|y_{t+1} - y_t\|^2 - \frac{\beta_t + \mu}{2} \|y_t^\star(x_t) - y_t\|^2.
\end{aligned}
\tag{19}
$$

By the definition of the PO update rule with stepsize $\gamma_t$, the following first-order optimality condition for the projection step at $y_{t+1}$ is satisfied

$$\left( y_{t+1} - y_t - \frac{1}{L_{yy} + \beta_t} \nabla_y \mathcal{L}_t(x_t, y_t) \right)^\top (y - y_{t+1}) \geq 0, \quad \forall y \in \mathcal{Y},$$

which implies

$$
\begin{aligned}
\frac{1}{2} \left( \|y_t - y_t^\star(x_t)\|^2 - \|y_{t+1} - y_t^\star(x_t)\|^2 - \|y_{t+1} - y_t\|^2 \right) &= (y_t - y_{t+1})^\top (y_{t+1} - y_t^\star(x_t)) \\
&\geq \frac{1}{L_{yy} + \beta_t} \nabla_y \mathcal{L}_t(x_t, y_t)^\top (y_t^\star(x_t) - y_{t+1}).
\end{aligned}
$$

Multiplying by $L_{yy} + \beta_t$ and substituting this bound into (19) will conclude the proof. $\qquad \square$

The following lemma establishes a recursive bound linking $H_t$ with the regularized payoff functions.

**Lemma A.13.** *Under the assumptions of Lemma A.12, for any $t \geq 0$,*

$$
\begin{aligned}
\mathcal{L}_t(x_t, y_t^\star(x_t)) - \mathcal{L}_t(x_t, y_{t+1}) &\leq \frac{L_{yy} - \mu}{2} \|y_t - y_t^\star(x_t)\|^2 - \frac{L_{yy} + \beta_t}{2} \|y_{t+1} - y_{t+1}^\star(x_{t+1})\|^2 \\
&\quad + (L_{yy} + \beta_t) \left( \frac{L_{yx} \|x_{t+1} - x_t\| + 2D_{\mathcal{Y}}(\beta_t - \beta_{t+1})}{\beta_{t+1} + \mu} \right) \sqrt{\frac{2H_{t+1}}{\beta_{t+1} + \mu}}.
\end{aligned}
$$

*Proof.* Observing

$$\|y_{t+1} - y_t^\star(x_t)\|^2 = \|y_{t+1} - y_{t+1}^\star(x_{t+1}) + y_{t+1}^\star(x_{t+1}) - y_t^\star(x_t)\|^2$$
$$\geq \|y_{t+1} - y_{t+1}^\star(x_{t+1})\|^2 + 2(y_{t+1} - y_{t+1}^\star(x_{t+1}))^\top (y_{t+1}^\star(x_{t+1}) - y_t^\star(x_t))$$

together with Lemma A.12 imply that

$$\mathcal{L}_t(x_t, y_t^\star(x_t)) - \mathcal{L}_t(x_t, y_{t+1}) \leq \frac{L_{yy} - \mu}{2}\|y_t - y_t^\star(x_t)\|^2 - \frac{L_{yy} + \beta_t}{2}\|y_{t+1} - y_{t+1}^\star(x_{t+1})\|^2$$
$$- (L_{yy} + \beta_t)(y_{t+1} - y_{t+1}^\star(x_{t+1}))^\top (y_{t+1}^\star(x_{t+1}) - y_t^\star(x_t))$$
$$\leq \frac{L_{yy} - \mu}{2}\|y_t - y_t^\star(x_t)\|^2 - \frac{L_{yy} + \beta_t}{2}\|y_{t+1} - y_{t+1}^\star(x_{t+1})\|^2$$
$$+ (L_{yy} + \beta_t)\|y_{t+1} - y_{t+1}^\star(x_{t+1})\| \cdot \|y_{t+1}^\star(x_{t+1}) - y_t^\star(x_t)\|.$$

Observe next that

$$\|y_{t+1}^\star(x_{t+1}) - y_t^\star(x_t)\| \leq \|y_{t+1}^\star(x_t) - y_t^\star(x_t)\| + \|y_{t+1}^\star(x_{t+1}) - y_{t+1}^\star(x_t)\|$$
$$\leq \frac{2D_{\mathcal{Y}}(\beta_t - \beta_{t+1})}{\beta_{t+1} + \mu} + \frac{L_{yx}\|x_{t+1} - x_t\|}{\beta_{t+1} + \mu},$$

where the inequality follows from Lemma F.5 and Lemma A.1 (iii). We thus deduce that

$$\mathcal{L}_t(x_t, y_t^\star(x_t)) - \mathcal{L}_t(x_t, y_{t+1}) \leq \frac{L_{yy} - \mu}{2}\|y_t - y_t^\star(x_t)\|^2 - \frac{L_{yy} + \beta_t}{2}\|y_{t+1} - y_{t+1}^\star(x_{t+1})\|^2$$
$$+ (L_{yy} + \beta_t)(\|y_{t+1} - y_{t+1}^\star(x_{t+1})\|) \left( \frac{L_{yx}\|x_{t+1} - x_t\| + 2D_{\mathcal{Y}}(\beta_t - \beta_{t+1})}{\beta_{t+1} + \mu} \right).$$

Applying Lemma A.1 (i) will conclude the proof. $\qquad\square$

## B. Proofs of Section 1

We provide the formal proofs for the relationships between the stationarity and sub-optimality metrics discussed in Section 1.2.1. The following result restates Lemma 1.3 from the main text in a more comprehensive form, detailing both directions of the inequality between the LMO and PO gap functions.

**Lemma B.1.** *If Assumption 1.1 holds then for any $(x, y) \in \mathcal{X} \times \mathcal{Y}$ and $\sigma > 0$, it holds that*

$$G_{\mathcal{X}}^{\text{LMO}}(x, y) \leq (\sigma\|\nabla_x\mathcal{L}(x, y)\| + D_{\mathcal{X}})\, G_{\mathcal{X}}^{\text{PO}}(x, y),$$

$$G_{\mathcal{X}}^{\text{PO}}(x, y) \leq \sqrt{\frac{G_{\mathcal{X}}^{\text{LMO}}(x, y)}{\sigma}}.$$

*Proof.* We begin with the first inequality. Let the projected point be denoted as

$$\bar{x} := \text{PO}_{\mathcal{X}}\left(x - \sigma\nabla_x\mathcal{L}(x, y)\right).$$

By the definition of the primal PO stationarity measure, we have $\|\bar{x} - x\| = \sigma G_{\mathcal{X}}^{\text{PO}}(x, y)$. From the first-order optimality condition of the projection onto the convex set $\mathcal{X}$, for any $v \in \mathcal{X}$ it holds that

$$(\bar{x} - x + \sigma\nabla_x\mathcal{L}(x, y))^\top (v - \bar{x}) \geq 0 \quad \Longleftrightarrow \quad \sigma\nabla_x\mathcal{L}(x, y)^\top (\bar{x} - v) \leq (\bar{x} - x)^\top (v - \bar{x}). \tag{20}$$

Therefore, we may conclude that

$$\sigma\nabla_x\mathcal{L}(x, y)^\top (x - v) = \sigma\nabla_x\mathcal{L}(x, y)^\top (x - \bar{x}) + \sigma\nabla_x\mathcal{L}(x, y)^\top (\bar{x} - v)$$
$$\leq \sigma\nabla_x\mathcal{L}(x, y)^\top (x - \bar{x}) + (\bar{x} - x)^\top (v - \bar{x})$$
$$\leq \sigma\|\nabla_x\mathcal{L}(x, y)\| \cdot \|x - \bar{x}\| + \|v - \bar{x}\| \cdot \|\bar{x} - x\|$$

$$\leq \left(\sigma\|\nabla_x\mathcal{L}(x,y)\| + D_{\mathcal{X}}\right)\sigma G_{\mathcal{X}}^{\mathrm{PO}}(x,y),$$

where the first inequality follows from (20), the second inequality follows from the Cauchy-Schwartz inequality, and last inequality follows from the compactness of $\mathcal{X}$. Dividing both sides by $\sigma$ and taking the maximum over $v \in \mathcal{X}$ yields the first claimed inequality.

To establish the second inequality, we set $v = x$ in the optimality condition (20), which results in

$$\sigma\nabla_x\mathcal{L}(x,y)^\top(\bar{x}-x) \leq (\bar{x}-x)^\top(x-\bar{x}) \quad \Longleftrightarrow \quad \|\bar{x}-x\|^2 \leq \nabla_x\mathcal{L}(x,y)^\top(x-\bar{x}).$$

Given that $\bar{x} \in \mathcal{X}$ the term on the right-hand side is bounded by the LMO gap, $\nabla_x\mathcal{L}(x,y)^\top(x-\bar{x}) \leq G_{\mathcal{X}}^{\mathrm{LMO}}(x,y)$. Recalling that $G_{\mathcal{X}}^{\mathrm{PO}}(x,y) = \|\bar{x}-x\|/\sigma$, we deduce

$$(\sigma G_{\mathcal{X}}^{\mathrm{PO}}(x,y))^2 \leq \sigma G_{\mathcal{X}}^{\mathrm{LMO}}(x,y) \iff G_{\mathcal{X}}^{\mathrm{PO}}(x,y) \leq \sqrt{\frac{G_{\mathcal{X}}^{\mathrm{LMO}}(x,y)}{\sigma}},$$

which completes the proof. $\qquad\square$

We now provide the proof for Lemma 1.5, which relies on the convexity-concavity of the payoff function $L$ and the application of Jensen's inequality to the average iterates.

**Lemma B.2.** *Suppose $\{x_t, y_t\}_{t=1}^T$ is any sequence in $\mathcal{X} \times \mathcal{Y}$. If Assumptions 1.1 and 1.4 hold, then we have*

$$G_{\mathcal{X},\mathcal{Y}}^{\mathrm{Dual}}(\bar{x}_T, \bar{y}_T) \leq \frac{1}{T}\sum_{t=1}^T \left(G_{\mathcal{X}}^{\mathrm{LMO}}(x_t, y_t) + G_{\mathcal{Y}}(x_t, y_t)\right),$$

*where $\bar{x}_T := \frac{1}{T}\sum_{t=1}^T x_t$ and $\bar{y}_T := \frac{1}{T}\sum_{t=1}^T y_t$ denote the average iterates.*

*Proof.* For any arbitrary pair $(x,y) \in \mathcal{X} \times \mathcal{Y}$, we have

$$\begin{aligned}
\sum_{t\in[T]} \left(G_{\mathcal{X}}^{\mathrm{LMO}}(x_t,y_t) + G_{\mathcal{Y}}(x_t,y_t)\right) &\geq \sum_{t\in[T]} \left(\nabla_x\mathcal{L}(x_t,y_t)^\top(x_t-x) + \mathcal{L}(x_t,y) - \mathcal{L}(x_t,y_t)\right) \\
&\geq \sum_{t\in[T]} \left(\mathcal{L}(x_t,y_t) - \mathcal{L}(x,y_t) + \mathcal{L}(x_t,y) - \mathcal{L}(x_t,y_t)\right) \\
&= \sum_{t\in[T]} \mathcal{L}(x_t,y) - \sum_{t\in[T]} \mathcal{L}(x,y_t) \\
&\geq T\left(\mathcal{L}(\bar{x}_T,y) - \mathcal{L}(x,\bar{y}_T)\right).
\end{aligned}$$

where the first inequality follows from the definition of the LMO and dual gaps, the second follows from the first-order characterization of convexity for $\mathcal{L}(\cdot, y_t)$, and the final inequality results from the application of Jensen's inequality to the convex function $\mathcal{L}(\cdot, y)$ and the concave function $\mathcal{L}(x, \cdot)$. Dividing both sides by $T$ and taking the supremum over $y \in \mathcal{Y}$ and the infimum over $x \in \mathcal{X}$ yields

$$\max_{y\in\mathcal{Y}}\mathcal{L}(\bar{x}_T,y) - \min_{x\in\mathcal{X}}\mathcal{L}(x,\bar{y}_T) \leq \frac{1}{T}\sum_{t=1}^T \left(G_{\mathcal{X}}^{\mathrm{LMO}}(x_t,y_t) + G_{\mathcal{Y}}(x_t,y_t)\right).$$

Recalling the definition of $G_{\mathcal{X},\mathcal{Y}}^{\mathrm{Dual}}(\bar{x}_T, \bar{y}_T)$, we conclude the proof. $\qquad\square$

*Remark* B.3 (Relationship to Game Stationarity and Oracle Metrics). To provide a unified perspective on the convergence of our algorithms, we relate our stationarity measures to the standard notion of $\epsilon$-*game stationarity*. A point $(\bar{x}, \bar{y}) \in \mathcal{X} \times \mathcal{Y}$ is considered $\epsilon$-game stationary for the minimax problem (1) if it satisfies the following inclusion relations

$$-\nabla_x\mathcal{L}(\bar{x},\bar{y}) \in \mathcal{N}_{\mathcal{X}}(\bar{x}) + \mathcal{B}(\epsilon) \quad \text{and} \quad \nabla_y\mathcal{L}(\bar{x},\bar{y}) \in \mathcal{N}_{\mathcal{Y}}(\bar{y}) + \mathcal{B}(\epsilon), \tag{21}$$

where $\mathcal{N}_{\mathcal{S}}(s)$ denotes the normal cone to the set $\mathcal{S}$ at point $s$, and $\mathcal{B}(\epsilon)$ represents the Euclidean ball of radius $O(\sqrt{\epsilon})$. The justification for the gap functions used in this work is that an $\epsilon$-solution in terms of $G_{\mathcal{X}}^{\mathrm{LMO}}/G_{\mathcal{X}}^{\mathrm{PO}}$ and $G_{\mathcal{Y}}$ implies an $\epsilon$-game stationary solution. This relationship is justified as follows.

For the dual variable $y$, the condition $G_{\mathcal{Y}}(\bar{x}, \bar{y}) \leq \epsilon$ implies that $(\bar{x}, \bar{y})$ is an $\epsilon$-*approximate Nash equilibrium* for the $y$-player. This means the player cannot improve the payoff by more than $\epsilon$ by deviating from $\bar{y}$. Furthermore, in the strongly concave settings ($\mu > 0$), a bound on $G_{\mathcal{Y}}$ directly implies the dual gradient inclusion in (21) via standard strong concavity inequalities. For the primal variable $x$, the stationarity requirement (21) is satisfied if $G_{\mathcal{X}}^{\mathrm{LMO}} \leq \epsilon$ as established in the projection-free literature (Boroun et al., 2023). Crucially, for our hybrid algorithms that utilize a projection oracle, Lemma 1.3 provides the necessary link. Specifically, the inequality $G_{\mathcal{X}}^{\mathrm{LMO}} \leq (\sigma \|\nabla_x L\| + D_{\mathcal{X}}) G_{\mathcal{X}}^{\mathrm{PO}}$ ensures that a bound on the primal PO gap automatically establishes a bound on the LMO gap up to a constant factor. Therefore, an $\epsilon$-convergence rate in the PO gap implies the primal inclusion in (21). Consequently, our unified analysis across all oracle configurations provides a rigorous guarantee for reaching an $\epsilon$-game stationary solution.

## C. Proofs of Section 2

In this section we provide the detailed convergence analysis for Algorithm 1. While the algorithm shares its fundamental structure with R-PDCG (Boroun et al., 2023, Algorithm 1), our analysis shows that the choice of novel parameter settings leads to stronger convergence guarantees. Specifically, we adopt the following *anytime* schedules for the stepsize and regularization parameters.

*Condition* C.1. The sequences $\{\tau_t, \beta_t\}_{t \geq 0}$ are chosen such that

$$\tau_t = (t+1)^{-a}, \quad \beta_t = C(t+1)^{-b},$$

for constants $a, b \in (0, 1]$ and $C \geq 0$ satisfying $\max\{C, \mu\} > 0$, where $\mu$ is the strong concavity modulus from Assumption 1.1 (ii).

The requirement for $C$ ensures that the framework remains flexible. If $\mathcal{L}(x, \cdot)$ is already $\mu$-strongly concave for all $x \in \mathcal{X}$ (i.e., Assumption 1.1 (ii) holds with $\mu > 0$), explicit smoothing is unnecessary, and we may set $C = 0$. By leveraging the recursive relationships established in Lemma A.5 and Lemma A.8, we derive a technical recursion for the discrepancy sequence $\{H_t\}_{t \geq 0}$. The following result establishes an explicit *non-asymptotic decay bound* for this sequence in terms of the iteration index $t$.

**Lemma C.2.** *Let $\{x_t, y_t\}_{t \geq 0}$ be the sequence generated by Algorithm 1 under parameters $\{\tau_t, \beta_t\}_{t \geq 0}$ satisfying Condition C.1 with $2a > b$. Under Assumption 1.1, there exists a constant $M_1 > 0$ such that for any $t \geq 0$*

$$H_t \leq \frac{M_1}{(t+1)^{(2a-b)/3}}.$$

*If, in addition, $\mu > 0$ and $C = 0$, there exists a constant $M_2 > 0$ such that for any $t \geq 0$*

$$H_t \leq M_2 \cdot \begin{cases} (t+1)^{-2a/3} & \text{if } a \in (0, 1), \\ \dfrac{\log(t+2)}{(t+1)^{2/3}} & \text{if } a = 1. \end{cases}$$

*The constants $M_1$ and $M_2$ depend only on the parameters $\{L_{xx}, L_{yx}, L_{yy}, C, \mu, D_{\mathcal{X}}, D_{\mathcal{Y}}, a, b\}$ and the initial iterates $\{H_0, \ldots, H_4\}$.*

*Proof.* From the Lipschitz continuity of $\nabla_y \mathcal{L}_t$, we have

$$-\mathcal{L}_t(x_t, y_{t+1}) \leq -\mathcal{L}_t(x_t, y_t) - \gamma_t \nabla_y \mathcal{L}_t(x_t, y_t)^\top (u_t - y_t) + \frac{L_{yy} + \beta_t}{2} \|u_t - y_t\|^2 \gamma_t^2.$$

Since the adaptive stepsize $\gamma_t$, defined in (12b), minimizes the right-hand side over the interval $[0, 1]$, the inequality holds for any arbitrary $\gamma \in [0, 1]$. By adding $\mathcal{L}_t(x_t, y_t^\star(x_t))$ to both sides and using the definition of $H_t$, we obtain

$$\mathcal{L}_t(x_t, y_t^\star(x_t)) - \mathcal{L}_t(x_t, y_{t+1}) \leq H_t - \gamma \nabla_y \mathcal{L}_t(x_t, y_t)^\top (u_t - y_t) + \frac{L_{yy} + \beta_t}{2} \|u_t - y_t\|^2 \gamma^2. \tag{22}$$

From the definition of $u_t$ as the dual LMO solution and the concavity of $\mathcal{L}_t(x_t, \cdot)$, we have

$$\nabla_y \mathcal{L}_t(x_t, y_t)^\top (u_t - y_t) \geq \nabla_y \mathcal{L}_t(x_t, y_t)^\top (y_t^\star(x_t) - y_t) \geq \mathcal{L}_t(x_t, y_t^\star(x_t)) - \mathcal{L}_t(x_t, y_t) = H_t \geq 0.$$

Using this observation, the fact that $\|u_t - y_t\| \leq D_{\mathcal{Y}}$ and the inequality (22), we deduce

$$\mathcal{L}_t(x_t, y_t^\star(x_t)) - \mathcal{L}_t(x_t, y_{t+1}) \leq (1-\gamma)H_t + \frac{(L_{yy} + \beta_t)D_{\mathcal{Y}}^2}{2}\gamma^2.$$

Thanks to Lemma A.5 and the above inequality, we arrive at the following recursion

$$H_{t+1} - \frac{\sqrt{2}\left(L_{yx}D_{\mathcal{X}}\tau_t + D_{\mathcal{Y}}(\beta_t - \beta_{t+1})\right)}{\sqrt{\beta_{t+1} + \mu}}\sqrt{H_{t+1}} \leq (1-\gamma)H_t + \frac{(L_{yy} + \beta_t)D^2}{2}\gamma^2 + 2L_{xx}D_{\mathcal{X}}^2\tau_t^2.$$

By applying the AM-GM inequality to the cross term involving $\sqrt{H_{t+1}}$, we obtain

$$\frac{\sqrt{2}\left(L_{yx}D_{\mathcal{X}}\tau_t + D_{\mathcal{Y}}(\beta_t - \beta_{t+1})\right)}{\sqrt{\beta_{t+1} + \mu}}\sqrt{H_{t+1}} = \frac{\sqrt{2}\left(L_{yx}D_{\mathcal{X}}\tau_t + D_{\mathcal{Y}}(\beta_t - \beta_{t+1})\right)}{\sqrt{\gamma(\beta_{t+1} + \mu)}}\sqrt{\gamma H_{t+1}}$$

$$\leq \frac{\gamma}{2}H_{t+1} + \frac{\left(L_{yx}D_{\mathcal{X}}\tau_t + D_{\mathcal{Y}}(\beta_t - \beta_{t+1})\right)^2}{\gamma(\beta_{t+1} + \mu)}. \tag{23}$$

Combining these expressions and multiplying the last term in the above inequality by two (to avoid introducing irrational constants in subsequent bounds), we arrive at the recursion

$$\left(1 - \frac{\gamma}{2}\right)H_{t+1} \leq (1-\gamma)H_t + \frac{(L_{yy} + \beta_t)D_{\mathcal{Y}}^2}{2}\gamma^2 + 2L_{xx}D_{\mathcal{X}}^2\tau_t^2 + \frac{2\left(L_{yx}D_{\mathcal{X}}\tau_t + D_{\mathcal{Y}}(\beta_t - \beta_{t+1})\right)^2}{\gamma(\beta_{t+1} + \mu)}.$$

Restricting $\gamma \in (0, 1]$, we have $1 - \frac{\gamma}{2} \in \left[\frac{1}{2}, 1\right)$ and $\frac{1-\gamma}{1-\gamma/2} = 1 - \frac{\gamma}{2-\gamma} \leq 1 - \frac{\gamma}{2}$. Hence, we have

$$H_{t+1} \leq \left(1 - \frac{\gamma}{2}\right)H_t + (L_{yy} + \beta_t)D_{\mathcal{Y}}^2\gamma^2 + 4L_{xx}D_{\mathcal{X}}^2\tau_t^2 + \frac{4\left(L_{yx}D_{\mathcal{X}}\tau_t + D_{\mathcal{Y}}(\beta_t - \beta_{t+1})\right)^2}{\gamma(\beta_{t+1} + \mu)},$$

which implies

$$\frac{H_{t+1}}{\left(1 - \frac{\gamma}{2}\right)^{t+1}} \leq \frac{H_t}{\left(1 - \frac{\gamma}{2}\right)^t} + \frac{1}{\left(1 - \frac{\gamma}{2}\right)^{t+1}}\left((L_{yy} + \beta_t)D_{\mathcal{Y}}^2\gamma^2 + 4L_{xx}D_{\mathcal{X}}^2\tau_t^2 + \frac{4\left(L_{yx}D_{\mathcal{X}}\tau_t + D_{\mathcal{Y}}(\beta_t - \beta_{t+1})\right)^2}{\gamma(\beta_{t+1} + \mu)}\right).$$

Using this recursively, for any $t \geq 1$, we obtain

$$\frac{H_t}{\left(1 - \frac{\gamma}{2}\right)^t} \leq H_0 + \sum_{i \in [t]}\frac{1}{\left(1 - \frac{\gamma}{2}\right)^i}\left((L_{yy} + \beta_{i-1})D_{\mathcal{Y}}^2\gamma^2 + 4L_{xx}D_{\mathcal{X}}^2\tau_{i-1}^2 + \frac{4\left(L_{yx}D_{\mathcal{X}}\tau_{i-1} + D_{\mathcal{Y}}(\beta_{i-1} - \beta_i)\right)^2}{\gamma(\beta_i + \mu)}\right). \tag{24}$$

Using the requirements in Condition C.1, we observe that

$$4L_{xx}D_{\mathcal{X}}^2\tau_{i-1}^2 = \frac{4L_{xx}\max\{C, \mu\}D_{\mathcal{X}}^2}{\max\{C, \mu\}i^{2a}} \leq \frac{8L_{xx}\max\{C, \mu\}D_{\mathcal{X}}^2}{Ci^{2a} + \mu i^{2a}} \leq \frac{8L_{xx}\max\{C, \mu\}D_{\mathcal{X}}^2}{Ci^{2a-b} + \mu i^{2a}}.$$

Moreover, we also have

$$\frac{4\left(L_{yx}D_{\mathcal{X}}\tau_{i-1} + D_{\mathcal{Y}}(\beta_{i-1} - \beta_i)\right)^2}{\gamma(\beta_i + \mu)} = \frac{4\left(L_{yx}D_{\mathcal{X}}i^{-a} + CD_{\mathcal{Y}}(i^{-b} - (i+1)^{-b})\right)^2}{\gamma(Ci^{-b} + \mu)}$$

$$\leq \frac{8(L_{yx}D_{\mathcal{X}})^2i^{-2a} + 8(CD_{\mathcal{Y}})^2(i^{-b} - (i+1)^{-b})^2}{\gamma(Ci^{-b} + \mu)}$$

$$\leq \frac{8(L_{yx}D_{\mathcal{X}})^2i^{-2a} + 8b^2(CD_{\mathcal{Y}})^2i^{-2(b+1)}}{\gamma(Ci^{-b} + \mu)}$$

$$\leq \frac{8\left((L_{yx}D_{\mathcal{X}})^2 + b^2(CD_{\mathcal{Y}})^2\right)i^{-2a}}{\gamma(Ci^{-b} + \mu)}, \tag{25}$$

where the first inequality follows from $(x + y)^2 \leq 2(x^2 + y^2)$, the second inequality follows from Lemma F.2, and the last inequality follows as $b + 1 > 1 \geq a$. Thus, we can further upper bound the last term in (24) as follows

$$
(L_{yy} + \beta_t) D_{\mathcal{Y}}^2 \gamma^2 + 4L_{xx} D_{\mathcal{X}}^2 \tau_t^2 + \frac{4 (L_{yx} D_{\mathcal{X}} \tau_t + D_{\mathcal{Y}}(\beta_t - \beta_{t+1}))^2}{\gamma(\beta_{t+1} + \mu)}
$$
$$
\leq (L_{yy} + C) D_{\mathcal{Y}}^2 \gamma^2 + \frac{8L_{xx} \max\{C, \mu\} D_{\mathcal{X}}^2}{Ci^{2a-b} + \mu i^{2a}} + \frac{8 ((L_{yx} D_{\mathcal{X}})^2 + b^2 (CD_{\mathcal{Y}})^2) i^{-2a}}{\gamma(Ci^{-b} + \mu)}
$$
$$
\leq (L_{yy} + C) D_{\mathcal{Y}}^2 \gamma^2 + \frac{8 (L_{xx} \max\{C, \mu\} D_{\mathcal{X}}^2 + (L_{yx} D_{\mathcal{X}})^2 + b^2 (CD_{\mathcal{Y}})^2)}{\gamma(Ci^{2a-b} + \mu i^{2a})}.
$$

Using the fact that

$$
\sum_{i \in [t]} \frac{1}{(1 - \frac{\gamma}{2})^i} = \frac{1}{(1 - \frac{\gamma}{2})} \frac{\frac{1}{(1-\frac{\gamma}{2})^t} - 1}{\frac{1}{(1-\frac{\gamma}{2})} - 1} = \frac{2}{\gamma} \left( \frac{1}{(1 - \frac{\gamma}{2})^t} - 1 \right) \leq \frac{2}{\gamma (1 - \frac{\gamma}{2})^t}
$$

and the recursion (24), we have

$$
\frac{H_t}{(1 - \frac{\gamma}{2})^t} \leq H_0 + \frac{2 (L_{yy} + C) D_{\mathcal{Y}}^2 \gamma}{(1 - \frac{\gamma}{2})^t} + \sum_{i \in [t]} \frac{8 (L_{xx} \max\{C, \mu\} D_{\mathcal{X}}^2 + (L_{yx} D_{\mathcal{X}})^2 + b^2 (CD_{\mathcal{Y}})^2)}{\gamma (1 - \frac{\gamma}{2})^i (Ci^{2a-b} + \mu i^{2a})}.
$$

Using Cauchy-Schwartz inequality, one can easily show that

$$
\left( \frac{C}{i^{2a-b}} + \frac{\mu}{i^{2a}} \right) (Ci^{2a-b} + \mu i^{2a}) \geq (C + \mu)^2 \implies \frac{(C + \mu)^2}{Ci^{2a-b} + \mu i^{2a}} \leq \left( \frac{C}{i^{2a-b}} + \frac{\mu}{i^{2a}} \right). \tag{26}
$$

Thus, the following recursion holds

$$
H_t \leq H_0 \left( 1 - \frac{\gamma}{2} \right)^t + 2 (L_{yy} + C) D_{\mathcal{Y}}^2 \gamma
$$
$$
+ \frac{1}{\gamma} \left( 1 - \frac{\gamma}{2} \right)^t \sum_{i \in [t]} \frac{8 (L_{xx} \max\{C, \mu\} D_{\mathcal{X}}^2 + (L_{yx} D_{\mathcal{X}})^2 + b^2 (CD_{\mathcal{Y}})^2)}{(C + \mu)^2 (1 - \frac{\gamma}{2})^i} \left( \frac{C}{i^{2a-b}} + \frac{\mu}{i^{2a}} \right).
$$

Applying Lemma F.4, for any $t \geq 1$ such that $t - \lfloor \frac{2}{\gamma} \rfloor \geq 0$, we obtain the following recursion

$$
H_t \leq H_0 \left( 1 - \frac{\gamma}{2} \right)^t + 2 (L_{yy} + C) D_{\mathcal{Y}}^2 \gamma
$$
$$
+ \frac{16 (L_{xx} \max\{C, \mu\} D_{\mathcal{X}}^2 + (L_{yx} D_{\mathcal{X}})^2 + b^2 (CD_{\mathcal{Y}})^2)}{(C + \mu)\gamma^2} \left( 1 - \frac{\gamma}{2} \right)^{t - \lfloor \frac{2}{\gamma} \rfloor + 1}
$$
$$
+ \frac{64 (L_{xx} \max\{C, \mu\} D_{\mathcal{X}}^2 + (L_{yx} D_{\mathcal{X}})^2 + b^2 (CD_{\mathcal{Y}})^2)}{(C + \mu)^2 \gamma^2} \left( \frac{CE(a, b))}{t^{2a-b}} + \mu \begin{cases} \frac{E(a,0)}{t^{2a}} & 0 < a < 1 \\ \frac{\log(t)}{t^2} & a = 1 \end{cases} \right),
$$

where $E(a, b)$ is defined in Lemma F.4. Observe that

$$
\left( 1 - \frac{\gamma}{2} \right)^t \leq \exp \left( -\frac{\gamma t}{2} \right) \quad \text{and} \quad \left( 1 - \frac{\gamma}{2} \right)^{t - \lfloor \frac{2}{\gamma} \rfloor + 1} \leq \left( 1 - \frac{\gamma}{2} \right)^{t - \frac{2}{\gamma}} \leq \exp \left( -\frac{\gamma t}{2} + 1 \right), \tag{27}
$$

where in the last inequality, we use the definition of floor function. If $\mu = 0$, setting $\gamma = t^{-(2a-b)/3}$ yields

$$
H_t \leq \left( H_0 + \frac{16 (L_{xx} CD_{\mathcal{X}}^2 + (L_{yx} D_{\mathcal{X}})^2 + b^2 (CD_{\mathcal{Y}})^2) \exp(1)}{C} t^{(4a-2b)/3} \right) \exp \left( -\frac{t^{(3-2a+b)/3}}{2} \right)
$$
$$
+ \frac{2C (L_{yy} + C) D^2 + 64E(a, b) (L_{xx} CD_{\mathcal{X}}^2 + (L_{yx} D_{\mathcal{X}})^2 + b^2 (CD_{\mathcal{Y}})^2)}{Ct^{(2a-b)/3}},
$$

On the other hand, if $\mu > 0$, setting $\gamma = t^{-2a/3}$ yields

$$H_t \leq \left( H_0 + \frac{16 \left( L_{xx} \mu D_{\mathcal{X}}^2 + (L_{yx} D_{\mathcal{X}})^2 \right) e}{\mu} t^{4a/3} \right) \exp \left( -\frac{t^{(3-2a)/3}}{2} \right)$$
$$+ \frac{4 L_{yy} D^2}{t^{2a/3}} + \frac{16 \left( L_{xx} \mu D_{\mathcal{X}}^2 + (L_{yx} D_{\mathcal{X}})^2 \right)}{\mu} \begin{cases} \frac{E(a,0)}{t^{2a/3}}, & 0 < a < 1 \\ \frac{\log(t)}{t^{2/3}}, & a = 1. \end{cases}$$

Recall that applying Lemma F.4 requires $t - \left\lfloor \frac{2}{\gamma} \right\rfloor \geq 0$. With our choices of $\gamma$, this condition reduces to $t - \left\lfloor 2t^{2/3} \right\rfloor \geq 0$, which holds for all $t \geq 5$. Therefore, the existence of such $M_1$ and $M_2$ is guaranteed. This competes the proof. $\qquad \square$

The guarantees established in Lemma C.2 can be further refined when the maximization domain $\mathcal{Y}$ is strongly convex.

**Lemma C.3.** *Let $\{x_t, y_t\}_{t \geq 0}$ be the sequence generated by Algorithm 1 under parameters $\{\tau_t, \beta_t\}_{t \geq 0}$ satisfying Condition C.1 with $2a > b$. Under Assumptions 1.1 and 1.2, there exists a constant $M_3 > 0$ such that for any $t \geq 0$*

$$H_t \leq \frac{M_3}{(t+1)^{a-b}}.$$

*If, in addition, $\mu > 0$ and $C = 0$, there exists a constant $M_4 > 0$ such that for any $t \geq 0$*

$$H_t \leq M_4 \cdot \begin{cases} (t+1)^{-a} & \text{if } a \in (0,1), \\ \dfrac{\log(t+2)}{t+1} & \text{if } a = 1. \end{cases}$$

*The constants $M_3$ and $M_4$ depend only on the parameters $\{L_{xx}, L_{yx}, L_{yy}, C, \mu, D_{\mathcal{X}}, D_{\mathcal{Y}}, a, b\}$ and the initial iterates $\{H_0, \ldots, H_{13}\}$.*

*Proof.* Applying Lemma A.5 and Lemma A.9, we obtain the following recursion

$$H_{t+1} - \frac{\sqrt{2} \left( L_{yx} D_{\mathcal{X}} \tau_t + D_{\mathcal{Y}}(\beta_t - \beta_{t+1}) \right)}{\sqrt{\beta_{t+1} + \mu}} \sqrt{H_{t+1}} \leq \max \left\{ \frac{1}{2}, 1 - \frac{\alpha \sqrt{(\beta_t + \mu) H_t}}{8\sqrt{2} \left( L_{yy} + \beta_t \right)} \right\} H_t + 2 L_{xx} D_{\mathcal{X}}^2 \tau_t^2,$$

We first analyze the term involving the max operator. For any $\gamma > 0$, by the AM-GM inequality, we have

$$\frac{\alpha H_t \sqrt{(\beta_t + \mu) H_t}}{8\sqrt{2} \left( L_{yy} + \beta_t \right)} = \frac{\alpha H_t \sqrt{(\beta_t + \mu) H_t}}{16\sqrt{2} \left( L_{yy} + \beta_t \right)} + \frac{\alpha H_t \sqrt{\beta_t H_t}}{16\sqrt{2} \left( L_{yy} + \beta_t \right)} + \frac{512 (L_{yy} + \beta_t)^2 \gamma^3}{27 \alpha^2 (\beta_t + \mu)} - \frac{512 (L_{yy} + \beta_t)^2 \gamma^3}{27 \alpha^2 (\beta_t + \mu)}$$

$$\geq 3 \left( \frac{\alpha H_t \sqrt{(\beta_t + \mu) H_t}}{16\sqrt{2} \left( L_{yy} + \beta_t \right)} \cdot \frac{\alpha H_t \sqrt{(\beta_t + \mu) H_t}}{16\sqrt{2} \left( L_{yy} + \beta_t \right)} \cdot \frac{512 (L_{yy} + \beta_t)^2 \gamma^3}{27 \alpha^2 (\beta_t + \mu)} \right)^{1/3} - \frac{512 (L_{yy} + \beta_t)^2 \gamma^3}{27 \alpha^2 (\beta_t + \mu)}$$

$$= \gamma H_t - \frac{512 (L_{yy} + \beta_t)^2 \gamma^3}{27 \alpha^2 (\beta_t + \mu)}.$$

Thus, we have

$$H_t - \frac{\alpha H_t \sqrt{(\beta_t + \mu) H_t}}{8\sqrt{2} \left( L_{yy} + \beta_t \right)} \leq (1 - \gamma) H_t + \frac{512 (L_{yy} + \beta_t)^2 \gamma^3}{27 \alpha^2 (\beta_t + \mu)}.$$

By selecting $\gamma \leq 1/2$, the right-hand side is at least $H_t/2$, allowing us to simplify the maximum term. Applying the AM-GM inequality (23) (and multiplying the last term by two to avoid irrational coefficients later), we obtain

$$\left( 1 - \frac{\gamma}{2} \right) H_{t+1} \leq (1 - \gamma) H_t + \frac{512 (L_{yy} + \beta_t)^2 \gamma^3}{27 \alpha^2 (\beta_t + \mu)} + \frac{2 \left( L_{yx} D_{\mathcal{X}} \tau_t + D_{\mathcal{Y}}(\beta_t - \beta_{t+1}) \right)^2}{\gamma (\beta_{t+1} + \mu)} + 2 L_{xx} D_{\mathcal{X}}^2 \tau_t^2.$$

Since $\gamma \in (0,1)$, we have $1 - \frac{\gamma}{2} \in \left[ \frac{1}{2}, 1 \right)$ and $\frac{1-\gamma}{1-\gamma/2} = 1 - \frac{\gamma}{2-\gamma} \leq 1 - \frac{\gamma}{2}$. Hence, we have

$$H_{t+1} \leq \left( 1 - \frac{\gamma}{2} \right) H_t + \frac{1024 (L_{yy} + \beta_t)^2 \gamma^3}{27 \alpha^2 (\beta_t + \mu)} + \frac{4 \left( L_{yx} D_{\mathcal{X}} \tau_t + D_{\mathcal{Y}}(\beta_t - \beta_{t+1}) \right)^2}{\gamma (\beta_{t+1} + \mu)} + 4 L_{xx} D_{\mathcal{X}}^2 \tau_t^2$$

By dividing both sides with $\left(1 - \frac{\gamma}{2}\right)^{-(t+1)}$, we obtain

$$\left(1 - \frac{\gamma}{2}\right)^{-(t+1)} H_{t+1}$$

$$\leq \left(1 - \frac{\gamma}{2}\right)^{-t} H_t + \left(1 - \frac{\gamma}{2}\right)^{-(t+1)} \left(\frac{1024(L_{yy} + \beta_t)^2 \gamma^3}{27\alpha^2(\beta_t + \mu)} + \frac{4\left(L_{yx} D_{\mathcal{X}} \tau_t + D_{\mathcal{Y}}(\beta_t - \beta_{t+1})\right)^2}{\gamma(\beta_{t+1} + \mu)} + 4L_{xx} D_{\mathcal{X}}^2 \tau_t^2\right).$$

Using the requirements in Condition C.1, and the inequalities (25) and (26), for any $t \geq 1$, it follows that

$$\left(1 - \frac{\gamma}{2}\right)^{-t} H_t \leq \left(1 - \frac{\gamma}{2}\right)^{-(t-1)} H_{t-1} + \frac{1024(L_{yy} + C)^2 \gamma^3}{27\alpha^2(C + \mu)^2}\left(1 - \frac{\gamma}{2}\right)^{-t}\left(Ct^b + \mu\right)$$

$$+ \frac{8\left((L_{yx} D_{\mathcal{X}})^2 + b^2(CD_{\mathcal{Y}})^2\right)}{(C + \mu)^2 \gamma}\left(1 - \frac{\gamma}{2}\right)^{-t}\left(\frac{C}{t^{2a-b}} + \frac{\mu}{t^{2a}}\right) + \left(1 - \frac{\gamma}{2}\right)^{-t} \frac{4L_{xx} D_{\mathcal{X}}^2}{t^{2a}}$$

$$\leq H_0 + \frac{1024(L_{yy} + C)^2 \gamma^3}{27\alpha^2(C + \mu)^2}\sum_{i \in [t]}\left(1 - \frac{\gamma}{2}\right)^{-i}\left(Ci^b + \mu\right)$$

$$+ \frac{8\left((L_{yx} D_{\mathcal{X}})^2 + b^2(CD_{\mathcal{Y}})^2\right)}{(C + \mu)^2 \gamma}\sum_{i \in [t]}\left(1 - \frac{\gamma}{2}\right)^{-i}\left(\frac{C}{i^{2a-b}} + \frac{\mu}{i^{2a}}\right) + \sum_{i \in [t]}\left(1 - \frac{\gamma}{2}\right)^{-i} \frac{4L_{xx} D_{\mathcal{X}}^2}{i^{2a}}.$$

Since $b > 0$, we have

$$\sum_{i \in [t]}\left(1 - \frac{\gamma}{2}\right)^{-i}\left(Ci^b + \mu\right) \leq \sum_{i \in [t]}\left(1 - \frac{\gamma}{2}\right)^{-i}\left(Ct^b + \mu\right) \leq \left(1 - \frac{\gamma}{2}\right)^{-t} \frac{2\left(Ct^b + \mu\right)}{\gamma}.$$

Applying Lemma F.4, for any $t$ such that $t - \left\lfloor \frac{2}{\gamma} \right\rfloor \geq 0$, we have

$$H_t$$

$$\leq \left(1 - \frac{\gamma}{2}\right)^t H_0 + \frac{2048(L_{yy} + C)^2 \gamma^2}{27\alpha^2(C + \mu)^2}\left(Ct^b + \mu\right) + \left(\frac{16\left((L_{yx} D_{\mathcal{X}})^2 + b^2(CD_{\mathcal{Y}})^2\right)}{(C + \mu)\gamma^2} + \frac{8L_{xx} D_{\mathcal{X}}^2}{\gamma}\right)\left(1 - \frac{\gamma}{2}\right)^{t - \left\lfloor \frac{2}{\gamma} \right\rfloor + 1}$$

$$+ \frac{64\left((L_{yx} D_{\mathcal{X}})^2 + b^2(CD_{\mathcal{Y}})^2\right)}{(C + \mu)^2 \gamma^2}\left(\frac{C(2a - b)}{t^{2a-b}} + \mu \cdot \begin{cases} \frac{E(a,0)}{t^{2a}} & 0 < a < 1 \\ \frac{\log(t)}{t^2} & a = 1 \end{cases}\right)$$

$$+ \left(1 - \frac{\gamma}{2}\right)^{-t} \frac{32L_{xx} D_{\mathcal{X}}^2}{\gamma} \cdot \begin{cases} \frac{E(a,0)}{t^{2a}} & 0 < a < 1 \\ \frac{\log(t)}{t^2} & a = 1 \end{cases}.$$

Using the bounds in (27) and setting $\gamma = t^{-a/2}/2$ yield

$$H_t \leq \left(H_0 + \frac{64\left((L_{yx} D_{\mathcal{X}})^2 + b^2(CD_{\mathcal{Y}})^2\right)e}{C}t^a + 16eL_{xx} D_{\mathcal{X}}^2 t^{a/2} + 64L_{xx} D_{\mathcal{X}}^2 \cdot \begin{cases} \frac{E(a,0)}{t^{3a/2}} & 0 < a < 1 \\ \frac{\log(t)}{t^{3/2}} & a = 1 \end{cases}\right) \exp\left(-\frac{t^{1-a/2}}{4}\right)$$

$$+ \frac{512(L_{yy} + C)^2}{27\alpha^2 Ct^{a-b}} + \frac{256E(a,b)\left((L_{yx} D_{\mathcal{X}})^2 + b^2(CD_{\mathcal{Y}})^2\right)}{Ct^{a-b}}.$$

If Assumption 1.1 additionally holds with $\mu > 0$ and $C = 0$, then

$$H_t \leq \left(H_0 + \frac{64(L_{yx} D_{\mathcal{X}})^2 e}{\mu}t^a + 16eL_{xx} D_{\mathcal{X}}^2 t^{a/2} + 64L_{xx} D_{\mathcal{X}}^2 \cdot \begin{cases} \frac{E(a,0)}{t^{3a/2}} & 0 < a < 1 \\ \frac{\log(t)}{t^{3/2}} & a = 1 \end{cases}\right) \exp\left(-\frac{t^{1-a/2}}{2}\right)$$

$$+ \frac{512L_{yy}}{27\alpha^2 \mu t^a} + \frac{256(L_{yx} D_{\mathcal{X}})^2}{\mu}\begin{cases} \frac{E(a,0)}{t^a} & 0 < a < 1 \\ \frac{\log(t)}{t} & a = 1 \end{cases}.$$

Recall that applying Lemma F.4 requires $t - \left\lfloor \frac{2}{\gamma} \right\rfloor \geq 0$. With our choices of $\gamma$, this condition reduces to $t - \lfloor 4t^{1/2} \rfloor \geq 0$, which holds for all $t \geq 14$. Therefore, the existence of such $M_3$ and $M_4$ is guaranteed. This competes the proof. $\qquad \square$

With these refined bounds on the discrepancy sequence $\{H_t\}_{t \geq 0}$ established, we are now ready to derive the convergence guarantees for Algorithm 1 across the nonconvex and convex settings discussed in Section 2.

### C.1. Proof of Theorem 2.1

The proof relies on the following lemma, which establishes an upper bound on the accumulated LMO gap provided that the discrepancy sequence $\{H_t\}_{t \geq 0}$ exhibits a non-asymptotic decay rate.

**Lemma C.4.** *Suppose Assumption 1.1 holds. Let $\{x_t\}_{t \geq 0}$ be a primal LMO sequence with stepsize $\tau_t = (t+1)^{-a}$ generated by the update rule (12a), let $\{y_t\}_{t \geq 0} \subset \mathcal{Y}$ be any dual sequence, and let $\beta_t = C(t+1)^{-b}$. Suppose $2a > b$, $a, b \in (0, 1]$, $C \geq 0$, $\max\{C, \mu\} > 0$ and we have*

$$H_t \leq \frac{M_t}{(t+1)^c}, \quad \forall t \geq 0$$

*for some non-decreasing, positive sequence $\{M_t\}_{t \geq 0}$ and a constant $c$ satisfying $0 < c \leq 2a - b$ if $C > 0$ and $0 < c \leq 2a$ if $C = 0$ and $\mu > 0$. Then, for any $T \geq 1$,*

$$\sum_{t=0}^{T} G_{\mathcal{X}}^{\mathrm{LMO}}(x_t, y_t) \leq \left( 2 \max_{x \in \mathcal{X}} |f(x)| + CD_{\mathcal{Y}}^2 \right)(T+1)^a + \sum_{t=0}^{T} \frac{2L_{yx}D_{\mathcal{X}}\sqrt{2CM_t} + CL_{xx}D_{\mathcal{X}}^2 + CL_{yx}^2 D_{\mathcal{X}}^2/(C+\mu)}{2(C+\mu)(t+1)^{(c-b)/2}}$$

$$+ \sum_{t=0}^{T} \frac{2L_{yx}D_{\mathcal{X}}\sqrt{2\mu M_t} + \mu L_{xx}D_{\mathcal{X}}^2 + \frac{\mu L_{yx}^2 D_{\mathcal{X}}^2}{C+\mu}}{2(C+\mu)(t+1)^{c/2}}.$$

*Proof.* We bound the the last inequality in Lemma A.6. Observe that

$$L_{yx}D_{\mathcal{X}}\sqrt{\frac{2H_t}{\beta_t + \mu}} + \left( L_{xx}D_{\mathcal{X}}^2 + \frac{L_{yx}^2 D_{\mathcal{X}}^2}{\beta_t + \mu} \right)\frac{\tau_t}{2}$$

$$\leq L_{yx}D_{\mathcal{X}}\sqrt{\frac{2M_t}{C(t+1)^{c-b} + \mu(t+1)^{2a+c}}} + \frac{L_{xx}D_{\mathcal{X}}^2}{2(t+1)^a} + \frac{L_{yx}^2 D_{\mathcal{X}}^2}{2C(t+1)^{a-b} + 2\mu(t+1)^a}.$$

Using Cauchy-Schwartz inequality, we have

$$\left( \frac{C}{(t+1)^{c-b}} + \frac{\mu}{(t+1)^c} \right) \left( C(t+1)^{c-b} + \mu(t+1)^c \right) \geq (C+\mu)^2$$

$$\implies \frac{1}{C(t+1)^{c-b} + \mu(t+1)^c} \leq \frac{1}{(C+\mu)^2} \left( \frac{C}{(t+1)^{c-b}} + \frac{\mu}{(t+1)^c} \right),$$

and similarly,

$$\frac{1}{C(t+1)^{a-b} + \mu(t+1)^a} \leq \frac{1}{(C+\mu)^2} \left( \frac{C}{(t+1)^{a-b}} + \frac{\mu}{(t+1)^a} \right).$$

Thus, we can further bound the above inequality as follows

$$L_{yx}D_{\mathcal{X}}\sqrt{\frac{2H_t}{\beta_t + \mu}} + \left( L_{xx}D_{\mathcal{X}}^2 + \frac{L_{yx}^2 D_{\mathcal{X}}^2}{\beta_t + \mu} \right)\frac{\tau_t}{2}$$

$$\leq L_{yx}D_{\mathcal{X}}\sqrt{\frac{2M_t}{(C+\mu)^2}\left( \frac{C}{(t+1)^{c-b}} + \frac{\mu}{(t+1)^c} \right)} + \frac{L_{xx}D_{\mathcal{X}}^2}{2(t+1)^a} + \frac{L_{yx}^2 D_{\mathcal{X}}^2}{2(C+\mu)^2}\left( \frac{C}{(t+1)^{a-b}} + \frac{\mu}{(t+1)^a} \right)$$

$$\leq \frac{L_{yx}D_{\mathcal{X}}\sqrt{2CM_t}}{(C+\mu)(t+1)^{(c-b)/2}} + \frac{L_{yx}D_{\mathcal{X}}\sqrt{2\mu M_t}}{(C+\mu)(t+1)^{c/2}} + \frac{L_{xx}D_{\mathcal{X}}^2}{2(t+1)^a} + \frac{L_{yx}^2 D_{\mathcal{X}}^2}{2(C+\mu)^2}\left( \frac{C}{(t+1)^{a-b}} + \frac{\mu}{(t+1)^a} \right),$$

where the last inequality follows from the basic inequality $\sqrt{x+y} \leq \sqrt{x} + \sqrt{y}$. Using the requirement $0 < c \leq 2a - b$, we may conclude that

$$L_{yx}D_{\mathcal{X}}\sqrt{\frac{2H_t}{\beta_t + \mu}} + \left( L_{xx}D_{\mathcal{X}}^2 + \frac{L_{yx}^2 D_{\mathcal{X}}^2}{\beta_t + \mu} \right)\frac{\tau_t}{2}$$

$$\leq \frac{L_{yx}D_{\mathcal{X}}\sqrt{2CM_t} + \frac{CL_{yx}^2 D_{\mathcal{X}}^2}{C+\mu}}{(C+\mu)(t+1)^{(c-b)/2}} + \frac{L_{yx}D_{\mathcal{X}}\sqrt{2\mu M_t} + \frac{\mu L_{yx}^2 D_{\mathcal{X}}^2}{C+\mu}}{(C+\mu)(t+1)^{c/2}} + \frac{L_{xx}D_{\mathcal{X}}^2}{2(t+1)^a}$$

The claim follows by a simple rearrangement of the inequality above and applying Lemma A.6. $\qquad \square$

We now provide the detailed versions of the results summarized in Theorem 2.1, including explicit constants and optimal parameter configurations. We first address the settings without additional geometric assumptions on $\mathcal{Y}$.

**Theorem C.5** (Detailed version of Theorem 2.1 (i)–(ii)). *Suppose $\{x_t, y_t\}_{t\geq 0}$ is the sequence generated by Algorithm 1 under $\{\tau_t, \beta_t\}_{t\geq 0}$ satisfying Condition C.1 with $2a > b$, $a, b \in (0, 1)$. If Assumption 1.1 holds, then for any $T \geq 0$,*

$$\frac{\sum_{t=0}^{T} G_{\mathcal{X}}^{\mathrm{LMO}}(x_t, y_t)}{T+1} \leq \frac{2 \max_{x \in \mathcal{X}} |f(x)| + CD_{\mathcal{Y}}^2}{(T+1)^{1-a}} + \frac{1}{T+1} \sum_{t=0}^{T} \frac{2L_{yx}D_{\mathcal{X}}\sqrt{2CM_1} + CL_{xx}D_{\mathcal{X}}^2 + L_{yx}^2 D_{\mathcal{X}}^2}{2C(t+1)^{(a-2b)/3}},$$

$$G_{\mathcal{Y}}(x_T, y_T) \leq \frac{M_1}{(T+1)^{(2a-b)/3}} + \frac{CD_{\mathcal{Y}}^2}{2(T+1)^b}.$$

*By optimizing over $a, b$, we obtain the optimal rate of $O(T^{-1/6})$ when $a = \frac{5}{6}, b = \frac{1}{6}$. If Assumption 1.1 holds with $\mu > 0$ and $C = 0$, then for any $T \geq 0$,*

$$\frac{\sum_{t=0}^{T} G_{\mathcal{X}}^{\mathrm{LMO}}(x_t, y_t)}{T+1} \leq \frac{2 \max_{x \in \mathcal{X}} |f(x)|}{(T+1)^{1-a}} + \frac{1}{T+1} \sum_{t=0}^{T} \frac{2L_{yx}D_{\mathcal{X}}\sqrt{2\mu M_2} + \mu L_{xx}D_{\mathcal{X}}^2 + L_{yx}^2 D_{\mathcal{X}}^2}{2\mu(t+1)^{a/3}},$$

$$G_{\mathcal{Y}}(x_T, y_T) \leq \frac{M_2}{(T+1)^{2a/3}},$$

*By optimizing over $a$, we obtain the optimal rate of $O(T^{-1/4})$ when $a = \frac{3}{4}$.*

*Proof.* The bounds on $G_{\mathcal{X}}^{\mathrm{LMO}}$ follow from an application of Lemma C.4. Specifically, by Lemma C.2, the decay condition on $H_t$ required in Lemma C.4 is satisfied with constant $c = (2a - b)/3$, where we set $b = 0$ when $\mu > 0$. Moreover, the conditions on $c$ in Lemma C.4 are trivially satisfied since $(2a - b)/3 \leq 2a - b$ and $2a/3 \leq 2a$. On the other hand, the bounds on $G_{\mathcal{Y}}$ follow from applying Lemma A.6 together with Lemma C.2. This completes the proof. $\square$

Next, we provide the detailed bounds for settings where the dual domain $\mathcal{Y}$ is strongly convex, enabling improved convergence rates.

**Theorem C.6** (Detailed version of Theorem 2.1 (iii)–(iv)). *Suppose $\{x_t, y_t\}_{t\geq 0}$ is the sequence generated by Algorithm 1 under $\{\tau_t, \beta_t\}_{t\geq 0}$ satisfying Condition C.1 with $1 > a > b > 0$. If Assumption 1.1 and Assumption 1.2 hold, then for any $T \geq 0$,*

$$\frac{\sum_{t=0}^{T} G_{\mathcal{X}}^{\mathrm{LMO}}(x_t, y_t)}{T+1} \leq \frac{2 \max_{x \in \mathcal{X}} |f(x)| + CD_{\mathcal{Y}}^2}{(T+1)^{1-a}} + \frac{1}{T+1} \sum_{t=0}^{T} \frac{2L_{yx}D_{\mathcal{X}}\sqrt{2CM_3} + CL_{xx}D_{\mathcal{X}}^2 + L_{yx}^2 D_{\mathcal{X}}^2}{2C(t+1)^{(a-2b)/2}},$$

$$G_{\mathcal{Y}}(x_T, y_T) \leq \frac{M_3}{(T+1)^{a-b}} + \frac{CD_{\mathcal{Y}}^2}{2(T+1)^b}.$$

*By optimizing over $a, b$, we obtain the optimal rate of $O(T^{-1/5})$ when $a = \frac{4}{5}, b = \frac{1}{5}$. If Assumption 1.1 holds with $\mu > 0$ and $C = 0$, then for any $T \geq 0$,*

$$\frac{\sum_{t=0}^{T} G_{\mathcal{X}}^{\mathrm{LMO}}(x_t, y_t)}{T+1} \leq \frac{2 \max_{x \in \mathcal{X}} |f(x)|}{(T+1)^{1-a}} + \frac{1}{T+1} \sum_{t=0}^{T} \frac{2L_{yx}D_{\mathcal{X}}\sqrt{2\mu M_4} + \mu L_{xx}D_{\mathcal{X}}^2 + L_{yx}^2 D_{\mathcal{X}}^2}{2\mu(t+1)^{a/2}},$$

$$G_{\mathcal{Y}}(x_T, y_T) \leq \frac{M_4}{(T+1)^a},$$

*By optimizing over $a$, we obtain the optimal rate of $O(T^{-1/3})$ when $a = \frac{2}{3}$.*

*Proof.* The bounds on $G_{\mathcal{X}}^{\mathrm{LMO}}$ follow from an application of Lemma C.4. Specifically, by Lemma C.3, the decay condition on $H_t$ required in Lemma C.4 is satisfied with constant $c = a - b$, where we set $b = 0$ when $\mu > 0$. Moreover, the conditions on $c$ in Lemma C.4 are trivially satisfied since $a - b \leq 2a - b$ and $a \leq 2a$. On the other hand, the bounds on $G_{\mathcal{Y}}$ follow from applying Lemma A.6 together with Lemma C.3. This completes the proof. $\square$

### C.2. Proof of Theorem 2.2

The proof relies on the following lemma, which establishes an upper bound on the accumulated primal gap provided that the discrepancy sequence $\{H_t\}_{t\geq 0}$ exhibits a non-asymptotic decay rate.

**Lemma C.7.** *Suppose Assumptions 1.1 and 1.4 hold. Let $\{x_t\}_{t\geq 0}$ be a primal LMO sequence with stepsize $\tau_t = (t+1)^{-a}$ generated by the update rule (12a), let $\{y_t\}_{t\geq 0} \subset \mathcal{Y}$ be any dual sequence, and let $\beta_t = C(t+1)^{-b}$. Suppose $1 \geq a > b > 0$, $C \geq 0$, $\max\{C, \mu\} > 0$ and we have*

$$H_t \leq \frac{M_t}{(t+1)^c}, \quad \forall t \geq 0$$

*for some non-decreasing, positive sequence $\{M_t\}_{t\geq 0}$ and a constant $c$ satisfying $b < c \leq 2a - b$ if $C > 0$ and $0 < c \leq 2a$ if $C = 0$ and $\mu > 0$. Then, for any $T \geq 1$,*

$$G_{\mathcal{X}}^{\mathrm{Opt}}(x_T) \leq \frac{CD_{\mathcal{Y}}^2}{2(1-b)T^{a+b-1}} + \frac{4L_{yx}D_{\mathcal{X}}\sqrt{2CM_t} + CL_{xx}D_{\mathcal{X}}^2 + \frac{CL_{yx}^2 D_{\mathcal{X}}^2}{C+\mu}}{(2-c+b)(C+\mu)T^{(2a-b+c-2)/2}}$$
$$+ \frac{4L_{yx}D_{\mathcal{X}}\sqrt{2\mu M_t} + \mu L_{xx}D_{\mathcal{X}}^2 + \frac{\mu L_{yx}^2 D_{\mathcal{X}}^2}{C+\mu}}{(2-c)(C+\mu)T^{(2a+c-2)/2}}.$$

*Proof.* Since $0 < a \leq 1$, it holds that

$$1 - \tau_t = \frac{(t+1)^a - 1}{(t+1)^a} \leq \frac{t^a}{(t+1)^a}.$$

We also note that by Lemma A.1 (vi), for any $t \geq 0$,

$$f_t(x_t) - f_t(x^\star) + \frac{D_{\mathcal{Y}}^2 \beta_t}{2} \geq \left( f(x_t) - \frac{D_{\mathcal{Y}}^2 \beta_t}{2} \right) - f(x^\star) + \frac{D_{\mathcal{Y}}^2 \beta_t}{2} \geq f(x_t) - f(x^\star) \geq 0. \tag{28}$$

Similar to Lemma C.4, one can prove that for any $t \geq 0$,

$$2L_{yx}D_{\mathcal{X}}\sqrt{\frac{2\tau_t^2 H_t}{\beta_t + \mu}} + \frac{L_{xx}D_{\mathcal{X}}^2 \tau_t^2}{2} + \frac{L_{yx}^2 D_{\mathcal{X}}^2 \tau_t^2}{2(\beta_t + \mu)}$$
$$\leq \frac{4L_{yx}D_{\mathcal{X}}\sqrt{2CM_t} + CL_{xx}D_{\mathcal{X}}^2 + \frac{CL_{yx}^2 D_{\mathcal{X}}^2}{C+\mu}}{2(C+\mu)(t+1)^{(2a-b+c)/2}} + \frac{4L_{yx}D_{\mathcal{X}}\sqrt{2\mu M_t} + \mu L_{xx}D_{\mathcal{X}}^2 + \frac{\mu L_{yx}^2 D_{\mathcal{X}}^2}{C+\mu}}{2(C+\mu)(t+1)^{(2a+c)/2}}.$$

Hence, Lemma A.7 implies

$$f_{t+1}(x_{t+1}) - f_{t+1}(x^\star) + \frac{CD_{\mathcal{Y}}^2}{2(t+2)^b}$$
$$\leq \frac{t^a}{(t+1)^a}\left( f_t(x_t) - f_t(x^\star) + \frac{CD^2}{2(t+1)^b} \right) + \frac{CD_{\mathcal{Y}}^2}{2(t+1)^{a+b}}$$
$$+ \frac{4L_{yx}D_{\mathcal{X}}\sqrt{2CM_t} + CL_{xx}D_{\mathcal{X}}^2 + \frac{CL_{yx}^2 D_{\mathcal{X}}^2}{C+\mu}}{2(C+\mu)(t+1)^{(2a-b+c)/2}} + \frac{4L_{yx}D_{\mathcal{X}}\sqrt{2\mu M_t} + \mu L_{xx}D_{\mathcal{X}}^2 + \frac{\mu L_{yx}^2 D_{\mathcal{X}}^2}{C+\mu}}{2(C+\mu)(t+1)^{(2a+c)/2}}.$$

We deduce that

$$t^a \left( f_t(x_t) - f_t(x^\star) + \frac{CD_{\mathcal{Y}}^2}{2(t+1)^b} \right)$$
$$\leq (t-1)^a \left( f_{t-1}(x_{t-1}) - f_{t-1}(x^\star) + \frac{CD^2}{2t^b} \right) + \frac{CD_{\mathcal{Y}}^2}{2t^b}$$
$$+ \frac{4L_{yx}D_{\mathcal{X}}\sqrt{2CM_{t-1}} + CL_{xx}D_{\mathcal{X}}^2 + \frac{CL_{yx}^2 D_{\mathcal{X}}^2}{C+\mu}}{2(C+\mu)t^{(c-b)/2}} + \frac{4L_{yx}D_{\mathcal{X}}\sqrt{2\mu M_{t-1}} + \mu L_{xx}D_{\mathcal{X}}^2 + \frac{\mu L_{yx}^2 D_{\mathcal{X}}^2}{C+\mu}}{2(C+\mu)t^{c/2}}$$

$$\leq \sum_{i=1}^{t} \left( \frac{CD_{\mathcal{Y}}^2}{2i^b} + \frac{4L_{yx}D_{\mathcal{X}}\sqrt{2CM_{i-1}} + CL_{xx}D_{\mathcal{X}}^2 + \frac{CL_{yx}^2 D_{\mathcal{X}}^2}{C+\mu}}{2(C+\mu)i^{(c-b)/2}} + \frac{4L_{yx}D_{\mathcal{X}}\sqrt{2\mu M_{i-1}} + \mu L_{xx}D_{\mathcal{X}}^2 + \frac{\mu L_{yx}^2 D_{\mathcal{X}}^2}{C+\mu}}{2(C+\mu)i^{c/2}} \right)$$

$$\leq \frac{CD_{\mathcal{Y}}^2 t^{1-b}}{2(1-b)} + \frac{\left( 4L_{yx}D_{\mathcal{X}}\sqrt{2CM_t} + CL_{xx}D_{\mathcal{X}}^2 + \frac{CL_{yx}^2 D_{\mathcal{X}}^2}{C+\mu} \right) t^{(2-c+b)/2}}{(2-c+b)(C+\mu)}$$

$$+ \frac{\left( 4L_{yx}D_{\mathcal{X}}\sqrt{2\mu M_t} + \mu L_{xx}D_{\mathcal{X}}^2 + \frac{\mu L_{yx}^2 D_{\mathcal{X}}^2}{C+\mu} \right) t^{(2-c)/2}}{(2-c)(C+\mu)},$$

where the last inequality follows from the fact that $(c-b)/2 \leq a - b < 1$. By (28), we have that

$$f(x_t) - f(x^\star) \leq f_t(x_t) - f_t(x^\star) + \frac{CD_{\mathcal{Y}}^2}{2(t+1)^b},$$

which concludes the proof. $\qquad\square$

We now provide the detailed versions of the results summarized in Theorem 2.2, including explicit constants and optimal parameter configurations. We first address the settings without additional geometric assumptions on $\mathcal{Y}$.

**Theorem C.8** (Detailed version of Theorem 2.2 (i)–(ii))**.** *Suppose $\{x_t, y_t\}_{t\geq 0}$ is the sequence generated by Algorithm 1 under $\{\tau_t, \beta_t\}_{t\geq 0}$ satisfying Condition C.1 with $1 \geq a > 2b > 0$. If Assumption 1.1 and Assumption 1.4 hold, then for any $T \geq 1$,*

$$G_{\mathcal{X}}^{\mathrm{Opt}}(x_T) \leq \frac{CD_{\mathcal{Y}}^2}{2(1-b)T^b} + \frac{3\left( 4L_{yx}D_{\mathcal{X}}\sqrt{2CM_1} + CL_{xx}D_{\mathcal{X}}^2 + L_{yx}^2 D_{\mathcal{X}}^2 \right)}{(6+4b-2a)CT^{(4a-2b-3)/3}},$$

$$G_{\mathcal{Y}}(x_T, y_T) \leq \frac{M_1}{(T+1)^{(2a-b)/3}} + \frac{CD_{\mathcal{Y}}^2}{2(T+1)^b}.$$

*By optimizing over $a, b$, we obtain the optimal rate of $O(T^{-1/5})$ when $a = 1, b = \frac{1}{5}$. If Assumption 1.1 additionally holds with $\mu > 0$ and $C = 0, a = 1$, then for any $T \geq 1$,*

$$G_{\mathcal{X}}^{\mathrm{Opt}}(x_T) \leq \frac{3\left( 4L_{yx}D_{\mathcal{X}}\sqrt{2\mu M_2 \log(T+2)} + \mu L_{xx}D_{\mathcal{X}}^2 + L_{yx}^2 D_{\mathcal{X}}^2 \right)}{4\mu T^{1/3}},$$

$$G_{\mathcal{Y}}(x_T, y_T) \leq \frac{M_2}{(T+1)^{2/3}}.$$

*Proof.* The bounds on $G_{\mathcal{X}}^{\mathrm{Opt}}$ follow from an application of Lemma C.7. Specifically, by Lemma C.2, the decay condition on $H_t$ required in Lemma C.7 is satisfied with constant $c = (2a-b)/3$, where we set $b = 0$ when $\mu > 0$. Moreover, the conditions on $c$ in Lemma C.4 are trivially satisfied since $b < (2a-b)/3 \leq 2a - b$ and $0 < 2a/3 \leq 2a$. On the other hand, the bounds on $G_{\mathcal{Y}}$ follow from applying Lemma A.6 together with Lemma C.2. This completes the proof. $\qquad\square$

Next, we provide the detailed bounds for settings where the dual domain $\mathcal{Y}$ is strongly convex, enabling improved convergence rates.

**Theorem C.9** (Detailed version of Theorem 2.2 (iii)–(iv))**.** *Suppose $\{x_t, y_t\}_{t\geq 0}$ is the sequence generated by Algorithm 1 under $\{\tau_t, \beta_t\}_{t\geq 0}$ satisfying Condition C.1 with $1 = a > 2b > 0, C \geq 0$ such that $\max\{C, \mu\} > 0$ for any $t \geq 0$. If Assumption 1.1, Assumption 1.4 and Assumption 1.2 hold, then for any $T \geq 1$,*

$$G_{\mathcal{X}}^{\mathrm{Opt}}(x_T) \leq \frac{CD_{\mathcal{Y}}^2}{2(1-b)T^b} + \frac{4L_{yx}D_{\mathcal{X}}\sqrt{2CM_3} + CL_{xx}D_{\mathcal{X}}^2 + L_{yx}^2 D_{\mathcal{X}}^2}{(2+2b-a)CT^{(3a-2b-2)/2}} \quad \& \quad G_{\mathcal{Y}}(x_T, y_T) \leq \frac{M_3}{(T+1)^{a-b}} + \frac{CD_{\mathcal{Y}}^2}{2(T+1)^b}.$$

*By optimizing over $a, b$, we obtain the optimal rate of $O(T^{-1/4})$ when $a = 1, b = \frac{1}{4}$. If Assumption 1.1 additionally holds with $\mu > 0$ and $C = 0, a = 1$, then for any $T \geq 1$,*

$$G_{\mathcal{X}}^{\mathrm{Opt}}(x_T) \leq \frac{4L_{yx}D_{\mathcal{X}}\sqrt{2\mu M_4 \log(T+2)} + \mu L_{xx}D_{\mathcal{X}}^2 + L_{yx}^2 D_{\mathcal{X}}^2}{\mu T^{1/2}} \quad \& \quad G_{\mathcal{Y}}(x_T, y_T) \leq \frac{M_4}{T+1}.$$

*Proof.* The bounds on $G_{\mathcal{X}}^{\mathrm{LMO}}$ follow from an application of Lemma C.4. Specifically, by Lemma C.3, the decay condition on $H_t$ required in Lemma C.4 is satisfied with constant $c = a - b$, where we set $b = 0$ when $\mu > 0$. Moreover, the conditions on $c$ in Lemma C.4 are trivially satisfied since $a - b \leq 2a - b$ and $a \leq 2a$. On the other hand, the bounds on $G_{\mathcal{Y}}$ follow from applying Lemma A.6 together with Lemma C.3. This completes the proof. $\qquad\square$

*Remark* C.10. While assuming knowledge of the global Lipschitz constant $L_{yy}$ is standard in minimax analysis, it can be restrictive in practice. However, Algorithm 1 is fully compatible with backtracking line-search to estimate local constants $L_{yy}^{(t)}$. Specifically, following Lemma A.8, it suffices at iteration $t$ to satisfy

$$-\mathcal{L}_t(x_t, y_{t+1}) \leq -\mathcal{L}_t(x_t, y_t) - \gamma_t \nabla_y \mathcal{L}_t(x_t, y_t)^\top (u_t - y_t) + \frac{L_{yy}^{(t)} + \beta_t}{2}\|u_t - y_t\|^2 \gamma_t^2. \tag{29}$$

Such an $L_{yy}^{(t)}$ can be estimated efficiently via the backtracking procedure in (Pedregosa et al., 2020).

## D. Proofs of Section 3

We start the analysis with establishing an upper bound on $H_t$, which through Lemma A.6 will then be used in Lemma D.2 to provide a bound on the dual gap.

**Lemma D.1.** *Suppose $\{x_t, y_t\}_{t \geq 0}$ is the sequence generated by Algorithm 2 under $\tau_t \in (0, 1)$, $\gamma_t = \frac{1}{L_{yy} + \beta_t}$ and $\max\{\beta_t, \mu\} > 0$ for any $t \geq 0$. If Assumption 1.1 holds, then for any $t \geq 0$,*

$$\begin{aligned}
H_{t+1} &\leq (L_{yy} - \mu)\|y_t - y_t^\star(x_t)\|^2 - (L_{yy} + \beta_t)\|y_{t+1} - y_{t+1}^\star(x_{t+1})\|^2 + 4 L_{xx} D_{\mathcal{X}}^2 \tau_t^2 \\
&\quad + \frac{2(L_{yy} + \mu + \beta_t + \beta_{t+1})^2 (L_{yx} D_{\mathcal{X}} \tau_t + 2 D_{\mathcal{Y}}(\beta_t - \beta_{t+1}))^2}{(\beta_{t+1} + \mu)^3}.
\end{aligned}$$

*Proof.* From Lemma A.5, Lemma A.13 as well as $\|x_{t+1} - x_t\| \leq D_{\mathcal{X}} \tau_t$, we have

$$\begin{aligned}
\mathcal{L}_t(x_t, y_t^\star(x_t)) - \mathcal{L}_t(x_t, y_{t+1}) &\leq \frac{L_{yy} - \mu}{2}\|y_t - y_t^\star(x_t)\|^2 - \frac{L_{yy} + \beta_t}{2}\|y_{t+1} - y_{t+1}^\star(x_{t+1})\|^2 \\
&\quad + (L_{yy} + \beta_t)\left(\frac{L_{yx} D_{\mathcal{X}} \tau_t + 2 D_{\mathcal{Y}}(\beta_t - \beta_{t+1})}{\beta_{t+1} + \mu}\right)\sqrt{\frac{2 H_{t+1}}{\beta_{t+1} + \mu}},
\end{aligned}$$

and

$$\mathcal{L}_t(x_t, y_t^\star(x_t)) - \mathcal{L}_t(x_t, y_{t+1}) \geq H_{t+1} - (L_{yx} D_{\mathcal{X}} \tau_t + 2 D_{\mathcal{Y}}(\beta_t - \beta_{t+1}))\sqrt{\frac{2 H_{t+1}}{\beta_{t+1} + \mu}} - 2 L_{xx} D_{\mathcal{X}}^2 \tau_t^2.$$

Thus, we obtain

$$\begin{aligned}
&H_{t+1} - (L_{yx} D_{\mathcal{X}} \tau_t + 2 D_{\mathcal{Y}}(\beta_t - \beta_{t+1}))\left(1 + \frac{L_{yy} + \beta_t}{\beta_{t+1} + \mu}\right)\sqrt{\frac{2 H_{t+1}}{\beta_{t+1} + \mu}} \\
&\leq \frac{L_{yy} - \mu}{2}\|y_t - y_t^\star(x_t)\|^2 - \frac{L_{yy} + \beta_t}{2}\|y_{t+1} - y_{t+1}^\star(x_{t+1})\|^2 + 2 L_{xx} D_{\mathcal{X}}^2 \tau_t^2,
\end{aligned}$$

which is equivalent to

$$\begin{aligned}
&\left(\sqrt{H_{t+1}} - \frac{(L_{yy} + \mu + \beta_t + \beta_{t+1})(L_{yx} D_{\mathcal{X}} \tau_t + 2 D_{\mathcal{Y}}(\beta_t - \beta_{t+1}))}{\sqrt{2}(\beta_{t+1} + \mu)^{3/2}}\right)^2 \\
&\leq \frac{L_{yy} - \mu}{2}\|y_t - y_t^\star(x_t)\|^2 - \frac{L_{yy} + \beta_t}{2}\|y_{t+1} - y_{t+1}^\star(x_{t+1})\|^2 + 2 L_{xx} D_{\mathcal{X}}^2 \tau_t^2 \\
&\quad + \frac{(L_{yy} + \beta_t + \beta_{t+1})^2 (L_{yx} D_{\mathcal{X}} \tau_t + 2 D_{\mathcal{Y}}(\beta_t - \beta_{t+1}))^2}{2(\beta_{t+1} + \mu)^3},
\end{aligned}$$

Using the fact that $2(a^2 + b^2) \geq (a+b)^2$ for

$$a = \sqrt{H_{t+1}} - \frac{(L_{yy} + \mu + \beta_t + \beta_{t+1})\left(L_{yx}D_{\mathcal{X}}\tau_t + 2D_{\mathcal{Y}}(\beta_t - \beta_{t+1})\right)}{\sqrt{2}(\beta_{t+1} + \mu)^{3/2}},$$

$$b = \frac{(L_{yy} + \mu + \beta_t + \beta_{t+1})\left(L_{yx}D_{\mathcal{X}}\tau_t + 2D_{\mathcal{Y}}(\beta_t - \beta_{t+1})\right)}{\sqrt{2}(\beta_{t+1} + \mu)^{3/2}},$$

we conclude the proof. $\qquad\square$

Lemma D.2 uses Lemma D.1 to provide a bound on the accumulated dual gap terms, as well as a bound on the accumulated error terms $\|y_t^\star(x_t) - y_t\|$, which will later be used to bound the accumulated primal gap terms.

**Lemma D.2.** *Suppose $\{x_t, y_t\}_{t\geq 0}$ is the sequence generated by Algorithm 2 under $\tau_t \in (0,1)$, $\gamma_t = \frac{1}{L_{yy} + \beta_t}$ and $\max\{\beta_t, \mu\} > 0$ for any $t \geq 0$. If Assumption 1.1 holds, then for any $T \geq 1$,*

$$\sum_{t\in[T]} G_{\mathcal{Y}}(x_t, y_t) \leq (L_{yy} - \mu)D_{\mathcal{Y}}^2 + \sum_{t=0}^{T-1}\left(4L_{xx}D_{\mathcal{X}}^2\tau_t^2 + \frac{D_{\mathcal{Y}}^2\beta_{t+1}}{2}\right)$$

$$+ \sum_{t=0}^{T-1}\frac{2(L_{yy} + \mu + \beta_t + \beta_{t+1})^2\left(L_{yx}D_{\mathcal{X}}\tau_t + 2D_{\mathcal{Y}}(\beta_t - \beta_{t+1})\right)^2}{(\beta_{t+1} + \mu)^3},$$

*and*

$$\left(\sum_{t\in[T]}\|y_t^\star(x_t) - y_t\|\right)^2 \leq \left(\sum_{t\in[T]}\frac{1}{\beta_{t-1} + \mu}\right)\left((L_{yy} - \mu)D_{\mathcal{Y}}^2 + 4L_{xx}D_{\mathcal{X}}^2\sum_{t=0}^{T-1}\tau_t^2\right)$$

$$+ \left(\sum_{t\in[T]}\frac{1}{\beta_{t-1} + \mu}\right)\sum_{t=0}^{T-1}\frac{2(L_{yy} + \mu + \beta_t + \beta_{t+1})^2\left(L_{yx}D_{\mathcal{X}}\tau_t + 2D_{\mathcal{Y}}(\beta_t - \beta_{t+1})\right)^2}{(\beta_{t+1} + \mu)^3}.$$

*Proof.* By summing up the inequality in Lemma D.1 with $t = 0, \cdots, T-1$, we obtain

$$\sum_{t\in[T]} H_t \leq (L_{yy} - \mu)\|y_0 - y_0^\star(x_0)\|^2 - \sum_{t=1}^{T-1}(\mu + \beta_{t-1})\|y_t - y_t^\star(x_t)\|^2 - (L_{yy} + \beta_{T-1})\|y_T - y_T^\star(x_T)\|^2$$

$$+ 4L_{xx}D_{\mathcal{X}}^2\sum_{t=0}^{T-1}\tau_t^2 + \sum_{t=0}^{T-1}\frac{2(L_{yy} + \mu + \beta_t + \beta_{t+1})^2\left(L_{yx}D_{\mathcal{X}}\tau_t + 2D_{\mathcal{Y}}(\beta_t - \beta_{t+1})\right)^2}{(\beta_{t+1} + \mu)^3}. \tag{30}$$

Using Lemma A.6 and the fact that $\|y_t - y_t^\star(x_t)\|^2 \geq 0$, we finish the proof for the first part.

Now we turn to the second part. By re-arranging (30), we have

$$\sum_{t\in[T]}(\beta_{t-1} + \mu)\|y_t - y_t^\star(x_t)\|^2 \leq (L_{yy} - \mu)\|y_0 - y_0^\star(x_0)\|^2 - (L_{yy} - \mu)\|y_T - y_T^\star(x_T)\|^2 - \sum_{t\in[T]} H_t + 4L_{xx}D_{\mathcal{X}}^2\sum_{t=0}^{T-1}\tau_t^2$$

$$+ \sum_{t=0}^{T-1}\frac{2(L_{yy} + \mu + \beta_t + \beta_{t+1})^2\left(L_{yx}D_{\mathcal{X}}\tau_t + 2D_{\mathcal{Y}}(\beta_t - \beta_{t+1})\right)^2}{(\beta_{t+1} + \mu)^3}.$$

By the fact that $H_t \geq 0$, $\|y_T - y_T^\star(x_T)\|^2 \geq 0$ and $\mu \leq L_{yy}$, we have

$$\sum_{t\in[T]}(\beta_{t-1} + \mu)\|y_t - y_t^\star(x_t)\|^2 \leq (L_{yy} - \mu)\|y_0 - y_0^\star(x_0)\|^2 + 4L_{xx}D_{\mathcal{X}}^2\sum_{t=0}^{T-1}\tau_t^2$$

$$+ \sum_{t=0}^{T-1}\frac{2(L_{yy} + \mu + \beta_t + \beta_{t+1})^2\left(L_{yx}D_{\mathcal{X}}\tau_t + 2D_{\mathcal{Y}}(\beta_t - \beta_{t+1})\right)^2}{(\beta_{t+1} + \mu)^3}.$$

By Cauchy-Schwartz inequality, we deduce that

$$
\left( \sum_{t \in [T]} \| y_t^\star(x_t) - y_t \| \right)^2 \leq \left( \sum_{t \in [T]} (\beta_{t-1} + \mu) \| y_t - y_t^\star(x_t) \|^2 \right) \left( \sum_{t \in [T]} \frac{1}{\beta_{t-1} + \mu} \right)
$$

$$
\leq \left( \sum_{t \in [T]} \frac{1}{\beta_{t-1} + \mu} \right) \left( (L_{yy} - \mu) \| y_0 - y_0^\star(x_0) \|^2 + 4 L_{xx} D_{\mathcal{X}}^2 \sum_{t=0}^{T-1} \tau_t^2 \right)
$$

$$
+ \left( \sum_{t \in [T]} \frac{1}{\beta_{t-1} + \mu} \right) \sum_{t=0}^{T-1} \frac{2(L_{yy} + \mu + \beta_t + \beta_{t+1})^2 \left( L_{yx} D_{\mathcal{X}} \tau_t + 2 D_{\mathcal{Y}} (\beta_t - \beta_{t+1}) \right)^2}{(\beta_{t+1} + \mu)^3},
$$

which concludes the proof. $\qquad\square$

Lemma D.4 simplifies the sum terms appearing in the bounds from Lemma D.2, using the parameter schedules in Condition D.3.

*Condition* D.3. Sequences $\{\tau_t, \beta_t\}_{t \geq 0}$ satisfy

$$
\tau_t = (t+1)^{-a} \quad \& \quad \beta_t = C(t+1)^{-b},
$$

such that $\max\{C, \mu\} > 0$ for any $t \geq 0$.

**Lemma D.4.** *Suppose $\{x_t, y_t\}_{t \geq 0}$ is the sequence generated by Algorithm 2 under $\{\tau_t, \beta_t\}_{t \geq 0}$ satisfying Condition D.3 with $0 < b < 1$. If Assumption 1.1 holds, then for any $T \geq 1$,*

$$
\sum_{t=0}^{T-1} \frac{2(L_{yy} + \mu + \beta_t + \beta_{t+1})^2 \left( L_{yx} D_{\mathcal{X}} \tau_t + 2 D_{\mathcal{Y}} (\beta_t - \beta_{t+1}) \right)^2}{(\beta_{t+1} + \mu)^3}
$$

$$
\leq \frac{2(L_{yy} + \mu + 2C)^2 (L_{yx} D_{\mathcal{X}} + 2bC D_{\mathcal{Y}})^2}{(C^2 + \mu^2)^2} \sum_{t=0}^{T-1} \left( \frac{C 4^b}{(t+1)^{2a-3b}} + \frac{\mu}{(t+1)^{2a}} \right),
$$

*and hence, if $\frac{2a-1}{3} \leq b < \frac{2a}{3}$ then*

$$
\left( \sum_{t \in [T]} \| y_t^\star(x_t) - y_t \| \right)^2
$$

$$
\leq \frac{(CT^{1+b} + \mu T)}{(C + \mu)^2} \left( (L_{yy} - \mu) D_{\mathcal{Y}}^2 + 4 L_{xx} D_{\mathcal{X}}^2 \sum_{t=0}^{T-1} \frac{1}{(t+1)^{2a}} \right)
$$

$$
+ \frac{2(L_{yy} + \mu + 2C)^2 (L_{yx} D_{\mathcal{X}} + 2bC D_{\mathcal{Y}})^2 (CT^{1+b} + \mu T)}{(C^2 + \mu^2)^2 (C + \mu)^2} \left( 4^b C \begin{cases} \frac{T^{1-2a+3b}}{1-2a+3b} & 2a - 3b < 1 \\ \log(T) + 1 & 2a - 3b = 1 \end{cases} + \sum_{t=0}^{T-1} \frac{\mu}{(t+1)^{2a}} \right).
$$

*Proof.* For any $t \geq 0$, we have

$$
\frac{2(L_{yy} + \mu + \beta_t + \beta_{t+1})^2 \left( L_{yx} D_{\mathcal{X}} \tau_t + 2 D_{\mathcal{Y}} (\beta_t - \beta_{t+1}) \right)^2}{(\beta_{t+1} + \mu)^3}
$$

$$
= \frac{2(L_{yy} + \mu + C(t+1)^{-b} + C(t+2)^{-b})^2 \left( L_{yx} D_{\mathcal{X}} (t+1)^{-a} + 2C D_{\mathcal{Y}} ((t+1)^{-b} - (t+2)^{-b}) \right)^2}{(C(t+2)^{-b} + \mu)^3}
$$

$$
\leq \frac{2(L_{yy} + \mu + 2C)^2 \left( L_{yx} D_{\mathcal{X}} (t+1)^{-a} + 2bC D_{\mathcal{Y}} (t+1)^{-(b+1)} \right)^2}{(C(t+2)^{-b} + \mu)^3},
$$

where in the last inequality, we use Lemma F.2. Moreover, using Cauchy-Schwartz inequality, we have

$$
\frac{1}{(C(t+2)^{-b} + \mu)^3} \leq \frac{1}{C^3(t+2)^{-3b} + \mu^3} \leq \frac{1}{(C^2 + \mu^2)^2} \left( \frac{C}{(t+2)^{-3b}} + \mu \right) \leq \frac{1}{(C^2 + \mu^2)^2} \left( \frac{C 4^b}{(t+1)^{-3b}} + \mu \right).
$$

Moreover, we also have

$$\left(L_{yx}D_{\mathcal{X}}(t+1)^{-a} + 2bCD_{\mathcal{Y}}(t+1)^{-(b+1)}\right)^2 \leq \left(L_{yx}D_{\mathcal{X}}(t+1)^{-a} + 2bCD_{\mathcal{Y}}(t+1)^{-a}\right)^2$$
$$= (L_{yx}D_{\mathcal{X}} + 2bCD_{\mathcal{Y}})^2(t+1)^{-2a}.$$

We conclude the proof by observing that

$$\frac{2(L_{yy} + \mu + \beta_t + \beta_{t+1})^2 \left(L_{yx}D_{\mathcal{X}}\tau_t + 2D_{\mathcal{Y}}(\beta_t - \beta_{t+1})\right)^2}{(\beta_{t+1} + \mu)^3}$$
$$\leq \frac{2(L_{yy} + \mu + 2C)^2(L_{yx}D_{\mathcal{X}} + 2bCD_{\mathcal{Y}})^2}{(C^2 + \mu^2)^2}\left(\frac{C4^b}{(t+1)^{2a-3b}} + \frac{\mu}{(t+1)^{2a}}\right).$$

This finishes the proof of the first part. Turning the second part, from Cauchy-Schwartz inequality, we have that

$$\frac{1}{\beta_t + \mu} = \frac{1}{C(t+1)^{-b} + \mu} \leq \frac{1}{(C+\mu)^2}\left(\frac{C}{(t+1)^{-b}} + \mu\right),$$

which implies

$$\sum_{t\in[T]}\frac{1}{\beta_{t-1} + \mu} \leq \frac{1}{(C+\mu)^2}\sum_{t\in[T]}\left(\frac{C}{t^{-b}} + \mu\right) \leq \frac{CT^{1+b} + \mu T}{(C+\mu)^2}.$$

Substituting this into Lemma D.2 and using the first part conclude the proof of the second part. $\square$

### D.1. Proof of Theorem 3.1

Piecing all the results together, we visit the first convergence guarantee for Algorithm 2 under the vanilla setting Assumption 1.1 as shown in Theorem D.5.

**Theorem D.5** (Detailed version of Theorem 3.1 (i), (ii)). *Suppose $\{x_t, y_t\}_{t\geq 0}$ is the sequence generated by Algorithm 2 under $\{\tau_t, \beta_t\}_{t\geq 0}$ satisfying Condition D.3 with $0 < b < a < 1$. If Assumption 1.1 holds and $1 - 2a + 3b > 0$, then for any $T \geq 1$,*

$$\frac{\sum_{t\in[T]} G_{\mathcal{X}}^{\mathrm{LMO}}(x_t, y_t)}{T} \leq \frac{2\max_{x\in\mathcal{X}}|f(x)| + CD_{\mathcal{Y}}^2}{T^{1-a}} + \frac{L_{xx}D_{\mathcal{X}}^2}{2(1-a)T^a} + \frac{L_{yx}^2D_{\mathcal{X}}^2}{2(1-a+b)CT^{a-b}}$$
$$+ \frac{L_{yx}D_{\mathcal{X}}}{T^{(1-b)/2}}\sqrt{\frac{(L_{yy}-\mu)D_{\mathcal{Y}}^2 + 4L_{xx}D_{\mathcal{X}}^2\sum_{t=0}^{T-1}(t+1)^{-2a}}{(1+b)C}}$$
$$+ \frac{2^{b+1/2}L_{yx}D_{\mathcal{X}}(L_{yy}+2C)(L_{yx}D_{\mathcal{X}} + 2bCD_{\mathcal{Y}})}{T^{a-2b}C^2\sqrt{1-2a+3b}},$$

$$\frac{\sum_{t\in[T]} G_{\mathcal{Y}}(x_t, y_t)}{T} \leq \frac{(L_{yy}-\mu)D_{\mathcal{Y}}^2 + 4L_{xx}D_{\mathcal{X}}^2\left(\sum_{t=0}^{T-1}(t+1)^{-2a}\right)}{T} + \frac{CD_{\mathcal{Y}}^2}{2(1-b)T^b}$$
$$+ \frac{2^{2b+1}(L_{yy}+2C)^2(L_{yx}D_{\mathcal{X}} + 2bCD_{\mathcal{Y}})^2}{T^{2a-3b}(1-2a+3b)C^3}.$$

*By optimizing over $a, b$, the optimal rate is $O\left(T^{-1/4}\right)$, which is obtained when $a = \frac{3}{4}, b = \frac{1}{4}$. If Assumption 1.1 additionally holds with $\mu > 0$ and $C = 0$, then for any $T \geq 1$,*

$$\frac{\sum_{t\in[T]} G_{\mathcal{X}}^{\mathrm{LMO}}(x_t, y_t)}{T} \leq \frac{2\max_{x\in\mathcal{X}}|f(x)|}{T^{1-a}} + \left(L_{xx}D_{\mathcal{X}}^2 + \frac{L_{yx}^2D_{\mathcal{X}}^2}{\mu}\right)\frac{1}{2(1-a)T^a}$$
$$+ \frac{L_{yx}D_{\mathcal{X}}\left(\mu^3(L_{yy}-\mu)D_{\mathcal{Y}}^2 + 2(2\mu^3 L_{xx}D_{\mathcal{X}}^2 + (L_{yx}D_{\mathcal{X}}(L_{yy}+\mu))^2)\sum_{t=0}^{T-1}(t+1)^{-2a}\right)^{1/2}}{\mu^2 T^{1/2}},$$

*and*

$$\frac{\sum_{t \in [T]} G_{\mathcal{Y}}(x_t, y_t)}{T} \le \frac{\mu^3 (L_{yy} - \mu) D_{\mathcal{Y}}^2 + 2(2\mu^3 L_{xx} D_{\mathcal{X}}^2 + (L_{yx} D_{\mathcal{X}} (L_{yy} + \mu))^2) \sum_{t=0}^{T-1} (t+1)^{-2a}}{\mu^3 T}.$$

*By optimizing over a, the optimal rate is $O\left(T^{-1/2}\right)$, which is obtained when $a = \frac{1}{2}$.*

*Proof.* The bounds on the dual gap follow from the first claim in Lemma D.2 and the first claim in Lemma D.4. The bounds on primal gap follows from the first inequality in the second claim in Lemma A.6, the second claim in Lemma D.2 and the second claim in Lemma D.4. □

## D.2. Proof of Theorem 3.2

We investigate the convergence of Algorithm 2 under convexity of the primal variable, i.e., Assumption 1.4. We start with Lemma D.6, which bounds the sub-optimality gap in terms of the $\|y_t - y_t^\star(x_t)\|$ terms.

**Lemma D.6.** *Suppose $\{x_t, y_t\}_{t \ge 0}$ is the sequence generated by Algorithm 2 under $\{\tau_t, \beta_t\}_{t \ge 0}$ satisfying Condition D.3 with $0 < b < a \le 1$. If Assumption 1.1 holds, then for any $T \ge 1$,*

$$G_{\mathcal{X}}^{\mathrm{Opt}}(x_T) \le \frac{2L_{yx} D_{\mathcal{X}}}{T^a} \sum_{t \in [T]} \|y_{t-1} - y_{t-1}^\star(x_{t-1})\| + \frac{CD_{\mathcal{Y}}^2}{2(1-b)T^{a+b-1}}$$

$$+ \frac{\left(L_{xx} D_{\mathcal{X}}^2 + \frac{\mu L_{yx}^2 D_{\mathcal{X}}^2}{(C+\mu)^2}\right)}{2} \begin{cases} \frac{1}{(1-a)T^{2a-1}} & a \in (0,1) \\ \frac{\log(T)+1}{T} & a = 1 \end{cases} + \frac{CL_{yx}^2 D_{\mathcal{X}}^2}{2(1-a+b)(C+\mu)^2 T^{2a-b-1}}$$

*Proof.* From Lemma A.7, we have

$$f_{t+1}(x_{t+1}) - f_{t+1}(x^\star) + \frac{D_{\mathcal{Y}}^2 \beta_{t+1}}{2}$$

$$\le (1 - \tau_t)\left(f_t(x_t) - f_t(x^\star) + \frac{D_{\mathcal{Y}}^2 \beta_t}{2}\right) + \frac{D_{\mathcal{Y}}^2 \tau_t \beta_t}{2} + 2L_{yx} D_{\mathcal{X}} \tau_t \|y_t - y_t^\star(x_t)\| + \left(L_{xx} D_{\mathcal{X}}^2 + \frac{L_{yx}^2 D_{\mathcal{X}}^2}{\beta_t + \mu}\right) \frac{\tau_t^2}{2}.$$

Since $0 < a \le 1$, it holds that

$$\frac{(t+1)^a - 1}{(t+1)^a} \le \frac{t^a}{(t+1)^a},$$

for any $t \ge 0$. Combining this with Equation (28), we have

$$f_{t+1}(x_{t+1}) - f_{t+1}(x^\star) + \frac{CD_{\mathcal{Y}}^2}{2(t+2)^b}$$

$$\le \frac{t^a}{(t+1)^a}\left(f_t(x_t) - f_t(x^\star) + \frac{CD_{\mathcal{Y}}^2}{2(t+1)^b}\right) + \frac{CD_{\mathcal{Y}}^2}{2(t+1)^{a+b}} + \frac{2L_{yx} D_{\mathcal{X}} \|y_t - y_t^\star(x_t)\|}{(t+1)^a}$$

$$+ \frac{L_{xx} D_{\mathcal{X}}^2}{2(t+1)^{2a}} + \frac{CL_{yx}^2 D_{\mathcal{X}}^2}{2(C+\mu)^2(t+1)^{2a-b}} + \frac{\mu L_{yx}^2 D_{\mathcal{X}}^2}{2(C+\mu)^2(t+1)^{2a}}$$

$$= \frac{t^a}{(t+1)^a}\left(f_t(x_t) - f_t(x^\star) + \frac{CD_{\mathcal{Y}}^2}{2(t+1)^b}\right) + \frac{CD_{\mathcal{Y}}^2}{2(t+1)^{a+b}} + \frac{2L_{yx} D_{\mathcal{X}} \|y_t - y_t^\star(x_t)\|}{(t+1)^a}$$

$$+ \frac{L_{xx} D_{\mathcal{X}}^2 + \frac{\mu L_{yx}^2 D_{\mathcal{X}}^2}{(C+\mu)^2}}{2(t+1)^{2a}} + \frac{CL_{yx}^2 D_{\mathcal{X}}^2}{2(C+\mu)^2(t+1)^{2a-b}}.$$

We also deduce that for any $t \ge 1$,

$$t^a\left(f_t(x_t) - f_t(x^\star) + \frac{CD_{\mathcal{Y}}^2}{2(t+1)^b}\right)$$

$$\leq (t-1)^a \left( f_{t-1}(x_{t-1}) - f_{t-1}(x^\star) + \frac{CD_{\mathcal{Y}}^2}{2t^b} \right) + 2L_{yx}D_{\mathcal{X}} \| y_{t-1} - y_{t-1}^\star(x_{t-1}) \|$$

$$+ \frac{CD_{\mathcal{Y}}^2}{2t^b} + \frac{L_{xx}D_{\mathcal{X}}^2 + \frac{\mu L_{yx}^2 D_{\mathcal{X}}^2}{(C+\mu)^2}}{2t^a} + \frac{CL_{yx}^2 D_{\mathcal{X}}^2}{2(C+\mu)^2 t^{a-b}}.$$

Using this for $t = 1, \cdots T - 1$, we have

$$T^a \left( f_T(x_T) - f_T(x^\star) + \frac{CD_{\mathcal{Y}}^2}{2(T+1)^b} \right)$$

$$\leq \sum_{t \in [T]} \left( 2L_{yx}D_{\mathcal{X}} \| y_{t-1} - y_{t-1}^\star(x_{t-1}) \| + \frac{CD_{\mathcal{Y}}^2}{2t^b} + \frac{L_{xx}D_{\mathcal{X}}^2 + \frac{\mu L_{yx}^2 D_{\mathcal{X}}^2}{(C+\mu)^2}}{2t^a} + \frac{CL_{yx}^2 D_{\mathcal{X}}^2}{2(C+\mu)^2 t^{a-b}} \right)$$

$$\leq 2L_{yx}D_{\mathcal{X}} \sum_{t \in [T]} \| y_{t-1} - y_{t-1}^\star(x_{t-1}) \| + \frac{CD_{\mathcal{Y}}^2 T^{1-b}}{2(1-b)}$$

$$+ \frac{\left( L_{xx}D_{\mathcal{X}}^2 + \frac{\mu L_{yx}^2 D_{\mathcal{X}}^2}{(C+\mu)^2} \right)}{2} \begin{cases} \frac{T^{1-a}}{1-a} & a \in (0,1) \\ \log(T) + 1 & a = 1 \end{cases} + \frac{CL_{yx}^2 D_{\mathcal{X}}^2 T^{1-a+b}}{2(1-a+b)(C+\mu)^2}$$

where in the last inequality, we use Lemma F.3. From Lemma A.1 (vi), we have

$$f(x_T) - f(x^\star) \leq f_T(x_T) - f_T(x^\star) + \frac{CD_{\mathcal{Y}}^2}{2(T+1)^b},$$

which concludes the proof. $\qquad\qquad\square$

Given Lemmas D.2 and D.6, the convergence guarantee then follows immediately.

**Theorem D.7** (Detailed version of Theorem 3.2 (i)). *Suppose $\{x_t, y_t\}_{t\geq 0}$ is the sequence generated by Algorithm 2 under $\{\tau_t, \beta_t\}_{t\geq 0}$ satisfying Condition D.3 with $0 < b < a \leq 1$. If Assumptions 1.1 and 1.4 hold, then for any $T \geq 1$,*

$$G_{\mathcal{X}}^{\mathrm{Opt}}(x_T) \leq \frac{2L_{yx}D_{\mathcal{X}}}{T^{(2a-b-1)/2}} \sqrt{\frac{(L_{yy} - \mu)D_{\mathcal{Y}}^2 + 4L_{xx}D_{\mathcal{X}}^2 \sum_{t=0}^{T-1}(t+1)^{-2a}}{(1+b)C}} + \frac{CD_{\mathcal{Y}}^2}{2(1-b)T^{a+b-1}}$$

$$+ \frac{2^{b+3/2}L_{yx}D_{\mathcal{X}}(L_{yy} + 2C)(L_{yx}D_{\mathcal{X}} + 2bCD_{\mathcal{Y}})}{C^2} \begin{cases} \frac{1}{\sqrt{(1-2a+3b)T^{2a-2b-1}}} & 2a - 3b < 1 \\ \frac{\sqrt{\log(T)+1}}{T^{(1-b)/2}} & 2a - 3b = 1 \end{cases}$$

$$+ \frac{\left( L_{xx}D_{\mathcal{X}}^2 + \frac{\mu L_{yx}^2 D_{\mathcal{X}}^2}{(C+\mu)^2} \right)}{2} \begin{cases} \frac{1}{(1-a)T^{2a-1}} & a \in (0,1) \\ \frac{\log(T)+1}{T} & a = 1 \end{cases} + \frac{CL_{yx}^2 D_{\mathcal{X}}^2}{2(1-a+b)(C+\mu)^2 T^{2a-b-1}}$$

$$\frac{\sum_{t \in [T]} G_{\mathcal{Y}}(x_t, y_t)}{T} \leq \frac{(L_{yy} - \mu)D_{\mathcal{Y}}^2 + 4L_{xx}D_{\mathcal{X}}^2 \left( \sum_{t=0}^{T-1}(t+1)^{-2a} \right)}{T} + \frac{CD_{\mathcal{Y}}^2}{2(1-b)T^b}$$

$$+ \frac{2^{2b+1}(L_{yy} + 2C)^2(L_{yx}D_{\mathcal{X}} + 2bCD_{\mathcal{Y}})^2}{T^{2a-3b}(1-2a+3b)C^3}.$$

*By optimizing over $a, b$, the optimal rate is $\widetilde{O}\left(T^{-1/3}\right)$, which is obtained when $a = 1, b = \frac{1}{3}$.*

*Proof.* The bound on the primal gap follows from Lemma D.6 and Lemma D.4. The bound on the dual gap follows from Theorem D.5. $\qquad\square$

When $\mathcal{L}$ is strongly concave in $y$, i.e., Assumption 1.1 holds with $\mu > 0$, obtaining accelerated rates requires modified arguments. In particular, we provide a refined bound on the term $\| y_t - y_t^\star(x_t) \|$ in Lemma D.8 below. Since we assume $\beta_t = 0$ for the rest of this section, we use the notation $y^\star(\cdot)$ for $y_t^\star(\cdot)$.

**Lemma D.8.** *Suppose $\{x_t, y_t\}_{t \geq 0}$ is the sequence generated by Algorithm 2 under $1 \geq \tau_t \geq \tau_{t+1} \geq 0$, $\beta_t = 0$ and $\gamma_t = \frac{1}{L_{yy}}$ for any $t \geq 0$. If Assumption 1.1 with $\mu > 0$ holds, then for any $t \geq 1$,*

$$\|y_t - y^\star(x_t)\| \leq \rho^t D_{\mathcal{Y}} + \frac{L_{yy} D_{\mathcal{X}}}{\mu(1-\rho)} \left( \frac{1}{t} \sum_{i \in [t]} \tau_{i-1} \right),$$

*and*

$$G_{\mathcal{Y}}(x_{t+1}, y_{t+1}) \leq (L_{yy} - \mu) \left( \rho^t D_{\mathcal{Y}} + \frac{L_{yy} D_{\mathcal{X}}}{\mu(1-\rho)} \left( \frac{1}{t} \sum_{i \in [t]} \tau_{i-1} \right) \right)^2 + 2 \left( 2 L_{xx} D_{\mathcal{X}}^2 + \frac{L_{yx}^2 D_{\mathcal{X}}^2}{\mu} \right) \tau_t^2,$$

*where*

$$\rho := \sqrt{\frac{L_{yy} - \mu}{L_{yy}}} \in [0, 1).$$

*Proof.* From Lemma A.12, we have that

$$0 \leq L(x_t, y^\star(x_t)) - L(x_t, y_{t+1}) \leq \frac{L_{yy} - \mu}{2} \|y_t - y^\star(x_t)\|^2 - \frac{L_{yy}}{2} \|y_{t+1} - y^\star(x_t)\|^2, \tag{31}$$

which implies

$$\|y_{t+1} - y^\star(x_t)\| \leq \rho \|y_t - y^\star(x_t)\|.$$

By triangle inequality and Lemma A.1 (iii), we deduce that

$$\|y_{t+1} - y^\star(x_{t+1})\| \leq \|y_{t+1} - y^\star(x_t)\| + \|y^\star(x_t) - y^\star(x_{t+1})\|$$
$$\leq \rho \|y_t - y^\star(x_t)\| + \frac{L_{yy}}{\mu} \|x_{t+1} - x_t\|$$
$$\leq \rho \|y_t - y^\star(x_t)\| + \frac{L_{yy} D_{\mathcal{X}}}{\mu} \tau_t.$$

Thus, we have

$$\frac{\|y_t - y^\star(x_t)\|}{\rho^t} \leq \frac{\|y_{t-1} - y^\star(x_{t-1})\|}{\rho^{t-1}} + \frac{L_{yy} D_{\mathcal{X}}}{\mu} \frac{\tau_{t-1}}{\rho^t}$$
$$\leq \|y_0 - y^\star(x_0)\| + \frac{L_{yy} D_{\mathcal{X}}}{\mu} \sum_{i \in [t]} \frac{\tau_{i-1}}{\rho^i}$$
$$\leq D_{\mathcal{Y}} + \frac{L_{yy} D_{\mathcal{X}}}{\mu} \sum_{i \in [t]} \frac{\tau_{i-1}}{\rho^i}$$

We have from Lemma F.1 with $a_i = \frac{1}{\rho^i}$ and $b_i = \tau_{i-1}$ that

$$t \sum_{i \in [t]} \frac{\tau_{i-1}}{\rho^i} \leq \left( \sum_{i \in [t]} \frac{1}{\rho^i} \right) \left( \sum_{i \in [t]} \tau_{i-1} \right) = \frac{\rho^{-t} - 1}{1 - \rho} \left( \sum_{i \in [t]} \tau_{i-1} \right).$$

which concludes the proof of the first part. We have from Lemma A.5 that

$$L(x_t, y^\star(x_t)) - L(x_t, y_{t+1}) \geq G_{\mathcal{Y}}(x_{t+1}, y_{t+1}) - \frac{L_{yx} D_{\mathcal{X}} \sqrt{2} \tau_t}{\sqrt{\mu}} \sqrt{G_{\mathcal{Y}}(x_{t+1}, y_{t+1})} - 2 L_{xx} D_{\mathcal{X}}^2 \tau_t^2$$
$$\geq G_{\mathcal{Y}}(x_{t+1}, y_{t+1}) - \frac{L D \sqrt{2} \tau_t}{\sqrt{\mu}} \sqrt{G_{\mathcal{Y}}(x_{t+1}, y_{t+1})} - 2 L D^2 \tau_t^2$$

$$= \left( \sqrt{G_{\mathcal{Y}}(x_{t+1}, y_{t+1})} - \frac{LD\tau_t}{\sqrt{2\mu}} \right)^2 - \left( \frac{LD\tau_t}{\sqrt{2\mu}} \right)^2 - 2LD^2\tau_t^2.$$

Therefore, from the fact that $2(a^2 + b^2) \geq (a + b)^2$, we have for any $t \geq 1$

$$G_{\mathcal{Y}}(x_{t+1}, y_{t+1}) \leq 2 \left( \sqrt{G_{\mathcal{Y}}(x_{t+1}, y_{t+1})} - \frac{LD\tau_t}{\sqrt{2\mu}} \right)^2 + 2 \left( \frac{LD\tau_t}{\sqrt{2\mu}} \right)^2$$

$$\leq 2 \left( L(x_t, y^\star(x_t)) - L(x_t, y_{t+1}) + \left( \frac{LD\tau_t}{\sqrt{2\mu}} \right)^2 + 2LD^2\tau_t^2 \right) + 2 \left( \frac{LD\tau_t}{\sqrt{2\mu}} \right)^2$$

$$\leq (L_{yy} - \mu)\|y_t - y^\star(x_t)\|^2 - L_{yy}\|y_{t+1} - y^\star(x_t)\|^2 + 2 \left( 2LD^2 + \frac{L^2D^2}{\mu} \right) \tau_t^2$$

$$\leq (L_{yy} - \mu) \left( \rho^t D + \frac{LD}{\mu(1-\rho)} \left( \frac{1}{t} \sum_{i \in [t]} \tau_{i-1} \right) \right)^2 + 2 \left( 2LD^2 + \frac{L^2D^2}{\mu} \right) \tau_t^2,$$

where in the third inequality, we use (31). □

From Lemma D.8, bounding the desired gap terms for the convergence guarantee is almost immediate.

**Theorem D.9** (Detailed version of Theorem 3.1 (ii)). *Suppose $\{x_t, y_t\}_{t \geq 0}$ is the sequence generated by Algorithm 2 under $\{\tau_t, \beta_t\}_{t \geq 0}$ satisfying Condition D.3 with $0 < a \leq 1$ and $C = 0$ for any $t \geq 0$. If Assumption 1.1 with $\mu > 0$ and Assumption 1.4 hold, then for any $T \geq 1$,*

$$G_{\mathcal{X}}^{\text{Opt}}(x_T) \leq \frac{2\rho L_{yx} D_{\mathcal{X}} D_{\mathcal{Y}}}{(1-\rho)T^a} + \frac{2L_{yx} L_{yy} D_{\mathcal{X}}^2}{\mu(1-\rho)} \begin{cases} \frac{1}{(1-a)^2 T^{2a-1}} & a \in (0,1) \\ \frac{(\log(T)+1)^2}{T} & a = 1 \end{cases}$$

$$+ \frac{\left( L_{xx} D_{\mathcal{X}}^2 + \frac{L_{yx}^2 D_{\mathcal{X}}^2}{\mu} \right)}{2} \begin{cases} \frac{1}{(1-a)T^{2a-1}} & a \in (0,1) \\ \frac{\log(T)+1}{T} & a = 1 \end{cases}$$

$$\frac{\sum_{t \in [T]} G_{\mathcal{Y}}(x_t, y_t)}{T} \leq \frac{\mu^3(L_{yy} - \mu)D_{\mathcal{Y}}^2 + 2(2\mu^3 L_{xx} D_{\mathcal{X}}^2 + (L_{yx} D_{\mathcal{X}}(L_{yy} + \mu))^2) \sum_{t=0}^{T-1}(t+1)^{-2a}}{\mu^3 T}.$$

*By optimizing over $a$, the optimal rate is $\widetilde{O}(T^{-1})$, which is obtained when $a = 1$.*

*Proof.* We have from Lemma D.8 that if $a \in (0, 1)$,

$$\sum_{t \in [T]} \|y_{t-1} - y_{t-1}^\star(x_{t-1})\| \leq \sum_{t \in [T]} \left( \rho^t D_{\mathcal{Y}} + \frac{L_{yy} D_{\mathcal{X}}}{\mu(1-\rho)} \left( \frac{1}{t} \sum_{i \in [t]} \frac{1}{i^a} \right) \right)$$

$$\leq \sum_{t \in [T]} \left( \rho^t D_{\mathcal{Y}} + \frac{L_{yy} D_{\mathcal{X}}}{\mu(1-a)(1-\rho)t^a} \right)$$

$$\leq \frac{D_{\mathcal{Y}}\rho}{1-\rho} + \frac{L_{yy} D_{\mathcal{X}} T^{1-a}}{\mu(1-a)^2(1-\rho)}$$

If $a = 1$ then

$$\sum_{t \in [T]} \|y_{t-1} - y_{t-1}^\star(x_{t-1})\| \leq \sum_{t \in [T]} \left( \rho^t D_{\mathcal{Y}} + \frac{L_{yy} D_{\mathcal{X}}}{\mu(1-\rho)} \left( \frac{1}{t} \sum_{i \in [t]} \frac{1}{i} \right) \right)$$

$$\leq \sum_{t \in [T]} \left( \rho^t D_{\mathcal{Y}} + \frac{L_{yy} D_{\mathcal{X}}(\log(t) + 1)}{\mu t} \right)$$

$$\leq \frac{D_{\mathcal{Y}}\rho}{1-\rho} + \frac{L_{yy} D_{\mathcal{X}}(\log(T) + 1)^2}{\mu(1-\rho)}$$

This concludes the proof of the bound on the primal gap by using Lemma D.6. The bound on the dual gap follows from Theorem D.5. □

*Remark* D.10. Following Lemma A.12, Algorithm 2 remains convergent if $L_{yy}$ is replaced by a local constant $L_{yy}^{(t)}$ satisfying

$$-\mathcal{L}_t(x_t, y_{t+1}) \leq -\mathcal{L}_t(x_t, y_t) - \nabla_y \mathcal{L}_t(x_t, y_t)^\top (y_{t+1} - y_t) + \frac{L_{yy}^{(t)} + \beta_t}{2} \|y_{t+1} - y_t\|^2.$$

This requirement aligns with the proximal gradient line-search framework in (Beck, 2017, Section 10.4.2).

# E. Proofs of Section 4

To start the analysis, recall that the primal quantity of interest here is $\frac{1}{\tau_t}\|x_{t+1} - x_t\|$. Lemma A.10 provides *almost* a telescoping bound for the terms $\frac{1}{\tau_t^2}\|x_{t+1} - x_t\|^2$ except for a nuisance term

$$\frac{1}{4}\min\left\{\frac{H_t^2}{(L_{yy} + \beta_t)D_{\mathcal{Y}}^2}, H_t\right\}.$$

Lemma E.1 provides a further bound on this nuisance term, so that a telescoping sum can be applied to Lemma A.10, which will be done in Lemma E.2.

**Lemma E.1.** *Suppose $\{x_t, y_t\}_{t\geq 0}$ is the sequence generated by Algorithm 3 under $\tau_t > 0$, $\beta_t \geq \beta_{t+1} \geq 0$ and $\max\{\beta_t, \mu\} > 0$ for any $t \geq 0$. If Assumption 1.1 holds, then for any $t \geq 0$,*

$$\frac{1}{4}\min\left\{\frac{H_t^2}{(L_{yy}+\beta_t)D_{\mathcal{Y}}^2}, H_t\right\} \leq q_t - q_{t+1} + \left(2L_{xx} + \frac{4L_{yx}^2}{\beta_{t+1}+\mu} + \frac{1}{2\tau_t}\right)\|x_{t+1} - x_t\|^2$$

$$+ \frac{D_{\mathcal{Y}}^2(\beta_t - \beta_{t+1})^2}{L_{yx}^2\tau_t} + \frac{4D_{\mathcal{Y}}^2(\beta_t - \beta_{t+1})^2}{\beta_{t+1}+\mu} + \frac{4L_{yx}^4(L_{yy}+\beta_{t+1})D_{\mathcal{Y}}^2\tau_t^2}{(\beta_{t+1}+\mu)^2}$$

*where for any $t \geq 0$,*

$$q_t := \max\left\{\frac{H_t}{2}, H_t - \frac{H_t^2}{2(L_{yy}+\beta_t)D_{\mathcal{Y}}^2}\right\} + \frac{1}{4}\min\left\{\frac{H_t^2}{(L_{yy}+\beta_t)D_{\mathcal{Y}}^2}, H_t\right\}.$$

*Proof.* Using Lemma A.2 and Lemma A.8, we obtain

$$H_{t+1} - \frac{\left(L_{yx}\|x_{t+1} - x_t\| + D_{\mathcal{Y}}(\beta_t - \beta_{t+1})\right)\sqrt{2}}{\sqrt{\beta_{t+1}+\mu}}\sqrt{H_{t+1}}$$

$$\leq \max\left\{\frac{H_t}{2}, H_t - \frac{H_t^2}{2(L_{yy}+\beta_t)D_{\mathcal{Y}}^2}\right\} + 2L_{xx}\|x_{t+1} - x_t\|^2. \tag{32}$$

If $H_{t+1} \leq (L_{yy}+\beta_{t+1})D_{\mathcal{Y}}^2$ then we have

$$q_{t+1} = H_{t+1} - \frac{H_{t+1}^2}{2(L_{yy}+\beta_{t+1})D_{\mathcal{Y}}^2} + \frac{H_{t+1}^2}{4(L_{yy}+\beta_{t+1})D_{\mathcal{Y}}^2} = H_{t+1} - \frac{H_{t+1}^2}{4(L_{yy}+\beta_{t+1})D_{\mathcal{Y}}^2}.$$

By AM-GM inequality for four numbers, we have

$$\left(L_{yx}\|x_{t+1} - x_t\| + D_{\mathcal{Y}}(\beta_t - \beta_{t+1})\right)\sqrt{\frac{2H_{t+1}}{\beta_{t+1}+\mu}}$$

$$= \left(\left(\frac{(L_{yy}+\beta_{t+1})^{1/4}\sqrt{2D_{\mathcal{Y}}}\left(L_{yx}\|x_{t+1}-x_t\| + D_{\mathcal{Y}}(\beta_t-\beta_{t+1})\right)}{\sqrt{\beta_{t+1}+\mu}}\right)^{1/3}\right)^3 \sqrt{\frac{H_{t+1}}{\sqrt{L_{yy}+\beta_{t+1}}D_{\mathcal{Y}}}}$$

$$\leq \frac{H_{t+1}^2}{4(L_{yy}+\beta_{t+1})D_{\mathcal{Y}}^2} + \frac{3}{4}\left(\frac{(L_{yy}+\beta_{t+1})^{1/4}\sqrt{2D_{\mathcal{Y}}}\left(L_{yx}\|x_{t+1}-x_t\| + D_{\mathcal{Y}}(\beta_t-\beta_{t+1})\right)}{\sqrt{\beta_{t+1}+\mu}}\right)^{4/3}$$

$$= H_{t+1} - q_{t+1} + \frac{3}{4}\left(\frac{(L_{yy}+\beta_{t+1})^{1/4}\sqrt{2D_{\mathcal{Y}}}\left(L_{yx}\|x_{t+1}-x_t\| + D_{\mathcal{Y}}(\beta_t-\beta_{t+1})\right)}{\sqrt{\beta_{t+1}+\mu}}\right)^{4/3}. \tag{33}$$

Combining (32), (33) and defining

$$A_t := \frac{3}{4} \left( \frac{(L_{yy} + \beta_{t+1})^{1/4} \sqrt{2D_{\mathcal{Y}}} \left( L_{yx} \|x_{t+1} - x_t\| + D_{\mathcal{Y}}(\beta_t - \beta_{t+1}) \right)}{\sqrt{\beta_{t+1} + \mu}} \right)^{4/3}$$

give us

$$q_{t+1} \leq \max \left\{ \frac{H_t}{2}, H_t - \frac{H_t^2}{2(L_{yy} + \beta_t)D_{\mathcal{Y}}^2} \right\} + 2L_{xx}\|x_{t+1} - x_t\|^2 + A_t,$$

which implies

$$\frac{1}{4} \min \left\{ \frac{H_t^2}{(L_{yy} + \beta_t)D_{\mathcal{Y}}^2}, H_t \right\} \leq q_t - q_{t+1} + 2L_{xx}\|x_{t+1} - x_t\|^2 + A_t,$$

If $H_{t+1} > (L_{yy} + \beta_{t+1})D_{\mathcal{Y}}^2$ then we have

$$q_{t+1} = \frac{H_{t+1}}{2} + \frac{H_{t+1}}{4} = H_{t+1} - \frac{H_{t+1}}{4}.$$

By AM-GM inequality for two numbers, we have

$$\frac{2 \left( L_{yx} \|x_{t+1} - x_t\| + D_{\mathcal{Y}}(\beta_t - \beta_{t+1}) \right)}{\sqrt{\beta_t + \mu}} \sqrt{\frac{H_{t+1}}{2}}$$

$$\leq \frac{H_{t+1}}{4} + \frac{2 \left( L_{yx} \|x_{t+1} - x_t\| + D_{\mathcal{Y}}(\beta_t - \beta_{t+1}) \right)^2}{\beta_t + \mu} \tag{34}$$

$$= H_{t+1} - q_{t+1} + \frac{2 \left( L_{yx} \|x_{t+1} - x_t\| + D_{\mathcal{Y}}(\beta_t - \beta_{t+1}) \right)^2}{\beta_t + \mu}.$$

Combining (32), (34) and defining

$$B_t := \frac{2 \left( L_{yx} \|x_{t+1} - x_t\| + D_{\mathcal{Y}}(\beta_t - \beta_{t+1}) \right)^2}{\beta_t + \mu}$$

give us

$$q_{t+1} \leq \max \left\{ \frac{H_t}{2}, H_t - \frac{H_t^2}{2(L_{yy} + \beta_t)D_{\mathcal{Y}}^2} \right\} + 2L_{xx}\|x_{t+1} - x_t\|^2 + B_t,$$

which implies

$$\frac{1}{4} \min \left\{ \frac{H_t^2}{(L_{yy} + \beta_t)D_{\mathcal{Y}}^2}, H_t \right\} \leq q_t - q_{t+1} + 2L_{xx}\|x_{t+1} - x_t\|^2 + B_t,$$

In both cases, we have

$$\frac{1}{4} \min \left\{ \frac{H_t^2}{(L_{yy} + \beta_t)D_{\mathcal{Y}}^2}, H_t \right\} \leq q_t - q_{t+1} + 2L_{xx}\|x_{t+1} - x_t\|^2 + \frac{2 \left( L_{yx} \|x_{t+1} - x_t\| + D_{\mathcal{Y}}(\beta_t - \beta_{t+1}) \right)^2}{\beta_t + \mu}$$

$$+ \frac{3}{4} \left( \frac{(L_{yy} + \beta_{t+1})^{1/4} \sqrt{2D_{\mathcal{Y}}} \left( L_{yx} \|x_{t+1} - x_t\| + D_{\mathcal{Y}}(\beta_t - \beta_{t+1}) \right)}{\sqrt{\beta_{t+1} + \mu}} \right)^{4/3}. \tag{35}$$

Using the inequality $(a + b)^2 \leq 2(a^2 + b^2)$ and the fact that $\beta_t \geq \beta_{t+1}$, we have that

$$\frac{2 \left( L_{yx} \|x_{t+1} - x_t\| + D_{\mathcal{Y}}(\beta_t - \beta_{t+1}) \right)^2}{\beta_t + \mu} \leq \frac{4L_{yx}^2 \|x_{t+1} - x_t\|^2}{\beta_t + \mu} + \frac{4D_{\mathcal{Y}}^2(\beta_{t+1} - \beta_t)^2}{\beta_{t+1} + \mu}$$

By AM-GM inequality for three numbers, we have

$$
\left( \frac{(L_{yy} + \beta_{t+1})^{1/4}\sqrt{2D_{\mathcal{Y}}}\left( L_{yx}\|x_{t+1} - x_t\| + D_{\mathcal{Y}}(\beta_t - \beta_{t+1})\right)}{\sqrt{\beta_{t+1} + \mu}} \right)^{4/3}
$$

$$
= \left( \left( \frac{\|x_{t+1} - x_t\| + L_{yx}^{-1}D(\beta_t - \beta_{t+1})}{\sqrt{2\tau_t}} \right)^{2/3} \right)^2 \left( \frac{2L_{yx}(L_{yy} + \beta_{t+1})^{1/4}\sqrt{D_{\mathcal{Y}}\tau_t}}{\sqrt{\beta_{t+1} + \mu}} \right)^{4/3}
$$

$$
\leq \frac{2}{3}\frac{\left( \|x_{t+1} - x_t\| + L_{yx}^{-1}D_{\mathcal{Y}}(\beta_t - \beta_{t+1})\right)^2}{2\tau_t} + \frac{1}{3}\frac{16L_{yx}^4(L_{yy} + \beta_{t+1})D_{\mathcal{Y}}^2\tau_t^2}{(\beta_{t+1} + \mu)^2}
$$

$$
\leq \frac{2\|x_{t+1} - x_t\|^2 + 2L_{yx}^{-2}D_{\mathcal{Y}}^2(\beta_t - \beta_{t+1})^2}{3\tau_t} + \frac{16L_{yx}^4(L_{yy} + \beta_{t+1})D^2\tau_t^2}{3(\beta_{t+1} + \mu)^2}.
$$

Substituting this bound into (35) concludes the proof. □

At this point, one may recognize that one major difference in analysis of Algorithm 3 and the previous Algorithms 1 and 2 is that we cannot follow the path where the bound on $H_t$ is established first and the bounds on primal and dual gaps follow from there. Here, two sequences $\{H_t\}_{t \geq 0}$ and $\|x_{t+1} - x_t\|$ intertwine as indicated by Lemma A.10 and Lemma E.1 and, hence, add additional complication for the analysis. However, such complication can be resolve as shown in Lemma E.2.

**Lemma E.2.** *Suppose $\{x_t, y_t\}_{t \geq 0}$ is the sequence generated by Algorithm 3 under $\beta_t \geq \beta_{t+1} \geq 0$ such that $\max\{\beta_t, \mu\} > 0$, $\tau_t \geq \tau_{t+1} > 0$ for any $t \geq 0$. If Assumption 1.1 holds, then for any $t \geq 0$,*

$$
\left( \frac{1}{4\tau_t} - \frac{5L_{xx}}{2} - \frac{13L_{yx}^2}{2(\beta_{t+1} + \mu)} \right)\|x_{t+1} - x_t\|^2
$$

$$
\leq r_t - r_{t+1} + \frac{D_{\mathcal{Y}}^2(\beta_t - \beta_{t+1})^2}{L_{yx}^2\tau_t} + \frac{4D_{\mathcal{Y}}^2(\beta_t - \beta_{t+1})^2}{\beta_{t+1} + \mu} + \frac{8L_{yx}^4(L_{yy} + \beta_{t+1})D_{\mathcal{Y}}^2\tau_t^2}{(\beta_{t+1} + \mu)^2}, \tag{36}
$$

*where for any $t \geq 0$, $q_t$ is defined as in Lemma E.1 and*

$$
r_t := f_t(x_t) + \frac{D_{\mathcal{Y}}^2\beta_t}{2} + q_t.
$$

*Hence, if it additionally holds that*

$$
G_1 := \inf_{t \geq 0}\left\{ \frac{1}{4} - \frac{5L_{xx}\tau_t}{2} - \frac{13L_{yx}^2\tau_t}{2(\beta_{t+1} + \mu)} \right\} > 0,
$$

*then*

$$
\sum_{t=0}^{T}\left\| \frac{x_{t+1} - x_t}{\tau_t} \right\|^2 \leq \frac{2\sup_{t \geq 0}|r_t|}{G_1\tau_T} + \sum_{t=0}^{T}\left( \frac{D_{\mathcal{Y}}^2(\beta_t - \beta_{t+1})^2}{G_1 L_{yx}^2\tau_t^2} + \frac{4D_{\mathcal{Y}}^2(\beta_t - \beta_{t+1})^2}{G_1(\beta_{t+1} + \mu)\tau_t} + \frac{8L_{yx}^4(L_{yy} + \beta_{t+1})D_{\mathcal{Y}}^2\tau_t}{G_1(\beta_{t+1} + \mu)^2} \right).
$$

*Proof.* Combining Lemma A.10 and Lemma E.1 gives us (36).

By dividing both sides of (36) by $\tau_t$ summing up the inequality for $t = 0, 1, \dots, T$, we obtain

$$
\sum_{t=0}^{T}\left( \frac{1}{4} - \frac{5L_{xx}\tau_t}{2} - \frac{13L_{yx}^2\tau_t}{2(\beta_{t+1} + \mu)} \right)\left\| \frac{x_{t+1} - x_t}{\tau_t} \right\|^2
$$

$$
\leq \frac{r_0}{\tau_0} + \sum_{t \in [T]}\left( \frac{1}{\tau_t} - \frac{1}{\tau_{t-1}} \right)r_t - \frac{r_{T+1}}{\tau_T} + \sum_{t=0}^{T}\left( \frac{D_{\mathcal{Y}}^2(\beta_t - \beta_{t+1})^2}{L_{yx}^2\tau_t^2} + \frac{4D_{\mathcal{Y}}^2(\beta_t - \beta_{t+1})^2}{(\beta_{t+1} + \mu)\tau_t} + \frac{8L_{yx}^4(L_{yy} + \beta_{t+1})D_{\mathcal{Y}}^2\tau_t}{(\beta_{t+1} + \mu)^2} \right)
$$

$$
\leq \left( \frac{1}{\tau_0} + \sum_{t \in [T]}\left( \frac{1}{\tau_t} - \frac{1}{\tau_{t-1}} \right) + \frac{1}{\tau_T} \right)\sup_{t \geq 0}|r_t|
$$

$$+ \sum_{t=0}^{T} \left( \frac{D_{\mathcal{Y}}^2 (\beta_t - \beta_{t+1})^2}{L_{yx}^2 \tau_t^2} + \frac{4 D_{\mathcal{Y}}^2 (\beta_t - \beta_{t+1})^2}{(\beta_{t+1} + \mu) \tau_t} + \frac{8 L_{yx}^4 (L_{yy} + \beta_{t+1}) D_{\mathcal{Y}}^2 \tau_t}{(\beta_{t+1} + \mu)^2} \right)$$

$$\leq \frac{2 \sup_{t \geq 0} |r_t|}{\tau_T} + \sum_{t=0}^{T} \left( \frac{D_{\mathcal{Y}}^2 (\beta_t - \beta_{t+1})^2}{L_{yx}^2 \tau_t^2} + \frac{4 D_{\mathcal{Y}}^2 (\beta_t - \beta_{t+1})^2}{(\beta_{t+1} + \mu) \tau_t} + \frac{8 L_{yx}^4 (L_{yy} + \beta_{t+1}) D_{\mathcal{Y}}^2 \tau_t}{(\beta_{t+1} + \mu)^2} \right).$$

By the definition of $G_1$, we finish the proof for the second claim. $\qquad\square$

We note that if $\tau_t \leq \tau_0$ for any $t \geq 0$ then by using (Beck, 2017, Theorem 10.9), we have

$$\left\| \frac{x_{t+1} - x_t}{\tau_t} \right\| = \left\| \frac{1}{\tau_t} \left( \mathrm{PO}_{\mathcal{X}} \left( x_t - \tau_t \nabla_x \mathcal{L}(x_t, y_t) \right) - x_t \right) \right\|$$

$$\geq \left\| \frac{1}{\tau_0} \left( \mathrm{PO}_{\mathcal{X}} \left( x_t - \tau_0 \nabla_x \mathcal{L}(x_t, y_t) \right) - x_t \right) \right\|$$

$$=: G_{\mathcal{X}}^{\mathrm{PO}}(x_t, y_t),$$

Using the above results in this section, we arrive at somewhat generic bounds on the accumulated primal and dual gaps in Lemma E.3 but as soon as we introduce specific choices for $\{\tau_t, \beta_t\}_{t \geq 0}$, the convergence rates follow immediately as shown in Theorem E.4.

**Lemma E.3.** *Suppose $\{x_t, y_t\}_{t \geq 0}$ is the sequence generated by Algorithm 3 under $\beta_t \geq \beta_{t+1} \geq 0$ such that $\max\{\beta_t, \mu\} > 0$, $\tau_t \geq \tau_{t+1} > 0$ for any $t \geq 0$. If Assumption 1.1 and that $G_1$ defined in Lemma E.2 is positive hold, then for any $t \geq 0$,*

$$\frac{1}{T+1} \sum_{t=0}^{T} G_{\mathcal{X}}^{\mathrm{PO}}(x_t, y_t) \leq \left( \frac{2 \sup_{t \geq 0} |r_t|}{G_1 (T+1) \tau_T} + \frac{1}{T+1} \sum_{t=0}^{T} \left( \frac{D_{\mathcal{Y}}^2 (\beta_t - \beta_{t+1})^2}{G_1 L_{yx}^2 \tau_t^2} \right. \right.$$

$$\left. \left. + \frac{4 D_{\mathcal{Y}}^2 (\beta_t - \beta_{t+1})^2}{G_1 (\beta_{t+1} + \mu) \tau_t} + \frac{8 L_{yx}^4 (L_{yy} + \beta_{t+1}) D_{\mathcal{Y}}^2 \tau_t}{G_1 (\beta_{t+1} + \mu)^2} \right) \right)^{1/2},$$

$$\frac{1}{T+1} \sum_{t=0}^{T} G_{\mathcal{Y}}(x_t, y_t) \leq \left( \frac{F_1}{T+1} \left( 2 \sup_{t \geq 0} |s_t| + \sum_{t=0}^{T} \left( \frac{(\beta_t - \beta_{t+1})^2}{L_{yx}^2 \tau_t} + \frac{4 (\beta_t - \beta_{t+1})^2}{\beta_{t+1} + \mu} \right) (U_1 + 1) D_{\mathcal{Y}}^2 \right. \right.$$

$$\left. \left. + \frac{4 (2 U_1 + 1) L_{yx}^4 (L_{yy} + \beta_{t+1}) D_{\mathcal{Y}}^2 \tau_t^2}{(\beta_{t+1} + \mu)^2} \right) \right)^{1/2} + \frac{D_{\mathcal{Y}}^2}{2(T+1)} \sum_{t=0}^{T} \beta_t,$$

*where for any $t \geq 0$,*

$$U_1 := \sup_{t \geq 0} \frac{2 L_{xx} + \frac{4 L_{yy}^2}{\beta_{t+1} + \mu} + \frac{1}{2\tau_t}}{\frac{1}{4\tau_t} - \frac{5 L_{xx}}{2} - \frac{12 L_{yy}^2 + L_{yx}^2}{2(\beta_{t+1} + \mu)}}, \quad \& \quad F_1 := \sup_{t \geq 0} \max\{H_t, (L_{yy} + \beta_t) D_{\mathcal{Y}}^2\}, \quad \& \quad s_t := q_t + U_1 r_t.$$

*Proof.* The first claim follows immediately from Lemma E.2 and the concavity of the square root function. Now, we turn to the second claim. From the definition of $F_1$, we have

$$\min \left\{ \frac{H_t^2}{(L_{yy} + \beta_t) D_{\mathcal{Y}}^2}, H_t \right\} \geq \min \left\{ \frac{H_t^2}{F_1}, H_t \right\} = \frac{H_t^2}{F_1}.$$

By using Lemma E.1 and Lemma E.2, we have

$$\frac{H_t^2}{4 F_1} \leq q_t - q_{t+1} + \left( 2 L_{xx} + \frac{4 L_{yy}^2}{\beta_{t+1} + \mu} + \frac{1}{2\tau_t} \right) \| x_{t+1} - x_t \|^2$$

$$+ \frac{D_{\mathcal{Y}}^2 (\beta_t - \beta_{t+1})^2}{L_{yx}^2 \tau_t} + \frac{4 D_{\mathcal{Y}}^2 (\beta_t - \beta_{t+1})^2}{\beta_{t+1} + \mu} + \frac{4 L_{yx}^4 (L_{yy} + \beta_{t+1}) D^2 \tau_t^2}{(\beta_{t+1} + \mu)^2}$$

$$\leq s_t - s_{t+1} + \frac{(U_1+1)D_{\mathcal{Y}}^2(\beta_t - \beta_{t+1})^2}{L_{yx}^2 \tau_t} + \frac{4(U_1+1)D_{\mathcal{Y}}^2(\beta_t - \beta_{t+1})^2}{\beta_{t+1} + \mu} + \frac{4(2U_1+1)L_{yx}^4(L_{yy} + \beta_{t+1})D_{\mathcal{Y}}^2\tau_t^2}{(\beta_{t+1} + \mu)^2}$$

Thus, by summing the above inequality with $t = 0, \cdots, T$, we have

$$\sum_{t=0}^{T} \frac{H_t^2}{F_1} \leq s_0 - s_{T+1} + \sum_{t=0}^{T} \left( \frac{1}{L_{yx}^2 \tau_t} + \frac{4}{\beta_{t+1} + \mu} \right) (U_1+1)D_{\mathcal{Y}}^2(\beta_t - \beta_{t+1})^2$$

$$+ \sum_{t=0}^{T} \frac{4(2U_1+1)L_{yx}^4(L_{yy} + \beta_{t+1})D_{\mathcal{Y}}^2\tau_t^2}{(\beta_{t+1} + \mu)^2}.$$

By using the convexity of $(\cdot)^2$ for the left-hand side, we obtain

$$\left( \frac{\sum_{t=0}^{T} H_t}{T+1} \right)^2 \leq \frac{F_1}{T+1} \left( s_0 - s_{T+1} + \sum_{t=0}^{T} \left( \frac{1}{L_{yx}^2 \tau_t} + \frac{4}{\beta_{t+1} + \mu} \right) (U_1+1)D_{\mathcal{Y}}^2(\beta_t - \beta_{t+1})^2 \right.$$

$$\left. + \sum_{t=0}^{T} \frac{4(2U_1+1)L_{yx}^4(L_{yy} + \beta_{t+1})D_{\mathcal{Y}}^2\tau_t^2}{(\beta_{t+1} + \mu)^2} \right).$$

Using this and Lemma A.6, we finish the proof for the second part. $\qquad\square$

**Theorem E.4** (Detailed version of Theorem 4.1 (i),(ii)). *Suppose $\{x_t, y_t\}_{t \geq 0}$ is the sequence generated by Algorithm 3 under*

$$\tau_t = \frac{1}{5} \left( A(t+1)^a + \frac{5L_{xx}}{2} + \frac{13L_{yx}^2(t+1)^b}{2C} \right)^{-1} \quad \& \quad \beta_t = C(t+1)^{-b},$$

*for any $t \geq 0$ for some $A > 0, C \geq 0$ such that $\max\{C, \mu\} > 0$, and $0 < a, b < 1$. If Assumption 1.1 holds, then for any $T \geq 0$,*

$$\frac{1}{T+1} \sum_{t=0}^{T} G_{\mathcal{X}}^{\mathrm{PO}}(x_t, y_t) \leq O \left( \frac{1}{T^{(1-a)/2}} + \frac{\log^{1/2}(T+1)\,\mathbf{1}_{2+2b-2a=1} + 1}{T^{\min\{2+2b-2a,1\}/2}} + \frac{1}{T^{1/2}} + \frac{1}{T^{(a-2b)/2}} \right)$$

$$\frac{1}{T+1} \sum_{t=0}^{T} G_{\mathcal{Y}}(x_t, y_t) \leq O \left( \frac{1}{T^{1/2}} + \frac{\log^{1/2}(T+1)\,\mathbf{1}_{2a-2b=1} + 1}{T^{\min\{2a-2b,1\}/2}} + \frac{1}{T^b} \right)$$

*By optimizing over $a, b$, the optimal rate is $\widetilde{O}(T^{-1/6})$, which is obtained at $a = \frac{2}{3}, b = \frac{1}{6}$. If Assumption 1.1 additionally holds with $\mu > 0$ and $C = 0$ then*

$$\frac{1}{T+1} \sum_{t=0}^{T} G_{\mathcal{X}}^{\mathrm{PO}}(x_t, y_t) \leq O \left( \frac{1}{T^{(1-a)/2}} + \frac{1}{T^{a/2}} \right),$$

$$\frac{1}{T+1} \sum_{t=0}^{T} G_{\mathcal{Y}}(x_t, y_t) \leq O \left( \frac{1}{T^{1/2}} + \frac{\log^{1/2}(T+1)\,\mathbf{1}_{2a=1} + 1}{T^{\min\{2a,1\}/2}} \right).$$

*By optimizing over $a$, the optimal rate is $O(T^{-1/4})$, which is obtained at $a = \frac{1}{2}$.*

*Proof.* This follows directly from Lemma E.3. $\qquad\square$

The convergence of Algorithm 3 under Assumption 1.4 follows immediately from Lemma A.11 and Lemma E.3

**Theorem E.5** (Detailed version of Theorem 4.2 (i) (ii)). *Suppose $\{x_t, y_t\}_{t \geq 0}$ is the sequence generated by Algorithm 3 under*

$$\tau_t = \frac{1}{5} \left( A(t+1)^a + \frac{5L_{xx}}{2} + \frac{13L_{yx}^2(t+1)^b}{2C} \right)^{-1}, \quad \& \quad \beta_t = C(t+1)^{-b},$$

*for any $t \geq 0$ for some $A > 0, C \geq 0$ such that $\max\{C, \mu\} > 0$, and $0 < a, b < 1$. If Assumption 1.1 and Assumption 1.4 hold, then for any $T \geq 0$,*

$$\frac{1}{T+1} \sum_{t=0}^{T} G_{\mathcal{X}}^{\mathrm{Opt}}(x_{t+1}) \leq O\left(\frac{1}{T^{1/2}} + \frac{\log^{1/2}(T+1)\,\mathbf{1}_{2a-2b=1} + 1}{T^{\min\{2a-2b,1\}/2}} + \frac{1}{T^{1-a}} + \frac{1}{T^b}\right)$$

$$\frac{1}{T+1} \sum_{t=0}^{T} G_{\mathcal{Y}}(x_t, y_t) \leq O\left(\frac{1}{T^{1/2}} + \frac{\log^{1/2}(T+1)\,\mathbf{1}_{2a-2b=1} + 1}{T^{\min\{2a-2b,1\}/2}} + \frac{1}{T^b}\right).$$

*By optimizing over $a, b$, the optimal rate is $O(T^{-1/3})$, which is obtained at $a = \frac{2}{3}, b = \frac{1}{3}$. If Assumption 1.1 additionally holds with $\mu > 0$ and $C = 0$ then*

$$\frac{1}{T+1} \sum_{t=0}^{T} G_{\mathcal{X}}^{\mathrm{Opt}}(x_{t+1}) \leq O\left(\frac{1}{T^{1/2}} + \frac{\log^{1/2}(T+1)\,\mathbf{1}_{2a=1} + 1}{T^{\min\{2a,1\}/2}} + \frac{1}{T^{1-a}}\right)$$

$$\frac{1}{T+1} \sum_{t=0}^{T} G_{\mathcal{Y}}(x_t, y_t) \leq O\left(\frac{1}{T^{1/2}} + \frac{\log^{1/2}(T+1)\,\mathbf{1}_{2a=1} + 1}{T^{\min\{2a,1\}/2}}\right).$$

*By optimizing over $a$, the optimal rate is $\widetilde{O}(T^{-1/2})$, which is obtained at $a = \frac{1}{2}$.*

*Proof.* This follows directly from Lemma A.11 and Theorem E.4. □

### E.1. Strongly-convex dual domain

This section investigates accelerated rates for Algorithm 3 under Assumption 1.2. The analysis is similar to the non-strongly convex result, except that some bounds can be refined further mainly by using Lemma A.9. Specifically, Lemmas E.6 to E.8 and Theorems E.9 and E.10 mirror Lemmas E.1 to E.3 and Theorems E.4 and E.5, respectively.

**Lemma E.6.** *Suppose $\{x_t, y_t\}_{t \geq 0}$ is the sequence generated by Algorithm 3 under $\beta_t \geq \beta_{t+1} \geq 0$ such that $\max\{\beta_t, \mu\} > 0$, $\tau_t \in (0,1)$ for any $t \geq 0$. If Assumption 1.1 and Assumption 1.2 hold, then for any $t \geq 0$,*

$$\frac{1}{4} \min\left\{H_t, \frac{\alpha\sqrt{(\beta_t + \mu)H_t^3}}{4\sqrt{2}\,(L_{yy} + \beta_t)}\right\} \leq r_r - r_{t+1} + \left(\frac{1}{8\tau_t} + 2L_{xx} + \frac{4L_{yx}^2}{\beta_t + \mu}\right)\|x_{t+1} - x_t\|^2$$

$$+ \frac{D_{\mathcal{Y}}^2(\beta_t - \beta_{t+1})^2}{8L_{yx}^2 \tau_t} + \frac{4D_{\mathcal{Y}}^2(\beta_t - \beta_{t+1})^2}{\beta_{t+1} + \mu} + \frac{2^{18}L_{yx}^6(L_{yy} + \beta_{t+1})^2\tau_t^3}{3\alpha^2(\beta_{t+1} + \mu)^4}$$

*where*

$$r_t := \max\left\{\frac{H_t}{2}, H_t - \frac{\alpha\sqrt{(\beta_t + \mu)H_t^3}}{8\sqrt{2}\,(L_{yy} + \beta_t)}\right\} + \frac{1}{4} \min\left\{H_t, \frac{\alpha\sqrt{(\beta_t + \mu)H_t^3}}{4\sqrt{2}\,(L_{yy} + \beta_t)}\right\}$$

*Proof.* Using Lemmas A.2 and A.9, we obtain

$$H_{t+1} - \frac{(L_{yx}\|x_{t+1} - x_t\| + D_{\mathcal{Y}}(\beta_t - \beta_{t+1}))\sqrt{2}}{\sqrt{\beta_{t+1} + \mu}}\sqrt{H_{t+1}}$$

$$\leq \max\left\{\frac{H_t}{2}, H_t - \frac{\alpha\sqrt{(\beta_t + \mu)H_t^3}}{8\sqrt{2}\,(L_{yy} + \beta_t)}\right\} + 2L_{xx}\|x_{t+1} - x_t\|^2. \tag{37}$$

By using the inequality $ab \leq (2a^{3/2} + b^3)/3$, derived from applying the AM-GM inequality, we have

$$L_{yx}\left(\|x_{t+1} - x_t\| + D_{\mathcal{Y}}L_{yx}^{-1}(\beta_t - \beta_{t+1})\right)\sqrt{\frac{2H_{t+1}}{\beta_{t+1} + \mu}}$$

$$= \frac{\alpha(L_{yy} + \beta_{t+1})^2(\beta_{t+1} + \mu)^2}{\sqrt{2}} \cdot \frac{4L_{yx}\left(\|x_{t+1} - x_t\| + D_{\mathcal{Y}}L_{yx}^{-1}(\beta_t - \beta_{t+1})\right)}{\alpha(L_{yy} + \beta_{t+1})(\beta_{t+1} + \mu)^2} \cdot \sqrt{\frac{H_{t+1}}{4(\beta_{t+1} + \mu)(L_{yy} + \beta_{t+1})^2}}$$

$$\leq \frac{\alpha(L_{yy} + \beta_{t+1})^2(\beta_{t+1} + \mu)^2}{3\sqrt{2}} \left( \frac{H_{t+1}^{3/2}}{8(\beta_{t+1} + \mu)^{3/2}(L_{yy} + \beta_{t+1})^3} + \frac{16L_{yx}^{3/2}(\|x_{t+1} - x_t\| + D_{\mathcal{Y}}L_{yx}^{-1}(\beta_t - \beta_{t+1}))^{3/2}}{\alpha^{3/2}(L_{yy} + \beta_{t+1})^{3/2}(\beta_{t+1} + \mu)^3} \right)$$

$$= \frac{\alpha\sqrt{(\beta_{t+1} + \mu)H_{t+1}^3}}{24\sqrt{2}(L_{yy} + \beta_{t+1})} + \frac{8\sqrt{2}L_{yx}^{3/2}(L_{yy} + \beta_{t+1})^{1/2}(\|x_{t+1} - x_t\| + D_{\mathcal{Y}}L_{yx}^{-1}(\beta_t - \beta_{t+1}))^{3/2}}{3\alpha^{1/2}(\beta_{t+1} + \mu)}.$$

By using the inequality $a^{3/2}b \leq (3a^2 + b^4)/4$, derived from applying the AM-GM inequality, we have

$$\frac{8\sqrt{2}L_{yx}^{3/2}(L_{yy} + \beta_{t+1})^{1/2}(\|x_{t+1} - x_t\| + D_{\mathcal{Y}}L_{yx}^{-1}(\beta_t - \beta_{t+1}))^{3/2}}{3\alpha^{1/2}(\beta_{t+1} + \mu)}$$

$$= \frac{4}{3} \frac{16\sqrt{2}L_{yx}^{3/2}(L_{yy} + \beta_{t+1})^{1/2}\tau_t^{3/4}}{\alpha^{1/2}(\beta_{t+1} + \mu)} \left( \left( \frac{\|x_{t+1} - x_t\| + D_{\mathcal{Y}}L_{yx}^{-1}(\beta_t - \beta_{t+1})}{4\sqrt{\tau_t}} \right)^{1/2} \right)^3$$

$$\leq \frac{(\|x_{t+1} - x_t\| + D_{\mathcal{Y}}L_{yx}^{-1}(\beta_t - \beta_{t+1}))^2}{16\tau_t} + \frac{2^{18}L_{yx}^6(L_{yy} + \beta_{t+1})^2\tau_t^3}{3\alpha^2(\beta_{t+1} + \mu)^4}$$

$$\leq \frac{\|x_{t+1} - x_t\|^2}{8\tau_t} + \frac{D_{\mathcal{Y}}^2 L_{yx}^{-2}(\beta_t - \beta_{t+1})^2}{8\tau_t} + \frac{2^{18}L_{yx}^6(L_{yy} + \beta_{t+1})^2\tau_t^3}{3\alpha^2(\beta_{t+1} + \mu)^4}.$$

Thus, we have

$$L_{yx}\left(\|x_{t+1} - x_t\| + D_{\mathcal{Y}}L_{yx}^{-1}(\beta_t - \beta_{t+1})\right)\sqrt{\frac{2H_{t+1}}{\beta_{t+1} + \mu}}$$

$$\leq \frac{\alpha\sqrt{(\beta_{t+1} + \mu)H_{t+1}^3}}{24\sqrt{2}(L_{yy} + \beta_{t+1})} + \frac{\|x_{t+1} - x_t\|^2}{8\tau_t} + \frac{D_{\mathcal{Y}}^2 L_{yx}^{-2}(\beta_t - \beta_{t+1})^2}{8\tau_t} + \frac{2^{18}L_{yx}^6(L_{yy} + \beta_{t+1})^2\tau_t^3}{3\alpha^2(\beta_{t+1} + \mu)^4} \tag{38}$$

If

$$\frac{H_{t+1}}{2} \leq H_{t+1} - \frac{\alpha\sqrt{(\beta_{t+1} + \mu)H_{t+1}^3}}{8\sqrt{2}\left(L_{yy} + \beta_{t+1}\right)} \iff \frac{\alpha\sqrt{(\beta_{t+1} + \mu)H_{t+1}^3}}{4\sqrt{2}\left(L_{yy} + \beta_{t+1}\right)} \leq H_{t+1},$$

then

$$r_{t+1} = H_{t+1} - \frac{\alpha\sqrt{(\beta_{t+1} + \mu)H_{t+1}^3}}{16\sqrt{2}(L_{yy} + \beta_{t+1})} \leq H_{t+1} - \frac{\alpha\sqrt{(\beta_{t+1} + \mu)H_{t+1}^3}}{24\sqrt{2}(L_{yy} + \beta_{t+1})}.$$

Combining (37) and (38) gives us

$$r_{t+1} \leq r_t - \frac{1}{4}\min\left\{ H_t, \frac{\alpha\sqrt{(\beta_t + \mu)H_t^3}}{4\sqrt{2}\left(L_{yy} + \beta_t\right)} \right\} + \left( 2L_{xx} + \frac{1}{8\tau_t} \right)\|x_{t+1} - x_t\|^2$$

$$+ \frac{D_{\mathcal{Y}}^2(\beta_t - \beta_{t+1})^2}{8L_{yx}^2\tau_t} + \frac{2^{18}L_{yx}^6(L_{yy} + \beta_{t+1})^2\tau_t^3}{3\alpha^2(\beta_{t+1} + \mu)^4}.$$

By AM-GM inequality for two numbers, we have

$$\frac{2\left(L_{yx}\|x_{t+1} - x_t\| + D_{\mathcal{Y}}(\beta_t - \beta_{t+1})\right)}{\sqrt{\beta_{t+1} + \mu}}\sqrt{\frac{H_{t+1}}{2}} \leq \frac{H_{t+1}}{4} + \frac{2\left(L_{yx}\|x_{t+1} - x_t\| + D_{\mathcal{Y}}(\beta_t - \beta_{t+1})\right)^2}{\beta_{t+1} + \mu}$$

$$\leq \frac{H_{t+1}}{4} + \frac{4L_{yx}^2\|x_{t+1} - x_t\|^2 + 4D_{\mathcal{Y}}^2(\beta_t - \beta_{t+1})^2}{\beta_{t+1} + \mu}, \tag{39}$$

If

$$\frac{H_{t+1}}{2} > H_{t+1} - \frac{\alpha\sqrt{(\beta_{t+1} + \mu)H_{t+1}^3}}{8\sqrt{2}\left(L_{yy} + \beta_{t+1}\right)} \iff \frac{\alpha\sqrt{(\beta_{t+1} + \mu)H_{t+1}^3}}{4\sqrt{2}\left(L_{yy} + \beta_{t+1}\right)} > H_{t+1},$$

then

$$r_{t+1} = \frac{3H_{t+1}}{4} = H_{t+1} - \frac{H_{t+1}}{4}.$$

Combining (37) and (39) gives us

$$r_{t+1} \leq r_t - \frac{1}{4} \min \left\{ H_t, \frac{\alpha \sqrt{(\beta_t + \mu)H_t^3}}{4\sqrt{2}\,(L_{yy} + \beta_t)} \right\} + \left( 2L_{xx} + \frac{4L_{yx}^2}{\beta_t + \mu} \right) \|x_{t+1} - x_t\|^2 + \frac{4D_{\mathcal{Y}}^2(\beta_t - \beta_{t+1})^2}{\beta_{t+1} + \mu}.$$

Hence, we conclude the proof. $\qquad\square$

**Lemma E.7.** *Suppose $\{x_t, y_t\}_{t\geq 0}$ is the sequence generated by Algorithm 3 under $\beta_t \geq \beta_{t+1} \geq 0$ such that $\max\{\beta_t, \mu\} > 0$, $\tau_t \geq \tau_{t+1} > 0$ for any $t \geq 0$. If Assumptions 1.1 and 1.2 hold, then for any $t \geq 0$,*

$$\left( \frac{13}{16\tau_t} - \frac{5L_{xx}}{2} - \frac{13L_{yx}^2}{2(\beta_t + \mu)} \right) \|x_{t+1} - x_t\|^2$$

$$\leq q_t - q_{t+1} + \frac{D_{\mathcal{Y}}^2(\beta_t - \beta_{t+1})^2}{8L_{yx}^2\tau_t} + \frac{4D_{\mathcal{Y}}^2(\beta_t - \beta_{t+1})^2}{\beta_{t+1} + \mu} + \frac{2^{19}L_{yx}^6(L_{yy} + \beta_{t+1})^2\tau_t^3}{3\alpha^2(\beta_{t+1} + \mu)^4}.$$
(40)

*where for any $t \geq 0$*

$$q_t := f_t(x_t) + \frac{D_{\mathcal{Y}}^2\beta_t}{2} + r_t.$$

*Hence, if it additionally holds that*

$$G_2 := \inf_{t\geq 0} \left\{ \frac{13}{16} - \frac{5L_{xx}\tau_t}{2} - \frac{13L_{yx}^2\tau_t}{2(\beta_t + \mu)} \right\} > 0,$$

*then*

$$\sum_{t=0}^{T} \left\| \frac{x_{t+1} - x_t}{\tau_t} \right\|^2 \leq \frac{2\sup_{t\geq 0}|q_t|}{G_2\tau_T} + \sum_{t=0}^{T} \left( \frac{D_{\mathcal{Y}}^2(\beta_t - \beta_{t+1})^2}{8G_2L_{yx}^2\tau_t^2} + \frac{4D_{\mathcal{Y}}^2(\beta_t - \beta_{t+1})^2}{G_2(\beta_{t+1} + \mu)\tau_t} + \frac{2^{19}L_{yx}^6(L_{yy} + \beta_{t+1})^2\tau_t^2}{3\alpha^2G_2(\beta_{t+1} + \mu)^4} \right).$$

*Proof.* By AM-GM inequality for three numbers, we have

$$L_{yx}\|x_{t+1} - x_t\|\sqrt{\frac{2H_t}{\beta_t + \mu}} = \frac{\alpha(L_{yy} + \beta_t)^2(\beta_t + \mu)^2}{\sqrt{2}} \frac{4L_{yx}\|x_{t+1} - x_t\|}{\alpha(L_{yy} + \beta_t)(\beta_t + \mu)^2} \sqrt{\frac{H_t}{4(\beta_t + \mu)(L_{yy} + \beta_t)^2}}$$

$$\leq \frac{\alpha(L_{yy} + \beta_t)^2(\beta_t + \mu)^2}{3\sqrt{2}} \left( \frac{H_t^{3/2}}{8(\beta_t + \mu)^{3/2}(L_{yy} + \beta_t)^3} + \frac{16L_{yx}^{3/2}(\|x_{t+1} - x_t\|)^{3/2}}{\alpha^{3/2}(L_{yy} + \beta_t)^{3/2}(\beta_t + \mu)^3} \right)$$

$$= \frac{\alpha\sqrt{(\beta_t + \mu)H_t^3}}{24\sqrt{2}(L_{yy} + \beta_t)} + \frac{8\sqrt{2}L_{yx}^{3/2}(L_{yy} + \beta_t)^{1/2}(\|x_{t+1} - x_t\|)^{3/2}}{3\alpha^{1/2}(\beta_t + \mu)}$$

$$\leq \frac{\alpha\sqrt{(\beta_t + \mu)H_t^3}}{16\sqrt{2}(L_{yy} + \beta_t)} + \frac{8\sqrt{2}L_{yx}^{3/2}(L_{yy} + \beta_t)^{1/2}(\|x_{t+1} - x_t\|)^{3/2}}{3\alpha^{1/2}(\beta_t + \mu)}.$$

By AM-GM inequality for four numbers, we also have

$$\frac{8\sqrt{2}L_{yx}^{3/2}(L_{yy} + \beta_t)^{1/2}(\|x_{t+1} - x_t\|)^{3/2}}{3\alpha^{1/2}(\beta_t + \mu)} = \frac{4}{3} \frac{16\sqrt{2}L_{yx}^{3/2}(L_{yy} + \beta_t)^{1/2}\tau_t^{3/4}}{\alpha^{1/2}(\beta_t + \mu)} \left( \left( \frac{\|x_{t+1} - x_t\|}{4\sqrt{\tau_t}} \right)^{1/2} \right)^3$$

$$\leq \frac{\|x_{t+1} - x_t\|^2}{16\tau_t} + \frac{2^{18}L_{yx}^6(L_{yy} + \beta_t)^2\tau_t^3}{3\alpha^2(\beta_t + \mu)^4}.$$

By AM-GM inequality for two numbers, we have

$$\frac{2L_{yx}\|x_{t+1} - x_t\|}{\sqrt{\beta_t + \mu}}\sqrt{\frac{H_t}{2}} \leq \frac{H_t}{4} + \frac{2L_{yx}^2\|x_{t+1} - x_t\|^2}{\beta_t + \mu}.$$

As a result, we have

$$L_{yx}\|x_{t+1} - x_t\|\sqrt{\frac{2H_t}{\beta_t + \mu}}$$

$$\leq \frac{1}{4}\min\left\{H_t, \frac{\alpha\sqrt{(\beta_t + \mu)H_t^3}}{4\sqrt{2}(L_{yy} + \beta_t)}\right\} + \left(\frac{1}{16\tau_t} + \frac{2L_{yx}^2}{\beta_t + \mu}\right)\|x_{t+1} - x_t\|^2 + \frac{2^{18}L_{yx}^6(L_{yy} + \beta_t)^2\tau_t^3}{3\alpha^2(\beta_t + \mu)^4}$$

$$\leq r_t - r_{t+1} + \left(\frac{3}{16\tau_t} + 2L_{xx} + \frac{6L_{yx}^2}{\beta_t + \mu}\right)\|x_{t+1} - x_t\|^2$$

$$+ \frac{D_{\mathcal{Y}}^2(\beta_t - \beta_{t+1})^2}{8L_{yx}^2\tau_t} + \frac{4D_{\mathcal{Y}}^2(\beta_t - \beta_{t+1})^2}{\beta_{t+1} + \mu} + \frac{2^{19}L_{yx}^6(L_{yy} + \beta_{t+1})^2\tau_t^3}{3\alpha^2(\beta_{t+1} + \mu)^4},$$

where the last inequality follows from Lemma E.6. Recall from (18) that we have

$$\left(\frac{1}{\tau_t} - \frac{L_{f_t}}{2}\right)\|x_{t+1} - x_t\|^2$$

$$\leq L_{yx}\|x_{t+1} - x_t\|\sqrt{\frac{2H_t}{\beta_t + \mu}} + f_t(x_t) + \frac{D_{\mathcal{Y}}^2\beta_t}{2} - \left(f_{t+1}(x_{t+1}) + \frac{D_{\mathcal{Y}}^2\beta_{t+1}}{2}\right).$$

We deduce the first part

$$\left(\frac{13}{16\tau_t} - \frac{5L_{xx}}{2} - \frac{13L_{yx}^2}{2(\beta_t + \mu)}\right)\|x_{t+1} - x_t\|^2$$

$$\leq q_t - q_{t+1} + \frac{D_{\mathcal{Y}}^2(\beta_t - \beta_{t+1})^2}{8L_{yx}^2\tau_t} + \frac{4D_{\mathcal{Y}}^2(\beta_t - \beta_{t+1})^2}{\beta_{t+1} + \mu} + \frac{2^{19}L_{yx}^6(L_{yy} + \beta_{t+1})^2\tau_t^3}{3\alpha^2(\beta_{t+1} + \mu)^4}.$$

By dividing both sides of (40) by $\tau_t$ and summing the inequality for $t = 0, \cdots, T$, we obtain

$$\sum_{t=0}^{T}\left(\frac{13}{16} - \frac{5L_{xx}\tau_t}{2} - \frac{13L_{yx}^2\tau_t}{2(\beta_t + \mu)}\right)\left\|\frac{x_{t+1} - x_t}{\tau_t}\right\|^2$$

$$\leq \frac{q_0}{\tau_0} + \sum_{t\in[T]}\left(\frac{1}{\tau_t} - \frac{1}{\tau_{t-1}}\right)q_t - \frac{q_{T+1}}{\tau_T} + \sum_{t=0}^{T}\frac{D_{\mathcal{Y}}^2(\beta_t - \beta_{t+1})^2}{8L_{yx}^2\tau_t^2} + \frac{4D_{\mathcal{Y}}^2(\beta_t - \beta_{t+1})^2}{(\beta_{t+1} + \mu)\tau_t} + \frac{2^{19}L_{yx}^6(L_{yy} + \beta_{t+1})^2\tau_t^2}{3\alpha^2(\beta_{t+1} + \mu)^4}$$

$$\leq \left(\frac{1}{\tau_0} + \sum_{t\in[T]}\left(\frac{1}{\tau_t} - \frac{1}{\tau_{t-1}}\right) + \frac{1}{\tau_T}\right)\sup_{t\geq 0}|q_t| + \sum_{t=0}^{T}\frac{D_{\mathcal{Y}}^2(\beta_t - \beta_{t+1})^2}{8L_{yx}^2\tau_t^2} + \frac{4D_{\mathcal{Y}}^2(\beta_t - \beta_{t+1})^2}{(\beta_{t+1} + \mu)\tau_t} + \frac{2^{19}L_{yx}^6(L_{yy} + \beta_{t+1})^2\tau_t^2}{3\alpha^2(\beta_{t+1} + \mu)^4}$$

$$\leq \frac{2\sup_{t\geq 0}|q_t|}{\tau_T} + \sum_{t=0}^{T}\frac{D_{\mathcal{Y}}^2(\beta_t - \beta_{t+1})^2}{8L_{yx}^2\tau_t^2} + \frac{4D_{\mathcal{Y}}^2(\beta_t - \beta_{t+1})^2}{(\beta_{t+1} + \mu)\tau_t} + \frac{2^{19}L_{yx}^6(L_{yy} + \beta_{t+1})^2\tau_t^2}{3\alpha^2(\beta_{t+1} + \mu)^4},$$

which concludes the proof of the second part. $\square$

**Lemma E.8.** *Suppose $\{x_t, y_t\}_{t\geq 0}$ is the sequence generated by Algorithm 3 under $\beta_t \geq \beta_{t+1} \geq 0$ such that $\max\{\beta_t, \mu\} > 0$, $\tau_t \geq \tau_{t+1} > 0$ for any $t \geq 0$. If Assumptions 1.1 and 1.2 hold, then for any $t \geq 0$,*

$$\frac{1}{T+1}\sum_{t=0}^{T}G_{\mathcal{X}}^{\text{PO}}(x_t, y_t)$$

$$\leq \left(\frac{2\sup_{t\geq 0}|q_t|}{G_2(T+1)\tau_T} + \frac{1}{T+1}\sum_{t=0}^{T}\left(\frac{D_{\mathcal{Y}}^2(\beta_t - \beta_{t+1})^2}{8G_2L_{yx}^2\tau_t^2} + \frac{4D_{\mathcal{Y}}^2(\beta_t - \beta_{t+1})^2}{G_2(\beta_{t+1} + \mu)\tau_t} + \frac{2^{19}L_{yx}^6(L_{yy} + \beta_{t+1})^2\tau_t^2}{3\alpha^2G_2(\beta_{t+1} + \mu)^4}\right)\right)^{1/2},$$

$$\frac{1}{T+1}\sum_{t=0}^{T}G_{\mathcal{Y}}(x_t, y_t)$$

$$\leq \frac{D_{\mathcal{Y}}^2}{2(T+1)} \sum_{t=0}^{T} \beta_t + \left( \frac{2F_2 \sup_{t\geq 0}|s_t|}{(T+1)(\beta_T+\mu)^{1/2}} \right)^{2/3}$$

$$+ \left( \frac{F_2}{T+1} \sum_{t=0}^{T} \left( (1+U_2)D_{\mathcal{Y}}^2 \left( \frac{(\beta_t-\beta_{t+1})^2}{8L_{yx}^2 \tau_t (\beta_t+\mu)^{1/2}} + \frac{4(\beta_t-\beta_{t+1})^2}{(\beta_{t+1}+\mu)^{3/2}} \right) + \frac{2^{18}(1+2U_2)L_{yx}^6(L_{yy}+\beta_{t+1})^2 \tau_t^3}{3\alpha^2(\beta_{t+1}+\mu)^{9/2}} \right) \right)^{2/3},$$

*where for any $t \geq 0$*

$$U_2 := \sup_{t\geq 0} \frac{\frac{1}{8\tau_t} + 2L_{xx} + \frac{4L_{yx}^2}{\beta_t+\mu}}{\frac{13}{16\tau_t} - \frac{5L_{xx}}{2} - \frac{13L_{yx}^2}{2(\beta_t+\mu)}}, \quad \& \quad F_2 := \sup_{t\geq 0} \max \left\{ \frac{4\sqrt{2}(L_{yy}+\beta_t)}{\alpha}, \sqrt{(\beta_t+\mu)H_t} \right\}, \quad \& \quad s_t := r_t + U_2 q_t.$$

*Proof.* The first claim follows immediately from Lemma E.2 and the convexity of the function $\|\cdot\|^2$. Turning to the second claim, we have from the definition of $F_2$ that

$$\min \left\{ H_t, \frac{\alpha\sqrt{(\beta_t+\mu)H_t^3}}{4\sqrt{2}\,(L_{yy}+\beta_t)} \right\} \geq \min \left\{ H_t, \frac{\sqrt{(\beta_t+\mu)H_t}}{F_2} H_t \right\} = \frac{\sqrt{(\beta_t+\mu)H_t^3}}{F_2}$$

From Lemma E.6 and Lemma E.7, we have

$$\frac{\sqrt{(\beta_t+\mu)H_t^3}}{F_2}$$

$$\leq r_r - r_{t+1} + U_2 \left( q_t - q_{t+1} + \frac{D_{\mathcal{Y}}^2(\beta_t-\beta_{t+1})^2}{8L_{yx}^2 \tau_t} + \frac{4D_{\mathcal{Y}}^2(\beta_t-\beta_{t+1})^2}{\beta_{t+1}+\mu} + \frac{2^{19}L_{yx}^6(L_{yy}+\beta_{t+1})^2 \tau_t^3}{3\alpha^2(\beta_{t+1}+\mu)^4} \right)$$

$$+ \frac{D_{\mathcal{Y}}^2(\beta_t-\beta_{t+1})^2}{8L_{yx}^2 \tau_t} + \frac{4D_{\mathcal{Y}}^2(\beta_t-\beta_{t+1})^2}{\beta_{t+1}+\mu} + \frac{2^{18}L_{yx}^6(L_{yy}+\beta_{t+1})^2 \tau_t^3}{3\alpha^2(\beta_{t+1}+\mu)^4},$$

which implies

$$\frac{H_t^{3/2}}{F_2} \leq \frac{s_t - s_{t+1}}{(\beta_t+\mu)^{1/2}} + (1+U_2)D_{\mathcal{Y}}^2 \left( \frac{(\beta_t-\beta_{t+1})^2}{8L_{yx}^2 \tau_t (\beta_t+\mu)^{1/2}} + \frac{4(\beta_t-\beta_{t+1})^2}{(\beta_{t+1}+\mu)^{3/2}} \right) + \frac{2^{18}(1+2U_2)L_{yx}^6(L_{yy}+\beta_{t+1})^2 \tau_t^3}{3\alpha^2(\beta_{t+1}+\mu)^{9/2}}.$$

By summing the inequality over $t = 0, \ldots, T$, we have

$$\sum_{t=0}^{T} \frac{H_t^{3/2}}{F_2} \leq \frac{s_0}{(\beta_0+\mu)^{1/2}} + \sum_{t\in[T]} \left( \frac{1}{(\beta_t+\mu)^{1/2}} - \frac{1}{(\beta_{t-1}+\mu)^{1/2}} \right) s_t - \frac{s_{T+1}}{(\beta_T+\mu)^{1/2}}$$

$$+ \sum_{t=0}^{T} \left( (1+U_2)D_{\mathcal{Y}}^2 \left( \frac{(\beta_t-\beta_{t+1})^2}{8L_{yx}^2 \tau_t (\beta_t+\mu)^{1/2}} + \frac{4(\beta_t-\beta_{t+1})^2}{(\beta_{t+1}+\mu)^{3/2}} \right) + \frac{2^{18}(1+2U_2)L_{yx}^6(L_{yy}+\beta_{t+1})^2 \tau_t^3}{3\alpha^2(\beta_{t+1}+\mu)^{9/2}} \right).$$

Using the convexity of $(\cdot)^{3/2}$, Lemma A.6, and the fact that

$$\frac{s_0}{(\beta_0+\mu)^{1/2}} + \sum_{t\in[T]} \left( \frac{1}{(\beta_t+\mu)^{1/2}} - \frac{1}{(\beta_{t-1}+\mu)^{1/2}} \right) s_t - \frac{s_{T+1}}{(\beta_T+\mu)^{1/2}}$$

$$\leq \left( \frac{1}{(\beta_0+\mu)^{1/2}} + \sum_{t\in[T]} \left( \frac{1}{(\beta_t+\mu)^{1/2}} - \frac{1}{(\beta_{t-1}+\mu)^{1/2}} \right) + \frac{1}{(\beta_T+\mu)^{1/2}} \right) \sup_{t\geq 0} |s_t|$$

$$= \frac{2\sup_{t\geq 0}|s_t|}{(\beta_T+\mu)^{1/2}},$$

we conclude the proof of the second part. $\qquad\square$

**Theorem E.9** (Detailed version of Theorem 4.2 (iii), (iv)). *Suppose $\{x_t, y_t\}_{t \geq 0}$ is the sequence generated by Algorithm 3 under*

$$\tau_t = \frac{3}{4}\left(A(t+1)^a + \frac{5L_{xx}}{2} + \frac{13L_{yx}^2(t+1)^b}{2C}\right)^{-1}, \quad \& \quad \beta_t = C(t+1)^{-b},$$

*for any $t \geq 0$ for some $A > 0, C \geq 0$ such that $\max\{C, \mu\} > 0$, and $0 < a, b < 1$. If Assumptions 1.1 and 1.2 hold, then for any $t \geq 0$,*

$$\frac{1}{T+1}\sum_{t=0}^{T} G_{\mathcal{X}}^{\mathrm{PO}}(x_t, y_t) \leq O\left(\frac{1}{T^{(1-a)/2}} + \frac{\log^{1/2}(T+1)\,\mathbf{1}_{2+2b-2a=1}+1}{T^{\min\{2+2b-2a,1\}/2}} + \frac{1}{T^{1/2}} + \frac{\log^{1/2}(T+1)\,\mathbf{1}_{2a-4b=1}+1}{T^{\min\{2a-4b,1\}/2}}\right),$$

$$\frac{1}{T+1}\sum_{t=0}^{T} G_{\mathcal{Y}}(x_t, y_t) \leq O\left(\frac{1}{T^{b}} + \frac{1}{T^{(2-b)/3}} + \frac{1}{T^{2/3}} + \frac{\log^{2/3}(T+1)\,\mathbf{1}_{6a-9b=2}+1}{T^{\min\{6a-9b,2\}/3}}\right).$$

*By optimizing over $a, b$, the optimal rate is $O(T^{-1/5})$, which is obtained at $a = \frac{3}{5}, b = \frac{1}{5}$. If Assumption 1.1 additionally holds with $\mu > 0$ and $C = 0$ then*

$$\frac{1}{T+1}\sum_{t=0}^{T} G_{\mathcal{X}}^{\mathrm{PO}}(x_t, y_t) \leq O\left(\frac{1}{T^{(1-a)/2}} + \frac{\log^{1/2}(T+1)\,\mathbf{1}_{2a=1}+1}{T^{\min\{2a,1\}/2}}\right)$$

$$\frac{1}{T+1}\sum_{t=0}^{T} G_{\mathcal{Y}}(x_t, y_t) \leq O\left(\frac{1}{T^{2/3}} + \frac{\log^{2/3}(T+1)\,\mathbf{1}_{6a=2}+1}{T^{\min\{6a,2\}/3}}\right).$$

*By optimizing over $a$, the optimal rate is $O(T^{-1/3})$, which is obtained at $a = \frac{1}{3}$.*

*Proof.* This follows directly from Lemma E.8. $\qquad\square$

**Theorem E.10** (Detailed version of Theorem 4.2 (iii)). *Suppose $\{x_t, y_t\}_{t \geq 0}$ is the sequence generated by Algorithm 3 under*

$$\tau_t = \frac{3}{4}\left(A(t+1)^a + \frac{5L_{xx}}{2} + \frac{13L_{yx}^2(t+1)^b}{2C}\right)^{-1}, \quad \& \quad \beta_t = C(t+1)^{-b},$$

*for any $t \geq 0$ for some $A > 0, C \geq 0$ such that $\max\{C, \mu\} > 0$, and $0 < a, b < 1$. If Assumption 1.1 additionally holds with $\mu > 0$ and $C = 0$ then*

$$\frac{1}{T+1}\sum_{t=0}^{T} G_{\mathcal{X}}^{\mathrm{Opt}}(x_{t+1}) \leq O\left(\frac{1}{T^{2/3}} + \frac{\log^{2/3}(T+1)\,\mathbf{1}_{6a=2}+1}{T^{\min\{6a,2\}/3}} + \frac{1}{T^{1-a}}\right)$$

*By optimizing over $a$, the optimal rate is $\widetilde{O}(T^{-2/3})$, which is obtained at $a = \frac{1}{3}$.*

*Proof.* This follows directly from Lemma A.11 and Theorem E.9. $\qquad\square$

*Remark* E.11. Following Equation (15) in the proof of Lemma A.8, Algorithm 3 can utilize a local Lipschitz estimate $L_{yy}^{(t)}$ at each iteration $t$. It is sufficient that $L_{yy}^{(t)}$ satisfies (29), a condition that can be ensured via the backtracking line-search procedure in (Pedregosa et al., 2020).

## F. Auxiliary Lemmas

**Lemma F.1** (Chebyshev's rearrangement inequality). *Suppose there are two sequences of real numbers $a_1 \leq \cdots \leq a_t$ and $b_1 \geq \cdots \geq b_t$. It holds that*

$$t\sum_{i\in[n]} a_i b_i \leq \left(\sum_{i\in[t]} a_i\right)\left(\sum_{i\in[t]} b_i\right).$$

*Proof.* Since one sequence is increasing while the other is decreasing, one have

$$(a_i - a_j)(b_i - b_j) \leq 0 \iff a_i b_i + a_j b_j \leq a_i b_j + a_j b_i.$$

for any $i, j \in [t]$. By adding $t^2$ inequalities, we obtain

$$2t \sum_{i \in [t]} a_i b_i = \sum_{i \in [t]} \sum_{j \in [t]} (a_i b_i + a_j b_j) \leq \sum_{i \in [t]} \sum_{j \in [t]} (a_i b_j + a_j b_i) = 2 \left( \sum_{i \in [t]} a_i \right) \left( \sum_{i \in [t]} b_i \right),$$

which concludes the proof. $\qquad\square$

**Lemma F.2.** *Given $\alpha, k > 0$ and $t \geq 0$, it holds that $(t + k)^{-\alpha} - (t + 1 + k)^{-\alpha} \leq \alpha t^{-(\alpha+1)}$.*

*Proof.* From mean value theorem, there exists $t < c_t < t + 1$ such that

$$
\begin{aligned}
(t + k)^{-\alpha} - (t + 1 + k)^{-\alpha} &= -\alpha(c_t + k)^{-(\alpha+1)}(t + k - (t + 1 + k)) \\
&= \alpha(c_t + k)^{-(\alpha+1)} \\
&\leq \alpha(t + k)^{-(\alpha+1)}.
\end{aligned}
$$

This concludes the proof. $\qquad\square$

**Lemma F.3.** *Given $s \geq 0$, it holds that for any $T \geq 1$, if $s \neq 1$*

$$\frac{(T + 1)^{1-s} - 1}{1 - s} \leq \sum_{t \in [T]} \frac{1}{t^s} \leq 1 + \frac{T^{1-s} - 1}{1 - s} \leq \frac{T^{1-s}}{1 - s},$$

*and otherwise,*

$$\log(T + 1) \leq \sum_{t \in [T]} \frac{1}{t} \leq \log(T) + 1.$$

*Proof.* For any $t \geq 2$, we have

$$\int_t^{t+1} \frac{dr}{r^s} \leq \frac{1}{t^s} \leq \int_{t-1}^t \frac{dr}{r^s}.$$

We note that the left-most inequality also holds for $t = 1$. Hence, we have

$$\int_1^{T+1} \frac{dr}{r^s} \leq \sum_{t \in [T]} \frac{1}{t^s} \leq 1 + \int_1^T \frac{dr}{r^s}.$$

By considering two cases: $s \neq 1$ and $s = 1$, we finish the proof. $\qquad\square$

**Lemma F.4.** *Given $\gamma \in (0, 1]$, $2a > b$, $a, b \in (0, 1]$, it holds that for any $t \geq \lfloor \frac{2}{\gamma} \rfloor$,*

$$\sum_{i \in [t]} \left(1 - \frac{\gamma}{2}\right)^{-i} \frac{1}{i^{2a-b}} \leq \frac{2}{\gamma} \left(1 - \frac{\gamma}{2}\right)^{-\lfloor \frac{2}{\gamma} \rfloor + 1} + \left(1 - \frac{\gamma}{2}\right)^{-t} \frac{8E(a, b)}{\gamma t^{2a-b}},$$

$$\sum_{i \in [t]} \left(1 - \frac{\gamma}{2}\right)^{-i} \frac{1}{i^{2a}} \leq \frac{2}{\gamma} \left(1 - \frac{\gamma}{2}\right)^{-\lfloor \frac{2}{\gamma} \rfloor} + \left(1 - \frac{\gamma}{2}\right)^{-t} \frac{8}{\gamma} \cdot \begin{cases} \frac{E(a,0)}{t^{2a}} & 0 < a < 1 \\ \frac{\log(t)}{t^2} & a = 1 \end{cases},$$

*where $E(a, b) := \begin{cases} \frac{1}{1-2a+b} & 2a - b < 1 \\ \frac{1}{2-2a+b} & 2a - b \geq 1 \end{cases}$.*

*Proof.* We only prove the first part since the second one can be proved with the same strategy. If $2a - b \geq 1$, we break it down into

$$\sum_{i \in \left[\lfloor \frac{2}{\gamma} \rfloor - 1\right]} \left(1 - \frac{\gamma}{2}\right)^{-i} i^{-(2a-b)} + \sum_{\lfloor \frac{2}{\gamma} \rfloor \leq i \leq t} \left(1 - \frac{\gamma}{2}\right)^{-i} i^{-(2a-b)}.$$

For the first component, we have

$$\sum_{i \in \left[\lfloor \frac{2}{\gamma} \rfloor - 1\right]} \left(1 - \frac{\gamma}{2}\right)^{-i} i^{-(2a-b)} \leq \sum_{i \in \left[\lfloor \frac{2}{\gamma} \rfloor - 1\right]} \left(1 - \frac{\gamma}{2}\right)^{-i} \leq \frac{2}{\gamma} \left(1 - \frac{\gamma}{2}\right)^{-\lfloor \frac{2}{\gamma} \rfloor + 1}.$$

For the second component we observe that $a_i := \left(1 - \frac{\gamma}{2}\right)^{-i} \frac{1}{i}$ with $\lfloor \frac{2}{\gamma} \rfloor \leq i \leq t$ form an increasing sequence since

$$\frac{a_{i+1}}{a_i} = \frac{1}{1 - \frac{\gamma}{2}} \frac{i}{i+1} \geq 1 \iff 1 - \frac{1}{i+1} \geq 1 - \frac{\gamma}{2} \iff i + 1 \geq \frac{2}{\gamma}.$$

Using Lemma F.1 with

$$a_i := \left(1 - \frac{\gamma}{2}\right)^{-i} \frac{1}{i}, \quad b_i := \frac{1}{i^{2a-b-1}},$$
$$a_i := \left(1 - \frac{\gamma}{2}\right)^{-i} \frac{1}{i^{1/2}}, \quad b_i := \frac{1}{i^{1/2}},$$
$$a_i := \left(1 - \frac{\gamma}{2}\right)^{-i}, \quad b_i := \frac{1}{i^{1/2}},$$

sequentially, we have

$$\sum_{\lfloor \frac{2}{\gamma} \rfloor \leq i \leq t} \left(1 - \frac{\gamma}{2}\right)^{-i} \frac{1}{i^{2a-b}} \leq \frac{1}{t - \lfloor \frac{2}{\gamma} \rfloor + 1} \left(\sum_{\lfloor \frac{2}{\gamma} \rfloor \leq i \leq t} \left(1 - \frac{\gamma}{2}\right)^{-i} \frac{1}{i}\right) \left(\sum_{\lfloor \frac{2}{\gamma} \rfloor \leq i \leq t} \frac{1}{i^{2a-b-1}}\right)$$

$$\leq \frac{1}{\left(t - \lfloor \frac{2}{\gamma} \rfloor + 1\right)^2} \left(\sum_{\lfloor \frac{2}{\gamma} \rfloor \leq i \leq t} \left(1 - \frac{\gamma}{2}\right)^{-i} \frac{1}{i^{1/2}}\right) \left(\sum_{\lfloor \frac{2}{\gamma} \rfloor \leq i \leq t} \frac{1}{i^{1/2}}\right) \left(\sum_{\lfloor \frac{2}{\gamma} \rfloor \leq i \leq t} \frac{1}{i^{2a-b-1}}\right)$$

$$\leq \frac{1}{\left(t - \lfloor \frac{2}{\gamma} \rfloor + 1\right)^3} \left(\sum_{\lfloor \frac{2}{\gamma} \rfloor \leq i \leq t} \left(1 - \frac{\gamma}{2}\right)^{-i}\right) \left(\sum_{\lfloor \frac{2}{\gamma} \rfloor \leq i \leq t} \frac{1}{i^{1/2}}\right)^2 \left(\sum_{\lfloor \frac{2}{\gamma} \rfloor \leq i \leq t} \frac{1}{i^{2a-b-1}}\right).$$

Using the fact that

$$\sum_{\lfloor \frac{2}{\gamma} \rfloor \leq i \leq t} \left(1 - \frac{\gamma}{2}\right)^{-i} \leq \sum_{i \in [t]} \left(1 - \frac{\gamma}{2}\right)^{-i} \leq \frac{2}{\gamma} \left(1 - \frac{\gamma}{2}\right)^{-t},$$

and for any $2 \leq k \leq t$,

$$\frac{1}{t - k + 1} \sum_{i=k}^{t} \frac{1}{i^s} \leq \frac{1}{t - k + 1} \int_{z=k-1}^{t} \frac{dz}{z^s} \leq \frac{t^{1-s} - (k-1)^{1-s}}{(1-s)(t-k+1)} \leq \frac{t^{1-s} - (k-1)t^{-s}}{(1-s)(t-k+1)} = \frac{1}{(1-s)t^s},$$

we have

$$\sum_{\lfloor \frac{2}{\gamma} \rfloor \leq i \leq t} \left(1 - \frac{\gamma}{2}\right)^{-i} \frac{1}{i^{2a-b}} \leq \frac{2}{\gamma} \left(1 - \frac{\gamma}{2}\right)^{-t} \left(\frac{1}{t - \lfloor \frac{2}{\gamma} \rfloor + 1} \sum_{\lfloor \frac{2}{\gamma} \rfloor \leq i \leq t} \frac{1}{i^{1/2}}\right)^2 \left(\frac{1}{t - \lfloor \frac{2}{\gamma} \rfloor + 1} \sum_{\lfloor \frac{2}{\gamma} \rfloor \leq i \leq t} \frac{1}{i^{2a-b-1}}\right)$$

$$\leq \frac{2}{\gamma}\left(1-\frac{\gamma}{2}\right)^{-t}\left(\frac{2}{\sqrt{t}}\right)^2\left(\frac{1}{(2-2a+b)t^{2a-b-1}}\right)$$

$$= \frac{8}{(2-2a+b)\gamma}\left(1-\frac{\gamma}{2}\right)^{-t}\frac{1}{t^{2a-b}},$$

where the second inequality follows from Lemma F.3. In summary, if $2a-b \geq 1$ then

$$\sum_{i\in[t]}\left(1-\frac{\gamma}{2}\right)^{-i}\frac{1}{i^{2a-b}} \leq \frac{2}{\gamma}\left(1-\frac{\gamma}{2}\right)^{-\lfloor\frac{2}{\gamma}\rfloor+1} + \frac{8}{(2-2a+b)\gamma}\left(1-\frac{\gamma}{2}\right)^{-t}\frac{1}{t^{2a-b}},$$

If $2a-b < 1$ then by using Lemma F.1 with $a_i = \left(1-\frac{\gamma}{2}\right)^{-i}$ and $b_i := i^{-(2a-b)}$, we have

$$\sum_{i\in[t]}\left(1-\frac{\gamma}{2}\right)^{-i}\frac{1}{i^{2a-b}} \leq \left(\sum_{i\in[t]}\left(1-\frac{\gamma}{2}\right)^{-i}\right)\left(\frac{1}{t}\sum_{i\in[t]}\frac{1}{i^{2a-b}}\right) \leq \frac{2}{(1-2a+b)\gamma}\left(1-\frac{\gamma}{2}\right)^{-t}\frac{1}{t^{2a-b}},$$

where the last inequality follows from Lemma F.3. This concludes the proof of the first bound. $\square$

**Lemma F.5.** *Under Assumption 1.1, for any $t \geq 0$ and $x \in \mathcal{X}$, we have*

$$\|y_t^\star(x) - y_{t+1}^\star(x)\| \leq \frac{2D_\mathcal{Y}(\beta_t - \beta_{t+1})}{\beta_{t+1} + \mu}.$$

*Proof.* From Lemma A.1 (i), we have

$$\frac{\beta_{t+1} + \mu}{2}\|y_t^\star(x) - y_{t+1}^\star(x)\|^2$$

$$\leq \mathcal{L}_{t+1}(x, y_{t+1}^\star(x)) - \mathcal{L}_{t+1}(x, y_t^\star(x))$$

$$\leq \nabla_y \mathcal{L}_{t+1}(x, y_t^\star(x))^\top(y_{t+1}^\star(x) - y_t^\star(x))$$

$$= \nabla_y \mathcal{L}_t(x, y_t^\star(x))^\top(y_{t+1}^\star(x) - y_t^\star(x)) + (\beta_t - \beta_{t+1})\left(y_t^\star(x) - y_0\right)^\top(y_{t+1}^\star(x) - y_t^\star(x)),$$

where the second inequality follows from the concavity of $\mathcal{L}_t(x, \cdot)$. By definition of $y_t^\star(x)$ and the concavity of $\mathcal{L}_t(x, \cdot)$, we have from the optimality condition that

$$\nabla_y \mathcal{L}_t(x, y_t^\star(x))^\top(y_{t+1}^\star(x) - y_t^\star(x)) \leq 0.$$

From Cauchy-Schwartz inequality and the fact that $\|y_t^\star(x) - y_0\| \leq D_\mathcal{Y}$, we deduce that

$$(y_t^\star(x) - y_0)^\top(y_{t+1}^\star(x) - y_t^\star(x)) \leq \|y_t^\star(x) - y_0\|\|y_{t+1}^\star(x) - y_t^\star(x)\|$$
$$\leq D_\mathcal{Y}\|y_{t+1}^\star(x) - y_t^\star(x)\|.$$

As a result,

$$\frac{\beta_{t+1} + \mu}{2}\|y_t^\star(x) - y_{t+1}^\star(x)\|^2 \leq D_\mathcal{Y}(\beta_t - \beta_{t+1})\|y_{t+1}^\star(x) - y_t^\star(x)\|,$$

which concludes the proof. $\square$

# G. Implementation Details

## G.1. Dictionary Learning

Following the implementation details in (Boroun et al., 2023, Appendix F), we generate the old dataset matrix $\mathbf{A} = \mathbf{DC} \in \mathbb{R}^{m\times n}$ where $\mathbf{D} \in \mathbb{R}^{m\times p}$ is generated randomly with elements drawn from the standard Gaussian distribution whose columns are scaled to have a unit $\ell_2$-norm, and $\mathbf{C} = \frac{1}{\|\mathbf{U}\|_2\|\mathbf{V}\|_2}\mathbf{UV}^\top$ where $\mathbf{U} \in \mathbb{R}^{p\times l}$ and $\mathbf{V} \in \mathbb{R}^{n\times l}$ are generated randomly with elements drawn from the standard Gaussian distribution. The matrix $\tilde{\mathbf{C}} \in \mathbb{R}^{q\times n}$ is generated by adding $q - p$ columns of zeros to $\mathbf{C}$. The new dataset $\mathbf{A}' \in \mathbb{R}^{m\times n'}$ is generated randomly with elements drawn from the standard Gaussian distribution. Additionally, we let $n = 500, m = 100, p = 50, l = 5, q = 60, n' = 103, \delta = 10^{-4}, r = 5, B = 1$.

In terms of initialization, all the methods start from the same initial point $x_0 = (\mathbf{D}'_0, \mathbf{C}'_0)$ and $y_0 = 0$ where $\mathbf{D}'$ is generated randomly with elements drawn from the uniform distribution in $[0, 0.1]$ whose columns are scaled to have a unit $\ell_2$-norm, and $\mathbf{C}'_0 = \mathbf{0}_{q \times n'}$.

Turning to our algorithms, for Algorithm 1, we set $\tau_t = \left(\frac{2}{t+10}\right)^{4/5}, \beta_t = \frac{10^{-2}}{(t+10)^{1/5}}$. For Algorithm 2, we set $\tau_t = \left(\frac{2}{t+10}\right)^{3/4}, \beta_t = \frac{10^{-2}}{(t+10)^{1/4}}$ and for Algorithm 3, we set $\tau_t = \frac{10^2}{(t+10)^{3/5}}, \beta_t = \frac{10^{-2}}{(t+10)^{1/5}}$. Using the notations adopted by the corresponding papers, for R-PDCG (Boroun et al., 2023, Algorithm 1), we set $\tau = \frac{1}{K^{5/6}}$ and $\mu = \frac{10^{-2}}{K^{1/6}}$ and for CG-RPGA (Boroun et al., 2023, Algorithm 2), we set $\tau = \frac{1}{K^{3/4}}$ and $\mu = \frac{10^{-2}}{K^{1/4}}$ and $\sigma = \frac{2}{L_{yy}+\bar{\mu}}$ where the maximum number of iterations $K$ is set to at $10^3$. We note that the strong convexity modulus of $\mathcal{Y} = [0, B]$ in this experiment is $1/B$ and $L_{yy} = 0$ as derived in Appendix H.1. For SPFW (Gidel et al., 2017, Algorithm 2), we use the vanilla stepsize $\gamma = \frac{2}{t+2}$ for any $t \geq 0$. For AGP (Xu et al., 2023, Algorithm 1), we choose the parameters as suggested in (Xu et al., 2023, Theorem 3.2). That is $b_k = 0, \beta_k = \bar{\eta} + \bar{\beta}_k, c_k = \frac{19}{20\bar{\rho}k^{1/4}}$ and $\gamma_k = \frac{1}{\bar{\rho}}$ where $\bar{\eta} = \bar{\rho} = 0.05, \bar{\beta}_k = \bar{\rho}L^2_{y(\mathbf{D}',\mathbf{C}')} + \frac{16\tau L^2_{y(\mathbf{D}',\mathbf{C}')}}{\bar{\rho}c_k^2} - 2\bar{\eta}$ and $\tau = 2\max\left\{\frac{19^2\left(L_{(\mathbf{D}',\mathbf{C}')(\mathbf{D}',\mathbf{C}')}+2\bar{\eta}-\bar{\rho}L^2_{y(\mathbf{D}',\mathbf{C}')}\right)}{20^2 \cdot 16\bar{\rho}L^2_{y(\mathbf{D}',\mathbf{C}')}}, 2\right\}$ where all the Lipschitz constants are derived in Appendix H.1. We also run each algorithms 300 seconds.

### G.2. Robust Multiclass Classification

In these experiments, we choose $r = 10, \lambda = \frac{\lambda'}{2n^2}$ where $\lambda' = 10$. For the initialization, we let $\Theta_0 = \mathbf{0}_{k \times d}$ and $y_0 = \frac{1}{n}\mathbf{1}_n$. Turning to our algorithms, for Algorithm 1 and Algorithm 2, we set $\tau_t = \frac{2}{t+10}, \beta_t = 0$. For Algorithm 3, we set $\tau_t = \frac{10^3}{(t+1)^{1/2}}, \beta_t = 0$. Using the notations adopted by the corresponding papers, for R-PDCG (Boroun et al., 2023, Algorithm 1), we set $\tau = \frac{10}{K^{3/4}}$ and $\mu = 0$ and for CG-RPGA (Boroun et al., 2023, Algorithm 2), we set $\tau = \frac{10}{K^{1/2}}, \mu = 0$ and $\sigma = \frac{2}{L_{yy}+\bar{\mu}}$ where $\bar{\mu} = \lambda$ the maximum number of iterations $K$ is set to at $10^4$. We note that $L_{yy} = \sqrt{2}\lambda'$ as derived in Appendix H.2. For SPFW (Gidel et al., 2017, Algorithm 2), we use the vanilla stepsize $\gamma = \frac{2}{t+2}$ for any $t \geq 0$. For AGP (Xu et al., 2023, Algorithm 1), we choose the parameters as suggested in (Xu et al., 2023, Theorem 3.1). That is $b_k = c_k = 0, \beta_k = \eta = 2\max\left\{L_{\Theta\Theta}, L^2_{y\Theta}\rho + \frac{4L^2_{y\Theta}}{\rho\mu^2}\right\}$ where $\rho = \frac{\mu}{4L^2_{yy}}$ and $\mu = \lambda$ with all the Lipschitz constants are derived in Appendix H.2.

## H. Lipschitz constants

We devote this section to derive the Lipschitz constants inherited from dictionary learning and robust multiclass classification problems.

### H.1. Dictionary Learning

Recall in the dictionary learning problem, we have

$$\mathcal{L}((\mathbf{D}',\mathbf{C}'),y) := \frac{1}{2n'}\|\mathbf{A}' - \mathbf{D}'\mathbf{C}'\|_F^2 + y\left(\frac{1}{2n'}\|\mathbf{A} - \mathbf{D}'\tilde{\mathbf{C}}\|_F^2 - \delta\right)$$

$$\mathcal{X} := \{(\mathbf{D}',\mathbf{C}') \in \mathbb{R}^{m \times q} \times \mathbb{R}^{q \times n'} \mid \|\mathbf{C}'\|_* \leq r, \|d'_j\|_2 \leq 1, \forall j \in [q]\}$$

$$\mathcal{Y} := [0, B]$$

We have that

$$\nabla_{\mathbf{D}'}\mathcal{L}((\mathbf{D}',\mathbf{C}'),y) = -\frac{1}{n'}(\mathbf{A}' - \mathbf{D}'\mathbf{C}')\mathbf{C}'^\top - \frac{1}{n}y(\mathbf{A} - \mathbf{D}'\tilde{\mathbf{C}})\tilde{\mathbf{C}}^\top,$$

$$\nabla_{\mathbf{C}'}\mathcal{L}((\mathbf{D}',\mathbf{C}'),y) = -\frac{1}{n'}\mathbf{D}'^\top(\mathbf{A}' - \mathbf{D}'\mathbf{C}'),$$

$$\nabla_y\mathcal{L}((\mathbf{D}',\mathbf{C}'),y) = \frac{1}{2n}\|\mathbf{A} - \mathbf{D}'\tilde{\mathbf{C}}\|_F^2 - \delta.$$

For any pair $(\mathbf{D}'_1, \mathbf{C}'_1), y_1$ and $(\mathbf{D}'_2, \mathbf{C}'_2), y_2$ in $\mathcal{X} \times \mathcal{Y}$, we have that

$$
\begin{aligned}
|\nabla_y \mathcal{L}((\mathbf{D}'_1, \mathbf{C}'_1), y_1) - \nabla_y \mathcal{L}((\mathbf{D}'_2, \mathbf{C}'_2), y_2)| &= \frac{1}{2n} |\|\mathbf{A} - \mathbf{D}'_1 \tilde{\mathbf{C}}\|^2_F - \|\mathbf{A} - \mathbf{D}'_1 \tilde{\mathbf{C}}\|^2_F| \\
&= \frac{1}{2n} |\operatorname{Tr}((\mathbf{D}'_1 + \mathbf{D}'_2)\tilde{\mathbf{C}} - 2\mathbf{A})^\top (\mathbf{D}'_1 - \mathbf{D}'_2)\tilde{\mathbf{C}}| \\
&\leq \frac{1}{2n} \|(\mathbf{D}'_1 + \mathbf{D}'_2)\tilde{\mathbf{C}} - 2\mathbf{A}\|_F \|(\mathbf{D}'_1 - \mathbf{D}'_2)\tilde{\mathbf{C}}\|_F,
\end{aligned}
$$

where the last inequality follows from Cauchy-Schwartz inequality. Using the fact that $\|\mathbf{X}\mathbf{Y}\|_F \leq \|\mathbf{X}\|_2 \|\mathbf{Y}\|_F$ for any pair $(\mathbf{X}, \mathbf{Y})$ with compatible dimension, we have

$$
\begin{aligned}
|\nabla_y \mathcal{L}((\mathbf{D}'_1, \mathbf{C}'_1), y_1) - \nabla_y \mathcal{L}((\mathbf{D}'_2, \mathbf{C}'_2), y_2)| &\leq \frac{1}{2n} \|(\mathbf{D}'_1 + \mathbf{D}'_2)\tilde{\mathbf{C}} - 2\mathbf{A}\|_F \|\tilde{\mathbf{C}}\|_2 \|\mathbf{D}'_1 - \mathbf{D}'_2\|_F \\
&\leq \frac{1}{2n} (\|(\mathbf{D}'_1 + \mathbf{D}'_2)\tilde{\mathbf{C}}\|_F + 2\|\mathbf{A}\|_F) \|\tilde{\mathbf{C}}\|_2 \|\mathbf{D}'_1 - \mathbf{D}'_2\|_F \\
&\leq \frac{1}{2n} (\|\mathbf{D}'_1 + \mathbf{D}'_2\|_F \|\tilde{\mathbf{C}}\|_2 + 2\|\mathbf{A}\|_F) \|\tilde{\mathbf{C}}\|_2 \|\mathbf{D}'_1 - \mathbf{D}'_2\|_F \\
&\leq \frac{(\sqrt{q}\|\tilde{\mathbf{C}}\|_2 + \|\mathbf{A}\|_F)\|\tilde{\mathbf{C}}\|_2}{n} \|(\mathbf{D}'_1, \mathbf{C}'_1) - (\mathbf{D}'_2, \mathbf{C}'_2)\|_F,
\end{aligned}
$$

where in the last inequality, we use the fact that $\|\mathbf{D}'_1\|_F, \|\mathbf{D}'_2\|_F \leq \sqrt{q}$.

Turning to partial gradient with respect to $\mathbf{C}'$, we have that

$$
\begin{aligned}
&-\nabla_{\mathbf{C}'} \mathcal{L}((\mathbf{D}'_1, \mathbf{C}'_1), y_1) + \nabla_{\mathbf{C}'} \mathcal{L}((\mathbf{D}'_2, \mathbf{C}'_2), y_2) \\
&= \frac{1}{n'} \mathbf{D}'^\top_1 (\mathbf{A}' - \mathbf{D}'_1 \mathbf{C}'_1) - \frac{1}{n'} \mathbf{D}'^\top_2 (\mathbf{A}' - \mathbf{D}'_2 \mathbf{C}'_2) \\
&= \frac{1}{n'} (\mathbf{D}'_1 - \mathbf{D}'_2)^\top \mathbf{A}' - \frac{1}{n'} \mathbf{D}'^\top_1 \mathbf{D}'_1 \mathbf{C}'_1 + \frac{1}{n'} \mathbf{D}'^\top_2 \mathbf{D}'_2 \mathbf{C}'_2 \\
&= \frac{1}{n'} (\mathbf{D}'_1 - \mathbf{D}'_2)^\top \mathbf{A}' - \frac{1}{n'} \mathbf{D}'^\top_1 \mathbf{D}'_1 (\mathbf{C}'_1 - \mathbf{C}'_2) + \frac{1}{n'} (\mathbf{D}'^\top_2 \mathbf{D}'_2 - \mathbf{D}'^\top_1 \mathbf{D}'_1) \mathbf{C}'_2 \\
&= \frac{1}{n'} (\mathbf{D}'_1 - \mathbf{D}'_2)^\top \mathbf{A}' - \frac{1}{n'} \mathbf{D}'^\top_1 \mathbf{D}'_1 (\mathbf{C}'_1 - \mathbf{C}'_2) + \frac{1}{n'} (\mathbf{D}'^\top_2 \mathbf{D}'_2 - \mathbf{D}'^\top_1 \mathbf{D}'_2 + \mathbf{D}'^\top_1 \mathbf{D}'_2 - \mathbf{D}'^\top_1 \mathbf{D}'_1) \mathbf{C}'_2 \\
&= \frac{1}{n'} (\mathbf{D}'_1 - \mathbf{D}'_2)^\top \mathbf{A}' - \frac{1}{n'} \mathbf{D}'^\top_1 \mathbf{D}'_1 (\mathbf{C}'_1 - \mathbf{C}'_2) + \frac{1}{n'} (\mathbf{D}'_2 - \mathbf{D}'_1)^\top \mathbf{D}'_2 \mathbf{C}'_2 + \frac{1}{n'} \mathbf{D}'^\top_1 (\mathbf{D}'_2 - \mathbf{D}'_1) \mathbf{C}'_2 \\
&= \frac{1}{n'} (\mathbf{D}'_1 - \mathbf{D}'_2)^\top (\mathbf{A}' - \mathbf{D}'_2 \mathbf{C}'_2) + \frac{1}{n'} \mathbf{D}'^\top_1 (\mathbf{D}'_2 - \mathbf{D}'_1) \mathbf{C}'_2 - \frac{1}{n'} \mathbf{D}'^\top_1 \mathbf{D}'_1 (\mathbf{C}'_1 - \mathbf{C}'_2)
\end{aligned}
$$

Using triangle inequality, we have

$$
\begin{aligned}
&\|\nabla_{\mathbf{C}'} \mathcal{L}((\mathbf{D}'_1, \mathbf{C}'_1), y_1) - \nabla_{\mathbf{C}'} \mathcal{L}((\mathbf{D}'_2, \mathbf{C}'_2), y_2)\|_F \\
&\leq \frac{1}{n'} \|\mathbf{D}'_1 - \mathbf{D}'_2\|_F \|\mathbf{A}' - \mathbf{D}'_2 \mathbf{C}'_2\|_F + \frac{1}{n'} \|\mathbf{D}'_1\|_F \|(\mathbf{D}'_2 - \mathbf{D}'_1)\mathbf{C}'_2\|_F + \frac{1}{n'} \|\mathbf{D}'_1\|^2_F \|\mathbf{C}'_1 - \mathbf{C}'_2\|_F \\
&\leq \frac{1}{n'} (\|\mathbf{A}'\|_F + \|\mathbf{C}'_2\|_2 \|\mathbf{D}'_2\|_F + \|\mathbf{C}'_2\|_2 \|\mathbf{D}'_1\|_F) \|\mathbf{D}'_1 - \mathbf{D}'_2\|_F + \frac{1}{n'} \|\mathbf{D}'_1\|^2_F \|\mathbf{C}'_1 - \mathbf{C}'_2\|_F \\
&\leq \frac{1}{n'} (\|\mathbf{A}'\|_F + 2r\sqrt{q}) \|\mathbf{D}'_1 - \mathbf{D}'_2\|_F + \frac{1}{n'} q \|\mathbf{C}'_1 - \mathbf{C}'_2\|_F
\end{aligned}
$$

Now, we investigate the partial gradient with respect to $\mathbf{D}'$, we have

$$
\begin{aligned}
&-\nabla_{\mathbf{D}'} \mathcal{L}((\mathbf{D}'_1, \mathbf{C}'_1), y_1) + \nabla_{\mathbf{D}'} \mathcal{L}((\mathbf{D}'_2, \mathbf{C}'_2), y_2) \\
&= \frac{1}{n'} (\mathbf{A}' - \mathbf{D}'_1 \mathbf{C}'_1) \mathbf{C}'^\top_1 + \frac{1}{n} y_1 (\mathbf{A} - \mathbf{D}'_1 \tilde{\mathbf{C}}) \tilde{\mathbf{C}}^\top - \frac{1}{n'} (\mathbf{A}' - \mathbf{D}'_2 \mathbf{C}'_2) \mathbf{C}'^\top_2 - \frac{1}{n} y_2 (\mathbf{A} - \mathbf{D}'_2 \tilde{\mathbf{C}}) \tilde{\mathbf{C}}^\top \\
&= \frac{1}{n'} (\mathbf{A}' - \mathbf{D}'_1 \mathbf{C}'_1) \mathbf{C}'^\top_1 - \frac{1}{n'} (\mathbf{A}' - \mathbf{D}'_2 \mathbf{C}'_2) \mathbf{C}'^\top_2 + \frac{1}{n} y_1 (\mathbf{A} - \mathbf{D}'_1 \tilde{\mathbf{C}}) \tilde{\mathbf{C}}^\top - \frac{1}{n} y_2 (\mathbf{A} - \mathbf{D}'_2 \tilde{\mathbf{C}}) \tilde{\mathbf{C}}^\top.
\end{aligned}
$$

We have that

$$\begin{aligned}
(\mathbf{A}' &- \mathbf{D}_1'\mathbf{C}_1')\mathbf{C}_1'^\top - (\mathbf{A}' - \mathbf{D}_2'\mathbf{C}_2')\mathbf{C}_2'^\top \\
&= \mathbf{A}'(\mathbf{C}_1' - \mathbf{C}_2')^\top - \mathbf{D}_1'\mathbf{C}_1'\mathbf{C}_1'^\top + \mathbf{D}_2'\mathbf{C}_2'\mathbf{C}_2'^\top \\
&= \mathbf{A}'(\mathbf{C}_1' - \mathbf{C}_2')^\top - (\mathbf{D}_1' - \mathbf{D}_2')\mathbf{C}_1'\mathbf{C}_1'^\top + \mathbf{D}_2'(\mathbf{C}_2'\mathbf{C}_2'^\top - \mathbf{C}_1'\mathbf{C}_1'^\top) \\
&= \mathbf{A}'(\mathbf{C}_1' - \mathbf{C}_2')^\top - (\mathbf{D}_1' - \mathbf{D}_2')\mathbf{C}_1'\mathbf{C}_1'^\top + \mathbf{D}_2'(\mathbf{C}_2'\mathbf{C}_2'^\top - \mathbf{C}_2'\mathbf{C}_1'^\top + \mathbf{C}_2'\mathbf{C}_1'^\top - \mathbf{C}_1'\mathbf{C}_1'^\top) \\
&= \mathbf{A}'(\mathbf{C}_1' - \mathbf{C}_2')^\top - (\mathbf{D}_1' - \mathbf{D}_2')\mathbf{C}_1'\mathbf{C}_1'^\top + \mathbf{D}_2'\mathbf{C}_2'(\mathbf{C}_2' - \mathbf{C}_1')^\top + \mathbf{D}_2'(\mathbf{C}_2' - \mathbf{C}_1')\mathbf{C}_1'^\top \\
&= (\mathbf{A}' - \mathbf{D}_2'\mathbf{C}_2')(\mathbf{C}_1' - \mathbf{C}_2')^\top + \mathbf{D}_2'(\mathbf{C}_2' - \mathbf{C}_1')\mathbf{C}_1'^\top - (\mathbf{D}_1' - \mathbf{D}_2')\mathbf{C}_1'\mathbf{C}_1'^\top,
\end{aligned}$$

and

$$\begin{aligned}
y_1(\mathbf{A} - \mathbf{D}_1'\tilde{\mathbf{C}})\tilde{\mathbf{C}}^\top - y_2(\mathbf{A} - \mathbf{D}_2'\tilde{\mathbf{C}})\tilde{\mathbf{C}}^\top &= (y_1 - y_2)\mathbf{A}\tilde{\mathbf{C}}^\top - (y_1\mathbf{D}_1' - y_2\mathbf{D}_2')\tilde{\mathbf{C}}\tilde{\mathbf{C}}^\top \\
&= (y_1 - y_2)\mathbf{A}\tilde{\mathbf{C}}^\top - (y_1\mathbf{D}_1' - y_2\mathbf{D}_1' + y_2\mathbf{D}_1' - y_2\mathbf{D}_2')\tilde{\mathbf{C}}\tilde{\mathbf{C}}^\top \\
&= (y_1 - y_2)(\mathbf{A} - \mathbf{D}_1'\tilde{\mathbf{C}})\tilde{\mathbf{C}}^\top - y_2(\mathbf{D}_1' - \mathbf{D}_2')\tilde{\mathbf{C}}\tilde{\mathbf{C}}^\top
\end{aligned}$$

In summary, we have

$$\begin{aligned}
-\nabla_{\mathbf{D}'}&\mathcal{L}((\mathbf{D}_1', \mathbf{C}_1'), y_1) + \nabla_{\mathbf{D}'}\mathcal{L}((\mathbf{D}_2', \mathbf{C}_2'), y_2) \\
&= \frac{1}{n'}(\mathbf{A}' - \mathbf{D}_2'\mathbf{C}_2')(\mathbf{C}_1' - \mathbf{C}_2')^\top + \frac{1}{n'}\mathbf{D}_2'(\mathbf{C}_2' - \mathbf{C}_1')\mathbf{C}_1'^\top \\
&\quad - (\mathbf{D}_1' - \mathbf{D}_2')\left(\frac{1}{n'}\mathbf{C}_1'\mathbf{C}_1'^\top + \frac{1}{n}y_2\tilde{\mathbf{C}}\tilde{\mathbf{C}}^\top\right) + \frac{1}{n}(y_1 - y_2)(\mathbf{A} - \mathbf{D}_1'\tilde{\mathbf{C}})\tilde{\mathbf{C}}^\top.
\end{aligned}$$

Thus, from the triangle inequality and the fact that $\|\mathbf{XY}\|_F \le \|\mathbf{X}\|_2\|\mathbf{Y}\|_F$ for any pair $(\mathbf{X}, \mathbf{Y})$ with compatible dimension, we have

$$\begin{aligned}
\| - &\nabla_{\mathbf{D}'}\mathcal{L}((\mathbf{D}_1', \mathbf{C}_1'), y_1) + \nabla_{\mathbf{D}'}\mathcal{L}((\mathbf{D}_2', \mathbf{C}_2'), y_2)\|_F \\
&\le \frac{1}{n'}\|\mathbf{A}' - \mathbf{D}_2'\mathbf{C}_2'\|_F\|\mathbf{C}_1' - \mathbf{C}_2'\|_F + \frac{1}{n'}\|\mathbf{D}_2'\|_F\|\mathbf{C}_2' - \mathbf{C}_1'\|_F\|\mathbf{C}_1'\|_2 \\
&\quad + \|\mathbf{D}_1' - \mathbf{D}_2'\|_F\left\|\frac{1}{n'}\mathbf{C}_1'\mathbf{C}_1'^\top + \frac{1}{n}y_2\tilde{\mathbf{C}}\tilde{\mathbf{C}}^\top\right\|_2 + \frac{1}{n}|y_1 - y_2|\|\mathbf{A} - \mathbf{D}_1'\tilde{\mathbf{C}}\|_F\|\tilde{\mathbf{C}}\|_2 \\
&\le \frac{1}{n'}(\|\mathbf{A}'\|_F + \|\mathbf{D}_2'\|_F\|\mathbf{C}_2'\|_2)\|\mathbf{C}_1' - \mathbf{C}_2'\|_F + \frac{1}{n'}\|\mathbf{D}_2'\|_F\|\mathbf{C}_2' - \mathbf{C}_1'\|_F\|\mathbf{C}_1'\|_2 \\
&\quad + \|\mathbf{D}_1' - \mathbf{D}_2'\|_F(\frac{1}{n'}\|\mathbf{C}_1'\|_2^2 + \frac{1}{n}y_2\|\tilde{\mathbf{C}}\|_2^2) + \frac{1}{n}|y_1 - y_2|(\|\mathbf{A}\|_F + \|\mathbf{D}_1'\|_F\|\tilde{\mathbf{C}}\|_2)\|\tilde{\mathbf{C}}\|_2 \\
&\le \frac{1}{n'}(\|\mathbf{A}'\|_F + 2r\sqrt{q})\|\mathbf{C}_1' - \mathbf{C}_2'\|_F + \|\mathbf{D}_1' - \mathbf{D}_2'\|_F\left(\frac{1}{n'}r^2 + \frac{1}{n}B\|\tilde{\mathbf{C}}\|_2^2\right) + \frac{1}{n}|y_1 - y_2|(\|\mathbf{A}\|_F + \sqrt{q}\|\tilde{\mathbf{C}}\|_2)\|\tilde{\mathbf{C}}\|_2.
\end{aligned}$$

As a result, we have

$$\begin{aligned}
\|\nabla_{(\mathbf{D}', \mathbf{C}')}&\mathcal{L}((\mathbf{D}_1', \mathbf{C}_1'), y_1) - \nabla_{(\mathbf{D}', \mathbf{C}')}\mathcal{L}((\mathbf{D}_2', \mathbf{C}_2'), y_2)\|_F \\
&\le \Bigg(\left(\left(\frac{1}{n'}(\|\mathbf{A}'\|_F + 2r\sqrt{q})\|\mathbf{D}_1' - \mathbf{D}_2'\|_F + \frac{1}{n'}q\|\mathbf{C}_1' - \mathbf{C}_2'\|_F\right)^2 \\
&\quad + \left(\frac{1}{n'}(\|\mathbf{A}'\|_F + 2r\sqrt{q})\|\mathbf{C}_1' - \mathbf{C}_2'\|_F + \|\mathbf{D}_1' - \mathbf{D}_2'\|_F\left(\frac{r^2}{n'} + \frac{B\|\tilde{\mathbf{C}}\|_2^2}{n}\right) + \frac{1}{n}|y_1 - y_2|(\|\mathbf{A}\|_F + \sqrt{q}\|\tilde{\mathbf{C}}\|_2)\|\tilde{\mathbf{C}}\|_2\right)^2\Bigg)^{1/2} \\
&\le \Bigg(2\left(\frac{(\|\mathbf{A}'\|_F + 2r\sqrt{q})^2}{n'^2}\|\mathbf{D}_1' - \mathbf{D}_2'\|_F^2 + \frac{q^2}{n'^2}\|\mathbf{C}_1' - \mathbf{C}_2'\|_F^2\right) \\
&\quad + 3\left(\frac{(\|\mathbf{A}'\|_F + 2r\sqrt{q})^2}{n'^2}\|\mathbf{C}_1' - \mathbf{C}_2'\|_F^2 + \|\mathbf{D}_1' - \mathbf{D}_2'\|_F^2\left(\frac{r^2}{n'} + \frac{B\|\tilde{\mathbf{C}}\|_2^2}{n}\right)^2 + |y_1 - y_2|^2\frac{(\|\mathbf{A}\|_F + \sqrt{q}\|\tilde{\mathbf{C}}\|_2)^2\|\tilde{\mathbf{C}}\|_2^2}{n^2}\right)\Bigg)^{1/2}
\end{aligned}$$

$$= \left( \left( \left( \frac{2(\|\mathbf{A}'\|_F + 2r\sqrt{q})^2}{n'^2} + 3\left( \frac{r^2}{n'} + \frac{B\|\tilde{\mathbf{C}}\|_2^2}{n} \right)^2 \right) \|\mathbf{D}'_1 - \mathbf{D}'_2\|_F^2 + \frac{2q^2 + 3(\|\mathbf{A}'\|_F + 2r\sqrt{q})^2}{n'^2} \|\mathbf{C}'_1 - \mathbf{C}'_2\|_F^2 \right.$$

$$\left. + \frac{3(\|\mathbf{A}\|_F + \sqrt{q}\|\tilde{\mathbf{C}}\|_2)^2\|\tilde{\mathbf{C}}\|_2^2}{n^2} |y_1 - y_2|^2 \right)^{1/2}$$

$$\leq \max \left\{ \sqrt{\frac{2(\|\mathbf{A}'\|_F + 2r\sqrt{q})^2}{n'^2} + 3\left( \frac{r^2}{n'} + \frac{B\|\tilde{\mathbf{C}}\|_2^2}{n} \right)^2}, \frac{\sqrt{2q^2 + 3(\|\mathbf{A}'\|_F + 2r\sqrt{q})^2}}{n'} \right\} \|(\mathbf{D}'_1, \mathbf{C}'_1) - (\mathbf{D}'_2, \mathbf{C}'_2)\|_F$$

$$+ \frac{\sqrt{3}(\|\mathbf{A}\|_F + \sqrt{q}\|\tilde{\mathbf{C}}\|_2)\|\tilde{\mathbf{C}}\|_2}{n} |y_1 - y_2|,$$

where in the second inequality, we use the fact that $(a + b)^2 \leq 2(a^2 + b^2)$ and $(a + b + c)^2 \leq 3(a^2 + b^2 + c^2)$ and in the third inequality, we use the fact that $a^2 + b^2 \leq (a + b)^2$ for any non-negative tuple $(a, b)$. Therefore, we can choose

$$L_{(\mathbf{D}',\mathbf{C}')(\mathbf{D}',\mathbf{C}')} := \max \left\{ \sqrt{\frac{2(\|\mathbf{A}'\|_F + 2r\sqrt{q})^2}{n'^2} + 3\left( \frac{r^2}{n'} + \frac{B\|\tilde{\mathbf{C}}\|_2^2}{n} \right)^2}, \frac{\sqrt{2q^2 + 3(\|\mathbf{A}'\|_F + 2r\sqrt{q})^2}}{n'} \right\}$$

$$L_{y(\mathbf{D}',\mathbf{C}')} := \frac{\sqrt{3}(\|\mathbf{A}\|_F + \sqrt{q}\|\tilde{\mathbf{C}}\|_2)\|\tilde{\mathbf{C}}\|_2}{n}$$

$$L_{yy} := 0.$$

## H.2. Robust Multiclass Classification

Recall in the dictionary learning problem, we have

$$\mathcal{L}(\Theta, y) := \frac{1}{n} \sum_{i \in [n]} y_i \log(1 + \exp(\mathrm{Tr}(\Theta \mathbf{A}_i))) - \frac{\lambda'}{2n^2} \|ny - \mathbf{1}_n\|_2^2,$$

$$\mathcal{X} := \{\Theta \in \mathbb{R}^{k \times d} \mid \|\Theta\|_* \leq r\}$$

$$\mathcal{Y} := \{y \in \mathbb{R}^n \mid \|ny - \mathbf{1}_n\|_2 \leq \rho\}$$

where $\lambda' = 2n^2\lambda$, $(a_i, b_i) \in \mathbb{R}^d \times [k]$ and matrices $\mathbf{A}_i$ have their $j$th columns as

$$\mathrm{col}(\mathbf{A}_i)_j = \begin{cases} a_i & j \in [k] \setminus \{b_i\} \\ -(k - 1)a_i & j = b_i \end{cases}$$

We have that

$$\nabla_\Theta \mathcal{L}(\Theta, y) := \sum_{i \in [n]} y_i \frac{\exp(\mathrm{Tr}(\Theta \mathbf{A}_i))}{1 + \exp(\mathrm{Tr}(\Theta \mathbf{A}_i))} \mathbf{A}_i$$

$$\nabla_y \mathcal{L}(\Theta, y) := (\log(1 + \exp(\mathrm{Tr}(\Theta \mathbf{A}_1))), \ldots, \log(1 + \exp(\mathrm{Tr}(\Theta \mathbf{A}_n)))) - \lambda' \left( y - \frac{1}{n} \mathbf{1}_n \right)$$

For any pair $(\Theta, y)$ and $(\Theta', y')$ in $\mathcal{X} \times \mathcal{Y}$, we have

$$\|\nabla_y \mathcal{L}(\Theta, y) - \nabla_y \mathcal{L}(\Theta', y')\|_2^2 = \sum_{i \in [n]} \left( \log(1 + \exp(\mathrm{Tr}(\Theta \mathbf{A}_1))) - \log(1 + \exp(\mathrm{Tr}(\Theta' \mathbf{A}_i))) - \lambda'(y_i - y'_i) \right)^2$$

$$\leq \sum_{i \in [n]} 2 \left( \log(1 + \exp(\mathrm{Tr}(\Theta \mathbf{A}_1))) - \log(1 + \exp(\mathrm{Tr}(\Theta' \mathbf{A}_i))) \right)^2 + 2\lambda'^2 (y_i - y'_i)^2.$$

By the intermediate value theorem, for any $i \in [n]$, there exists $\Theta_i \in [\Theta, \Theta']$ such that

$$|\log(1 + \exp(\mathrm{Tr}(\Theta'\mathbf{A}_i))) - \log(1 + \exp(\mathrm{Tr}(\Theta\mathbf{A}_i)))| = \frac{\exp(\mathrm{Tr}(\Theta_i\mathbf{A}_i))}{1 + \exp(\mathrm{Tr}(\Theta_i\mathbf{A}_i))}|\mathrm{Tr}\left((\Theta' - \Theta)\mathbf{A}_i\right)| \le \|\mathbf{A}_i\|_F\|\Theta' - \Theta\|_F,$$

which implies

$$\|\nabla_y\mathcal{L}(\Theta, y) - \nabla_y\mathcal{L}(\Theta', y')\|_2 \le \left(2\sum_{i \in [n]}\|\mathbf{A}_i\|_F^2\right)^{1/2}\|\Theta' - \Theta\|_F + \sqrt{2}\lambda'\|y - y'\|_2.$$

By the triangle inequality, we also have that

$$\|\nabla_\Theta\mathcal{L}(\Theta, y) - \nabla_\Theta\mathcal{L}(\Theta', y')\|_F \le \left\|\sum_{i \in [n]} y_i\frac{\exp(\mathrm{Tr}(\Theta\mathbf{A}_i))}{1 + \exp(\mathrm{Tr}(\Theta\mathbf{A}_i))}\mathbf{A}_i - \sum_{i \in [n]} y_i'\frac{\exp(\mathrm{Tr}(\Theta\mathbf{A}_i))}{1 + \exp(\mathrm{Tr}(\Theta\mathbf{A}_i))}\mathbf{A}_i\right\|_F$$

$$+ \left\|\sum_{i \in [n]} y_i'\frac{\exp(\mathrm{Tr}(\Theta\mathbf{A}_i))}{1 + \exp(\mathrm{Tr}(\Theta\mathbf{A}_i))}\mathbf{A}_i - \sum_{i \in [n]} y_i'\frac{\exp(\mathrm{Tr}(\Theta'\mathbf{A}_i))}{1 + \exp(\mathrm{Tr}(\Theta'\mathbf{A}_i))}\mathbf{A}_i\right\|_F.$$

We observe that

$$\left\|\sum_{i \in [n]} y_i\frac{\exp(\mathrm{Tr}(\Theta\mathbf{A}_i))}{1 + \exp(\mathrm{Tr}(\Theta\mathbf{A}_i))}\mathbf{A}_i - \sum_{i \in [n]} y_i'\frac{\exp(\mathrm{Tr}(\Theta\mathbf{A}_i))}{1 + \exp(\mathrm{Tr}(\Theta\mathbf{A}_i))}\mathbf{A}_i\right\|_F \le \sum_{i \in [n]}|y_i - y_i'|\frac{\exp(\mathrm{Tr}(\Theta\mathbf{A}_i))}{1 + \exp(\mathrm{Tr}(\Theta\mathbf{A}_i))}\|\mathbf{A}_i\|_F$$

$$\le \sum_{i \in [n]}|y_i - y_i'|\|\mathbf{A}_i\|_F$$

$$\le \left(\sum_{i \in [n]}\|\mathbf{A}_i\|_F^2\right)^{1/2}\|y - y'\|_2,$$

and

$$\left\|\sum_{i \in [n]} y_i'\frac{\exp(\mathrm{Tr}(\Theta\mathbf{A}_i))}{1 + \exp(\mathrm{Tr}(\Theta\mathbf{A}_i))}\mathbf{A}_i - \sum_{i \in [n]} y_i'\frac{\exp(\mathrm{Tr}(\Theta'\mathbf{A}_i))}{1 + \exp(\mathrm{Tr}(\Theta'\mathbf{A}_i))}\mathbf{A}_i\right\|_F$$

$$\le \sum_{i \in [n]} y_i'\left|\frac{\exp(\mathrm{Tr}(\Theta\mathbf{A}_i))}{1 + \exp(\mathrm{Tr}(\Theta\mathbf{A}_i))} - \frac{\exp(\mathrm{Tr}(\Theta'\mathbf{A}_i))}{1 + \exp(\mathrm{Tr}(\Theta'\mathbf{A}_i))}\right|\|\mathbf{A}_i\|_F.$$

By the intermediate value theorem, for any $i \in [n]$, there exists $\Theta_i' \in [\Theta, \Theta']$ such that

$$\left|\frac{\exp(\mathrm{Tr}(\Theta\mathbf{A}_i))}{1 + \exp(\mathrm{Tr}(\Theta\mathbf{A}_i))} - \frac{\exp(\mathrm{Tr}(\Theta'\mathbf{A}_i))}{1 + \exp(\mathrm{Tr}(\Theta'\mathbf{A}_i))}\right| = \frac{\exp(\mathrm{Tr}(\Theta_i'\mathbf{A}_i))}{(1 + \exp(\mathrm{Tr}(\Theta_i'\mathbf{A}_i)))^2}\left|\mathrm{Tr}(\mathbf{A}_i^\top(\Theta' - \Theta))\right|$$

$$\le \frac{1}{4}\|\mathbf{A}_i\|_F\|\Theta' - \Theta\|_F.$$

Thus, we deduce that

$$\left\|\sum_{i \in [n]} y_i'\frac{\exp(\mathrm{Tr}(\Theta\mathbf{A}_i))}{1 + \exp(\mathrm{Tr}(\Theta\mathbf{A}_i))}\mathbf{A}_i - \sum_{i \in [n]} y_i'\frac{\exp(\mathrm{Tr}(\Theta'\mathbf{A}_i))}{1 + \exp(\mathrm{Tr}(\Theta'\mathbf{A}_i))}\mathbf{A}_i\right\|_F \le \frac{1}{4}\sum_{i \in [n]} y_i'\|\mathbf{A}_i\|_F^2\|\Theta' - \Theta\|_F$$

$$\le \frac{1}{4}\max_{i \in [n]}\|\mathbf{A}_i\|_F^2\|\Theta' - \Theta\|_F.$$

In summary, we have

$$\|\nabla_\Theta \mathcal{L}(\Theta, y) - \nabla_\Theta \mathcal{L}(\Theta', y')\|_F \le \frac{1}{4} \max_{i \in [n]} \|\mathbf{A}_i\|_F^2 \|\Theta' - \Theta\|_F + \left( \sum_{i \in [n]} \|\mathbf{A}_i\|_F^2 \right)^{1/2} \|y - y'\|_2.$$

Therefore, we can choose

$$L_{\Theta\Theta} := \frac{1}{4} \max_{i \in [n]} \|\mathbf{A}_i\|_F^2 = \frac{(k-1)^2 + (k-1)}{4} \max_{i \in [n]} \|a_i\|_2^2 = \frac{(k-1)k}{4} \max_{i \in [n]} \|a_i\|_2^2$$

$$L_{y\Theta} := \left( \frac{(k-1)k}{2} \sum_{i \in [n]} \|a_i\|_2^2 \right)^{1/2}$$

$$L_{yy} := \sqrt{2}\lambda'.$$

