# OpenReview forum: "Projection-Free Algorithms for Minimax Problems"
_ICML.cc/2026/Conference — ICML 2026 regular_

### Official Review · Reviewer_MCBF · 2026-03-06

**Soundness:** 3
**Presentation:** 3
**Significance:** 3
**Originality:** 2
**Overall Recommendation:** 4
**Confidence:** 2

**Summary:**

The paper discusses "projection-free" primal methods for minimax problems, where "projection-free" is interpreted as having access to a linear programming oracle. The paper is well written and relevant to the machine learning community. Experimental results highlight the fact that the algorithms perform well in practice.

**Compliance With Llm Reviewing Policy:**

Affirmed.

**Key Questions For Authors:**

see point in the weaknesses.

**Limitations:**

yes

**Strengths And Weaknesses:**

Strengths:
- presentation is on point
- the paper is complete and presents rates for various types of oracles (linear minimization/projection) and various problem settings (convex-concave, nonconvex-concave, strongly-convex strongly-concave, nonconvex strongly-concave)
- the rates are plausible (although I haven't checked the proofs/appendix) and agree with the state-of-the-art
- assumptions are relatively weak; some results require a regularity condition on one of the feasible sets

Weaknesses:
- Alg. 1/2 have already been proposed in the literature (although this is clearly acknowledged), so the contributions lie in the analysis rather than providing new algorithmic ideas
- key concern: it would be great if the authors could consider the work [R1], that deals with primal methods for monotone (and/or strongly monotone) VIs. [R1] achieves rates of O(1/eps^2) for example on montone VIs (which comprises convex-concave games as special cases) while only requiring local linear approximations of the feasible set. The quadratic programming oracle assumed in [R1] has linear constraints and a strongly convex objective, so it can be solved with (accelerated) dual ascent in O(log(1/eps)). Combining the two, gives rates of O(1/eps^2 log(1/eps)), which significantly outperform the rates in the current manuscript, while avoiding access to *any* linear/quadratic programming oracle. I might be overlooking something, but I believe this would be an important point to at least discuss in the manuscript.


[R1] Zhang et al., "Primal methods for variational inequality problems with functional constraints," Math. Prog., 2025

---

> ### Author Rebuttal · Authors · 2026-03-30
>
> We thank the reviewer for their positive assessment of our paper. We address the specific technical concerns below.
>
> **1. Novelty in Algorithm Design and Analysis:**
> While we acknowledge that the structure of Algorithms 1 and 2 appears in prior work, we would like to highlight two fundamental contributions that distinguish our work:
>
> *   **Anytime Property:** Unlike previous methods (e.g., Boroun et al., 2023) that require a pre-specified iteration budget $T$ or target accuracy $\epsilon$ to tune parameters, our algorithms utilize an **anytime schedule**. This is of significant practical value as it allows the algorithm to maintain convergence properties throughout execution without prior knowledge of the termination point.
>
> *   **Flexibility of the Dual Set:** Existing single-loop LMO methods heavily rely on the **strong convexity** of the dual feasible set $\mathcal{Y}$. Our analysis and parameter choices eliminate this requirement, providing the first convergence guarantees for settings where $\mathcal{Y}$ is a general compact convex set.
>
> **2. Comparison with Zhang et al. [R1] (Math. Prog., 2025):**
> We thank the reviewer for pointing us to this very recent and relevant work. We will include a discussion of [R1] in our literature review to provide a more complete picture. However, we believe there are two critical distinctions:
>
> *   **Oracle Complexity vs. Set Geometry:** [R1] assumes access to a **Quadratic Programming (QP) oracle** and focuses on **smooth functional constraints**. In the problem classes we study, such as **nuclear norm balls**, the QP oracle (or local linear approximation of the set) is computationally equivalent to a **full projection**. For these sets, a single projection requires a full SVD, which is the exact bottleneck we seek to avoid. Our work relies strictly on the **Linear Minimization Oracle (LMO)**, which, for a nuclear norm ball, only requires the leading singular vector (via a power method), making our per-iteration cost orders of magnitude lower than a QP-based approach.
>
> *   **Nonconvexity:** [R1] focuses on monotone and strongly monotone Variational Inequalities (VIs), which correspond to the convex-concave regime. A primary contribution of our work is the analysis of the **nonconvex-concave (NC-C)** setting, establishing state-of-the-art rates for single-loop algorithms in this broader and more challenging problem class.
>
> In summary, while [R1] provides excellent rates for monotone VIs with functional constraints, our work addresses a different computational regime where **projections (and QPs) are prohibitively expensive** compared to LMOs, particularly in the nonconvex regime.
>
> We thank the reviewer again for their insightful feedback. We hope our response and planned revisions have addressed your concerns. If there are any further points we could clarify, or any additional suggestions that could help improve the manuscript and your assessment of the work, we would be very happy to address them.
>
> [//]: <> (We were wondering if there are any other points we can clarify that could further improve the manuscript and your assessment of the work?)

---

> > ### Author Rebuttal · Reviewer_MCBF · 2026-04-01
> >
> > I would like to thank you for addressing the questions in the rebuttal well and providing additional clarification. I am in favor of acceptance as also indicated with my score. However, I believe the contributions overall are more on the incremental/technical side and therefore keep a score of 4.

---

### Official Review · Reviewer_P2FM · 2026-03-12

**Soundness:** 2
**Presentation:** 2
**Significance:** 2
**Originality:** 2
**Overall Recommendation:** 5
**Confidence:** 4

**Summary:**

The paper explores LMO-based methods for solving saddle problems. Various formulations are being studied: the objective function, optimization sets, and LMO/projection oracles. The paper provides extensive new results in this area, primarily in terms of assumptions on the problem. There are also complementary experiments.

**Compliance With Llm Reviewing Policy:**

Affirmed.

**Final Justification:**

Good paper! Vote for acceptance!

**Key Questions For Authors:**

Asked

**Limitations:**

Yes

**Strengths And Weaknesses:**

__Strengths:__

1) It provides a good contribution to the theory of solving saddle point problems with constraints using LMO oracles.

2) The results of the work are quite broad and consider various formulations, both from the point of view of the objective function, the set, and from the point of view of the LMO and PO oracles.

3) The rejection of additional assumptions (other than standard compactness and convexity) on sets $X$ and $Y$ looks like an important detail.

4) In most cases, the authors outperform their competitors in theoretical terms.

5) The paper is well written and structured.

__Weaknesses:__

1) From the point of view of algorithm design, there is no contribution. But this is ok

2) The results for the C-C and SC-C cases do not look optimistic. It is interesting to understand these problems of a) setting, b) method, or c) analysis. This will give a good complete picture of the area.

3) The DRO statement looks a bit strange, the KL regularizer looks more natural, although I understand that this is necessary for an honest theory.

The design of the experiments is a bit strange. It took me a moment to realize that the first picture of Figure 1 is a dictionary problem, and the other two are already DRO. The same for Table 3. I would divide the results into two parts.

---

> ### Author Rebuttal · Authors · 2026-03-30
>
> We are very grateful to the reviewer for their positive evaluation our work. We sincerely thank the reviewer for supporting our paper and for the constructive suggestions to improve the clarity of our work.
>
> **1. Novelty in Algorithm Design:**
> While we acknowledge the structural similarities to existing methods, we would like to highlight two key design choices that distinguish our work and enable our improved theoretical results:
>
> *   **Anytime Design:** Unlike previous projection-free minimax methods that require a fixed iteration budget $T$ or target accuracy $\epsilon$ to tune stepsizes, our algorithms use an anytime schedule. This makes them robust for practical use cases where the necessary budget is unknown a priori.
>
> *   **Dual Step-size and Flexibility:** A major technical difference from R-PDCG (Boroun et al., 2023) is our use of **a closed-loop dual stepsize**. R-PDCG requires knowledge of the strong-convexity modulus of the dual set $\mathcal{Y}$ and requires that modulus to be positive. Our design does not require knowledge or existence of strong convexity, allowing our framework to converge on general convex sets where previous single-loop methods were inapplicable.
>
> **2. Understanding Convex-Concave Rates:**
> The reviewer raises an excellent point regarding the rates in the C-C and C-SC settings. We believe these rates are limited by two distinct factors:
>
> *   **Asymmetric/Lopsided Analysis:** Our framework is primarily designed for the nonconvex-concave setting. By regularizing the dual variable, we deliberately break the symmetry inherent in convex strong duality ($\min \max = \max \min$) to prioritize the primal minimization. This lopsided regularization is optimal for NC-C but yields sub-optimal rates in the symmetric C-C case compared to specialized convex methods.
>
> *   **Gradient Estimation Error:** In a single-loop structure, the algorithm uses an inexact estimate of the gradient of the smoothed primal function $f_\beta$. In the projection-based setting, extragradient or optimistic methods fix this by using two updates. To our knowledge, such an extra-LMO improvement for single-loop projection-free optimization is currently an open problem in the field.
>
> **3. DRO Formulation (KL vs. $\chi^2$):**
> We agree that the KL regularizer is a natural choice in many DRO contexts. However, the KL divergence involves logarithmic terms that are not Lipschitz smooth, which violates the core smoothness assumptions required by our framework. Following the DRO literature (Gotoh et al., 2018; Duchi & Namkoong, 2019), we utilize the **Pearson $\chi^2$-divergence** because it is more algorithmically friendly (Lipschitz smooth) and, as shown in those works, all $\phi$-divergences are asymptotically equivalent and lead to the same variance regularization effects.
>
> **4. Organization of Numerical Results:**
> We thank the reviewer for the observation regarding Figure 1 and Table 3. We initially grouped them due to page constraints, but we agree that separating the Dictionary Learning (NC-C) and DRO (C-SC) results into distinct parts will significantly improve readability. We will address this in the final version.
>
> We thank the reviewer again for their positive and insightful feedback.

---

> > ### Author Rebuttal · Reviewer_P2FM · 2026-04-04
> >
> > Thanks to the authors for the paper! Vote for acceptance! Good luck!

---

### Official Review · Reviewer_mVGC · 2026-03-13

**Soundness:** 2
**Presentation:** 2
**Significance:** 2
**Originality:** 3
**Overall Recommendation:** 4
**Confidence:** 3

**Summary:**

This paper develops a unified projection-free framework for solving minimax optimization problems, particularly in convex-concave and nonconvex-concave settings. Building on a dual dynamic smoothing approach, the authors design single-loop algorithms that rely on linear minimization oracles (LMO) and projection oracles (PO), yielding LMO-LMO, LMO-PO, and PO-LMO variants. The analysis establishes convergence guarantees for a range of structural regimes, including convex, strongly convex, and nonconvex cases. Numerical experiments on dictionary learning and robust classification demonstrate the effectiveness and stability of the proposed methods.

**Compliance With Llm Reviewing Policy:**

Affirmed.

**Final Justification:**

All of my comments has been addressed. Hence, I have increased the score to 4.

**Key Questions For Authors:**

Please address my concerns and comments provided in the weakness section. I will increase the score if all of my comments are properly taken care of.

**Limitations:**

There are no limitations or potential negative societal impact of this work.

**Strengths And Weaknesses:**

**Strengths:**

The paper is well written, has a lot of results.

**Weaknesses:**

I am addressing my questions/comments here.

1. The paper compares projection free methods and talks about LMO computation. Can LMO be computed for all sublevel sets of convex functions? What is the complexity of LMO computation?

2. In some algorithms combination of LMO and PO oracles are used. So there should be an inherent assumption that the projection oracle is easy to compute and same for LMO. If PO is used, then how are the "Projection-free" words in the title justified?

3. The paper compares several projection-free methods in the literature review. It would also be valuable to include comparisons with the ADMM-based paper "Solving Constrained Variational Inequalities via a First-Order Interior Point-Based Method" and the randomized feasibility-update method "Randomized Feasibility-Update Algorithms for Stochastic Variational Inequality Problems", both of which study variational inequalities for monotone maps and are also projection-free approaches.

4. I am curious why Assumption 1.2 is required and where it is used in the analysis. Are the improved rates in certain cases derived specifically because of this assumption, or can those results be obtained without it?

5. The quantity $G_{\mathcal{Y}}(x,y)$ defined in Equation (3) is used as a merit function in all the theorems when stating convergence rates. I am unclear why this function constitutes a valid merit function for the saddle-point problem. For a function to serve as a merit function, one would typically require that $G_{\mathcal{Y}}(x*,y*) = 0$ for all $(x*,y*)$ in the solution set and $G_{\mathcal{Y}}(x,y) > 0$ for all other $(x,y)$. I have similar concerns regarding the merit functions $G_{\mathcal X}^{\mathrm{LMO}}(x,y)$ and $G_{\mathcal X}^{\mathrm{PO}}(x,y)$ defined in Equations (4a) and (4b), which are also used to establish convergence rates.

At a high level, it appears that $G_{\mathcal{Y}}(x,y)$ measures optimality with respect to $\mathcal{Y}$ for a fixed $x$, while $G_{\mathcal X}^{\mathrm{LMO}}(x,y)$ and $G_{\mathcal X}^{\mathrm{PO}}(x,y)$ measure optimality with respect to $\mathcal{X}$. However, the standard merit function for saddle-point problems is typically the dual gap $G_{\mathcal X,\mathcal Y}^{\mathrm{Dual}}(x,y)$. I would therefore appreciate clarification on the validity of these alternative merit functions, since they do not appear to satisfy the usual definition of a merit function for the full saddle-point problem.

6. In the simulation section, it would be helpful to compare the proposed methods with other fast-converging algorithms, such as ADMM.

---

> ### Author Rebuttal · Authors · 2026-03-30
>
> We thank the reviewer for their detailed feedback and the opportunity to clarify the core components of our work. We would like to highlight that we intend to analyze a central theme regarding the unified design of projection-free structures, specifically for settings where we may not have access to projection oracles over $\mathcal{X}$, $\mathcal{Y}$, or both.
>
> **1. On LMO Computation and Complexity:**
> We appreciate the opportunity to clarify the definition of the Linear Minimization Oracle (LMO). In the context of projection-free optimization (Jaggi, 2013), the LMO is defined over the **feasible sets** $\mathcal{X}$ and $\mathcal{Y}$, rather than the sublevel sets of the objective function. The primary motivation for using an LMO is that for many complex sets, it is significantly cheaper than a Projection Oracle (PO). For example:
>
> *   **Nuclear norm ball (used in our experiments):** The LMO requires finding only the leading singular vector (via the power method/Lanczos), whereas a PO requires a full SVD.
>
> *   **$\ell_1$-ball:** The LMO is a simple search for the maximum coordinate, while the PO requires a more complex sort-and-shift operation.
> We would like to stress that our framework **does not** require any specific properties for the sublevel sets of the payoff function $\mathcal{L}$.
>
> **2. Justification of the Title "Projection-Free":**
> The reviewer makes a valid point regarding the hybrid algorithms. We chose this title because our primary contribution is the **first single-loop, anytime framework for the fully projection-free (LMO-LMO) setting** in nonconvex-concave problems (Algorithm 1). We include the LMO-PO and PO-LMO variants (Algorithms 2 and 3) to provide a complete, unified framework that adapts to the specific needs of a problem; for example, when one set is a simple box (easy to project) but the other is a nuclear norm ball (requires an LMO).
>
> **3. Comparison with ADMM and Randomized VI:**
> We thank the reviewer for pointing us to these interesting directions. ADMM-type methods and randomized feasibility methods are powerful, but they often operate on different assumptions. Specifically, ADMM typically requires solving a **proximal subproblem** at each step, which, for the complex constraints we study, is even more expensive than projection. Furthermore, most existing VI methods are primarily designed for the **monotone (convex-concave) setting**, whereas a core contribution of our work is addressing the **more general nonconvex-concave** regime. Our framework is designed to avoid those expensive subproblems entirely. To address the reviewer's comment, we will add a discussion of these works in Section 1 to place our contribution in the wider context of Variational Inequalities.
>
> **4. The Role of Assumption 1.2 (Strongly Convex Sets):**
> We would like to clarify that Assumption 1.2 is **not necessary** for the convergence of our algorithms. As shown in Theorems 2.1, 3.1, and 4.1, our methods converge under standard smooth assumptions. We only introduce Assumption 1.2 as an **optional geometric property** that allows our framework to achieve **accelerated rates** (we refer the reviewer to Table 1). If the set is not strongly convex, the algorithm still functions correctly at the standard rate.
>
> **5. Validity of Merit Functions ($G_X, G_Y$):**
> The reviewer's concern regarding the duality gap $G_{\rm Dual}$ is very relevant. We chose to use $G_X$ and $G_Y$ for two reasons:
>
> *   **Nonconvexity:** In the nonconvex-concave regime, the standard duality gap may not vanish at local optima, making it an unreliable measure of progress. $G_X$ and $G_Y$ (representing "$\epsilon$-game stationarity") are the standard benchmarks in recent literature (Boroun et al., 2023; Xu et al., 2023).
>
> *   **Link to Duality Gap:** For the convex-concave case, we prove in **Lemma 1.5** that our measures **directly bound the standard duality gap**. Thus, by proving rates for $G_X$ and $G_Y$, we are implicitly providing rates for $G_{\rm Dual}$. Furthermore, by regularizing the dual, we are able to analyze the **primal optimality gap** ($G_{\rm Opt}$) in the convex setting, which yields even faster rates.
>
> **6. Comparison with Fast Methods like ADMM in Simulations:**
> In our experiments, we focus on **single-loop** methods where the cost per iteration is dominated by the LMO. An ADMM implementation for Dictionary Learning would require a full SVD at each iteration to handle the nuclear norm constraint and leads to a double loop algorithm. Our LMO-based approach requires only the leading singular vector, which is orders of magnitude faster per iteration. Furthermore, ADMM is typically motivated by functional constraints (e.g., $Ax+By=c$), which are absent in our set-constrained formulation.
>
> We thank the reviewer again. If there are any further points we could clarify, or any additional suggestions that could help improve the manuscript and your assessment of the work, we would be very happy to address them.

---

> > ### Author Rebuttal · Reviewer_mVGC · 2026-04-03
> >
> > Thank you for the detailed rebuttal. Most of my concerns have been addressed.
> >
> > However, I still do not fully understand the clarification regarding the validity of the merit function. Can it be formally shown that demonstrating progress with respect to only $G_X$ or only $G_Y$ implies $\epsilon$-stationarity of the game? I am not convinced that this is sufficient. Typically, one needs to combine stationarity measures over both sets $X$ and $Y$ and define a merit function on the Cartesian product $X \times Y$ to properly characterize convergence to a saddle point.
> >
> > For instance, in Xu et al. (2023), a combined merit function is used to capture this joint stationarity. Whereas in this current work, separate convergence rates are established with respect to $G_X$ and $G_Y$. Could you clarify whether these guarantees independently imply $\epsilon$-stationarity, or whether they should be combined into a single merit function against which convergence should be measured?

---

> > > ### Author Response · Authors · 2026-04-03
> > >
> > > We sincerely thank the reviewer for this important clarification. The reviewer is correct that Xu et al. (2023) use a combined merit function to capture joint stationarity. This is also the case in Boroun et al. (2023), where the overall stationarity measure is given by the sum of the stationarity measures with respect to $x$ and $y$. In our setting, the natural combined merit is precisely $G_X+G_Y$, where $G_X$ denotes the primal stationarity measure (either $G_X^{\mathrm{LMO}}$ or $G_X^{\mathrm{PO}}$, depending on the oracle used) and $G_Y$ is the dual gap. Since our analysis establishes convergence guarantees for these two quantities **simultaneously**, the same order immediately yields a guarantee for the combined merit $G_X+G_Y$. We also emphasize that the numerical results presented in Figure 1 are already based on the combined merit function, as stated on line 403 (page 8, second column). We will revise the manuscript to make the definition of this combined form more explicit for the reader.
> > >
> > > More specifically, in the convex-concave setting, Lemma 1.5 shows that the standard saddle-point duality gap at the averaged iterates is bounded by the average combined merit $G_X^{\mathrm{LMO}}+G_Y$. Moreover, by Lemma 1.3, when the primal variable is updated via a projection oracle, $G_X^{\mathrm{PO}}$ can be converted into $G_X^{\mathrm{LMO}}$ up to a constant factor, so the same conclusion also applies to the hybrid PO-based variants. In this sense, $G_X+G_Y$ directly controls the duality gap.
> > >
> > > We also clarify the connection to $\epsilon$-game stationarity, which is discussed in Remark B.3. In particular, Remark B.3 explains why the oracle-based gaps used in our analysis imply the standard approximate stationarity conditions for the game. On the dual side, a bound on $G_Y$ means that the y-player cannot improve the payoff by more than $\epsilon$, and in the strongly concave case it further implies the corresponding dual gradient inclusion. On the primal side, a bound on $G_X^{\mathrm{LMO}}$ yields the primal stationarity inclusion, while Lemma 1.3 transfers the same conclusion to the projection-based setting through $G_X^{\mathrm{PO}}$. Therefore, controlling the combined merit $G_X+G_Y$ captures the joint notion of $\epsilon$-game stationarity that the reviewer is asking about.
> > >
> > > We presented $G_X$ and $G_Y$ separately mainly to distinguish the LMO and PO settings for the primal update, and because $G_Y$ leads to a finer dual analysis in the convex-concave regime. However, we agree that presenting the guarantees explicitly in terms of the combined merit $G_X+G_Y$ would improve clarity. In the revision, we will expand the discussion around Lemma B.2 and Remark B.3 and incorporate it into Section 1.2.1 so that this connection is stated upfront.

---

### Official Review · Reviewer_71Jc · 2026-03-15

**Soundness:** 3
**Presentation:** 2
**Significance:** 3
**Originality:** 3
**Overall Recommendation:** 4
**Confidence:** 3

**Summary:**

This paper studies constrained smooth minimax optimization and tries to connect projection-based and projection-free methods under a unified framework. The proposed dual dynamic smoothing idea avoids the inner maximization step, and the paper develops three single-loop variants under different oracle settings: LMO-LMO, LMO-PO, and PO-LMO. The topic is relevant, and the theoretical part is generally well developed.

**Compliance With Llm Reviewing Policy:**

Affirmed.

**Ethical Review Concerns:**

None.

**Final Justification:**

Thank you for the author's feedback; I will maintain my score.

**Key Questions For Authors:**

Please refer to what is described in "Strengths And Weaknesses".

**Limitations:**

Please refer to what is described in "Strengths And Weaknesses".

**Strengths And Weaknesses:**

I think the paper has a solid theoretical framework, and the unified view across different oracle settings is a useful contribution. The presentation is also mostly clear.

My main concerns are the following.

First, the definition of the stationarity measure needs more explanation. In Eq. (4b), (G_X^{PO}) uses a fixed parameter (\sigma>0), but in Theorem 4.1 it seems to be tied to the initial step size (\tau_0). Since the algorithm uses a decaying step size (\tau_t), it would be helpful to explain why this fixed choice is appropriate, or to discuss whether an iteration-dependent choice such as (\sigma=\tau_t) is possible.

Second, the experiments do not clearly isolate the effect of dynamic smoothing. Since (\beta_t) is presented as a key part of the method, I think the paper should include a simple ablation comparing dynamic smoothing with a fixed smoothing parameter, and possibly with no smoothing when applicable.

Third, some empirical claims should be stated more carefully. For example, the text says that LMO-PO is the most effective configuration, but Figure 1 suggests that PO-LMO performs better at some points. I would suggest making this statement more precise.

Finally, the paper emphasizes the anytime and practical nature of the method, but Algorithms 1 and 2 require knowledge of the global constant (L_{yy}) for step-size selection. In practice, this quantity may be difficult to know or estimate well. This limitation should be acknowledged more clearly.

---

> ### Author Rebuttal · Authors · 2026-03-30
>
> We thank the reviewer for their constructive feedback and the positive assessment of our theoretical framework. We address the specific concerns below:
>
> *   **On the Stationarity Measure ($\sigma$ vs. $\tau_t$):** We chose to define $G_X^{PO}$ with a fixed $\sigma = \tau_0$ to ensure a consistent, non-varying metric for benchmarking convergence across the entire trajectory. If $\sigma$ varied with $t$, the **target** itself would shift, making the $O(\cdot)$ rates harder to interpret as physical progress. However, we confirm that our analysis holds for any fixed $\sigma > 0$. Redefining it as $\sigma = \tau_t$ is technically feasible and yields equivalent rates (up to constant factors), but we believe the fixed choice is more standard for stationarity analysis. We will clarify this choice in Section 1.2.1, and can revise the definition to $\sigma = \tau_t$ if the reviewer prefers.
>
> *   **Effect of Dynamic Smoothing:** We appreciate the suggestion. Our current comparisons already provide some supporting evidence. In particular, the experiments against Boroun et al. (2023) illustrate the difference between **dynamic smoothing** and a **fixed smoothing** parameter. We also include the method of Gidel et al. (2017), which **does not employ any smoothing**. These results indicate that dynamic smoothing improves early-to-mid-stage convergence compared to a fixed $\beta$ tuned for high-accuracy regimes. Empirically, we also observe convergence to a better final solution, as the algorithm benefits from its anytime nature and is not hindered by slow progress in the early-to-mid phases. We will expand the discussion in the numerical section to provide a more details on the impact of the dynamic smoothing framework.
>
> *   **Empirical Claims:** The reviewer is correct that the performance of PO-LMO and LMO-PO varies across different phases and datasets. Our statement that LMO-PO is most effective refers to its superior stability and final convergence accuracy. We will revise the text to more precisely characterize these trade-offs, specifically noting where PO-LMO shows competitive early-stage progress.
>
> *   **Knowledge of $L_{yy}$ and Practical Implementation:** We thank the reviewer for this insightful practical point. The requirement of knowing the global Lipschitz constant $L_{yy}$ is a standard assumption in the analysis of first-order minimax methods, but we agree that it can be a limitation in practice. However, our framework is fully compatible with a standard backtracking line-search to estimate a local Lipschitz constant $\hat{L}_{yy}$ at each iteration. This allows the algorithms to be tune-free regarding global constants without changing the convergence rates. We will add a Remark in the paper detailing this. In prticular,
>     *   **For Dual LMO Updates (Algorithms 1 and 3):** The step size $\gamma_t$ (and the LMO progress) relies on the quadratic upper bound established in **Lemma A.8 (Eq. 15)**. One can search for a local $\hat{L}\_{yy}$ that satisfies
> $$-\mathcal{L}\_t (x\_t, y\_{t+1})  \leq -\mathcal{L}\_t(x\_t, y\_t) - \gamma\_t \nabla\_y \mathcal{L}\_t(x\_t, y\_t)^\top (u\_t - y\_t) + \frac{\hat{L}\_{yy} + \beta\_t}{2} \gamma_t^2 \|\|u\_t - y\_t \|\|^2$$
> This ensures that the dual progress used to bound the discrepancy $H_t$ in Lemma A.8 remains valid. From a practical standpoint, $\hat{L}\_{yy}$ can be estimated efficiently via the backtracking line-search procedure of **Pedregosa et al. (2020)**.
>
>     *   **For Dual PO updates (Algorithm 2):** Backtracking for projected gradient steps is well-established. As shown in the proof of **Lemma A.12**, the dual projection ensures progress as long as $\gamma_t \leq 1/(L_{yy} + \beta_t)$. One can use backtracking to find a $\hat{L}\_{yy}$ such that:
>     $$-\mathcal{L}\_t(x\_t, y\_{t+1}) \leq -\mathcal{L}\_t(x\_t, y\_t) - \nabla\_y \mathcal{L}\_t(x_t, y_t)^\top (y\_{t+1} - y\_t) + \frac{\hat{L}\_{yy} + \beta\_t}{2} \|\| y\_{t+1} - y\_t \|\|^2$$
>     This condition (equivalent to the one in Lemma A.12 rewritten for the objective $-\mathcal{L}$) mirrors standard proximal gradient line-search procedures described in **Beck (2017, Section 10.4.2)** and ensures the descent required for the rates in Theorem 3.1 and 3.2.
>
>     Given the length of the paper, we decided to have the current presentation using $L_{yy}$ for clarity of the core framework. Nonetheless, we will add a Remark about the application of line-search algorithms to ensure a practical tune-free version.
>
> We thank the reviewer again for their insightful feedback. We hope our response and planned revisions have addressed the main concerns. If there are any further points we could clarify, or any additional suggestions that could help improve the manuscript and your assessment of the work, we would be very happy to address them.
>
> [//]: <> (We were wondering if there are any other points we can clarify that could further improve the manuscript and your assessment of the work?)

---

> > ### Author Rebuttal · Reviewer_71Jc · 2026-04-01
> >
> > Thank you for the detailed rebuttal. I appreciate the clarifications on points (1), (3), and (4), and I do not have further questions on those aspects.
> >
> > Regarding point (2), I find the response thoughtful, but it does not fully resolve my concern about whether the paper cleanly isolates the effect of the smoothing component itself. In particular, I still think a more direct ablation comparing dynamic smoothing against a fixed smoothing parameter would be important to support the paper’s empirical claims.
> >
> > Overall, the rebuttal is helpful, but this experimental concern remains.

---

> > > ### Author Response · Authors · 2026-04-02
> > >
> > > We thank the reviewer for their feedback. We address the concerns raised below:
> > >
> > > *   **Effect of Dynamic Smoothing:** We would like to note our second experiment on datasets `RCV1` and `NEWS20` belongs to convex-strongly concave settings so we do not implement any smoothing scheme. Thus, to investigate the effect of smoothing, we revisit the dictionary learning experiment, which belongs to nonconvex-concave settings. Specifically, for each of `LMO-LMO`, `LMO-PO`, `PO-LMO` methods, we implement three versions which are dynamic smoothing, smoothing with fixed parameter $0.001$ and no smoothing (i.e., smoothing with fixed parameter $0$). Below, a summary of our experiment after running each algorithm for $5$ minutes is reported. This further shows that our algorithm under dynamic smoothing has better empirical performance against other alternatives
> > >
> > > | Method | Dynamic | Static | No smoothing |
> > > | :--- | :---: | :---: | :---: |
> > > | `LMO-LMO` | 1.791e-15 | 8.82e-5 | 8.82e-5|
> > > | `LMO-PO` | 1.86e-16 | 1.99e-16 | 2.19e-16 |
> > > | `PO-LMO` | 1.74e-5 | 1.74e-3 | 1.81e-3 |
> > >
> > >
> > > We thank the reviewer again for their insightful feedback. We hope our response and planned revisions have addressed the main concerns. If there are any further points we could clarify, or any additional suggestions that could help improve the manuscript and your assessment of the work, we would be very happy to address them.

---

### Decision · Program_Chairs · 2026-04-30

**Decision:**

Accept (regular)

**Comment:**

The paper presents a solid and well-developed theoretical framework for projection-free algorithms in constrained minimax optimization. The unified dual dynamic smoothing perspective, together with the coverage of multiple oracle settings (LMO, PO, and hybrid variants) and problem classes (nonconvex–concave, nonconvex–strongly concave, and convex–concave), is viewed as a meaningful contribution. The analysis is generally clear, technically sound, and broad in scope, and the resulting anytime convergence guarantees are appreciated.

That said, several concerns limit the overall impact. The definition of the stationarity measure and its dependence on algorithm parameters requires further justification, and some assumptions may limit practical applicability. Also, the experimental evaluation is not fully convincing: it does not clearly isolate the role of dynamic smoothing, contains some presentation inconsistencies, and would benefit from better organization and more careful empirical claims.

Overall, the paper makes a solid theoretical contribution. The issues above are relatively minor compared to the contribution of this work. Therefore, an acceptance is recommended.